# Multiproxy evidence of the Neoglacial expansion of Atlantic Water to eastern Svalbard

Joanna Pawłowska[1*], Magdalena Łącka[1], Małgorzata Kucharska[1], Jan Pawlowski[1,2], Marek Zajączkowski[1]

[1]Institute of Oceanology Polish Academy of Sciences, Sopot, 81-712, Poland

[2]Department of Genetics and Evolution, University of Geneva, Geneva, CH 1211, Switzerland

*Correspondence to*: Joanna Pawłowska (pawlowska@iopan.pl)

**Abstract.** The main goal of this study is to reconstruct the paleoceanographic development of Storfjorden during the Neoglacial (~ 4 cal ka BP). Storfjorden is one of the most important brine factories in the European Arctic and is responsible for deep water production. Moreover, it is a climate-sensitive area influenced by two contrasting water masses: warm and saline Atlantic Water (AW) and cold and fresh Arctic Water (ArW). Herein, a multiproxy approach was applied to provide evidence for existing interactions between the inflow of AW and sea-ice coverage, which are the major drivers of environmental changes in Storfjorden. The sedimentary and microfossil records indicate that a major reorganization of oceanographic conditions in Storfjorden occurred at ~ 2.7 cal ka BP. The cold conditions and the less-pronounced presence of AW in Storfjorden during the early phase of the Neoglacial were the prerequisite conditions for the formation of extensive sea-ice cover. The period after ~ 2.7 cal ka BP was characterized by alternating short-term cooling and warming intervals. Warming was associated with pulsed inflows of AW and sea-ice melting that stimulated phytoplankton blooms and organic matter supply to the bottom. The cold phases were characterized by heavy and densely packed sea ice resulting in decreased productivity. The ancient environmental DNA (aDNA) records of foraminifera and diatoms support the occurrence of the major pulses of AW (~2.3 and ~1.7 cal ka BP) and the variations in sea-ice cover. The episodes of enhanced AW inflow were marked by an increase in the percentage of DNA sequences of monothalamous foraminifera associated with the presence of fresh phytodetritus. Cold and less productive intervals were marked by an increased proportion of monothalamous taxa

known only from environmental sequencing. The diatom aDNA record indicates that primary production was continuous during the Neoglacial, regardless of the sea-ice conditions. However, the colder periods were characterized by the presence of diatom taxa associated with sea ice, whereas the present-day diatom assemblage is dominated by open-water taxa.

## 1. Introduction

The northward flow of Atlantic Water (AW) is one of the major contributors of heat to the Arctic Ocean (Polyakov et al., 2017). Recent oceanographic data indicate a warming trend due to an increased inflow of AW towards the Arctic Ocean (Rudels et al., 2015, Polyakov et al., 2017). AW has been present along the western margin of Svalbard for at least the last 12,000 years (e.g., Werner et al., 2011; Rasmussen et al., 2014). One of the major intrusions of AW occurred during the early Holocene (10.8 – 6.8 cal ka BP). A distinct cooling and freshening of the west Spitsbergen shelf bottom water masses occurred during the mid-late Holocene (6.8 – 1 cal ka BP) and was accompanied by glacier readvances in Svalbard, leading to the present-day conditions (Ślubowska-Woldengen et al., 2007; Telesiński et al., 2018). The paleoceanographic conditions in the Svalbard margins correlate closely to the sea surface temperature (SST) variations in the Nordic Seas and confirm that the Svalbard area is highly sensitive to fluctuations in the inflow of AW (Ślubowska-Woldengen et al., 2007; Werner et al., 2013). Conversely, until the 1990s eastern Svalbard was recognized as an area exclusively influenced by the East Spitsbergen Current (ESC), which carries cold, less saline Arctic Water (ArW) from the Barents Sea (e.g., Quadfasel et al., 1988; Piechura et al., 1996). However, recent studies have revealed that the oceanography of the area is much more complicated (e.g., Skogseth et al., 2007; Geyer et al., 2010). Oceanographic data obtained from conductivity–temperature sensors attached to *Delphinapterus leucas* show a substantial contribution of AW to Storfjorden, East Spitsbergen (Lydersen et al., 2002). Recently, Hansen et al. (2011) suggested the presence of AW in Storfjorden during the early Holocene warming (11 – 6.8 cal ka BP), which was further confirmed by the foraminiferal and sedimentary records of Łącka et al. (2015).

The latter part of the Holocene, the so-called Neoglacial cooling (~ 4 cal ka BP), in the European Arctic is characterized by a declined summer insolation at northern latitudes (Berger, 1978) that correlates to a decline in summer SST (e.g., Andersen et al., 2004; Risebrobakken et al., 2010; Rasmussen et al., 2014; Ivanova et al., 2019). The cooling of the surface waters and the limited AW inflow towards the Nordic Seas led to the formation of an extended sea-ice cover in West Spitsbergen margin (Müller et al., 2012). In addition, the

southwestern and eastern shelf of Spitsbergen experienced a strengthening of the East Spitsbergen Current and/or Jan Mayen Current leading to an intensification of ArW inflow and the formation of extensive sea-ice cover (e.g., Sarnthein et al., 2003; Berben et al., 2014). Therefore, the Neoglacial is usually considered a generally cold period (e.g., Consolaro et al., 2018). However, the records from Storfjorden and the Barents Sea suggest that the Neoglacial was a period of variable oceanographic conditions with strong temperature and salinity gradients (Martrat et al., 2003; Risebrobakken et al. 2010; Sarnthein et al., 2003; Łącka et al., 2015; 2019). In addition, there is evidence of episodic intensifications of the warm AW inflow towards western Svalbard at that time (e.g.; Rasmussen et al., 2012).

According to Nilsen et al. (2008), the critical parameter controlling the fjord–shelf exchange is the density difference between the fjord water masses and the AW. The local winter ice production and the formation of brine-enriched waters determine the density of local water masses, which is a key factor that enables AW to penetrate into fjords during the spring and summer. Moreover, the production of brine-enriched waters and the associated deep-water overflow are key contributors to large-scale ocean circulation (Killworth, 1983). In this respect, Storfjorden is especially important because it is one of the few areas where brine-enriched waters have been frequently observed (Haarpainter et al., 2001). In recent decades, reduced brine formation has occurred during the periods with the most intensive AW advection to Storfjorden and less sea-ice formation in the Barents Sea, while intense brine formation re-establishes during periods of recurrent cooling (Årthun et al., 2011;Rasmussen and Thomsen, 2014).

The aim of this study is to reconstruct the paleoceanographic development of Storfjorden during the Neoglacial at multicentennial resolution. We assumed that the periodic intensification of the AW inflow to the West Spitsbergen shelf during the Neoglacial resulted in the appearance of AW also in eastern Spitsbergen, similar to the conditions in the early Holocene (e.g., Łącka et al., 2015), affecting the density and extent of sea-ice cover in the area. A multiproxy approach comprising sedimentary, microfossil and molecular records was applied to provide evidence for the interactions between the inflow of AW and sea-ice coverage in Storfjorden. The ancient environmental DNA (aDNA) analysis targeted diatoms and nonfossilized monothalamous foraminifera. Both these of groups are hardly preserved in fossil records from the Svalbard fjords (Pawłowska et al., 2014) and shelf areas (Zimmermann et al., 2019 and references therein). Recent studies have demonstrated that analyses of genetic material obtained directly from environmental samples (so-called environmental DNA) are an efficient method for performing biodiversity surveys across time and space (Thomsen and

Willerslev, 2015). The content of environmental DNA samples may be analyzed by DNA metabarcoding, which consists of high-throughput sequencing of taxonomically informative DNA fragments called metabarcodes. The identification of short, species-specific DNA fragments (so called "barcodes") allows us to obtain species-level assignments of modern and ancient DNA sequences (Herbert et al., 2003). The further demonstration that DNA can be preserved in the environment across geological timescales opened new avenues for palaeoclimatic and palaeoceanographic studies. Recent studies have demonstrated the preservation of DNA in marine sediments for tens to hundreds thousands of years. An aDNA approach was successfully applied to trace the Holocene history of dinoflagellates, haptophytes (e.g., Coolen et al., 2009, 2013; Boere et al., 2009) and foraminifera in deep sea (Lejzerowicz et al., 2013) and coastal areas (Pawłowska et al., 2014; 2016). The study of Pawłowska et al. (2016) was the first attempt to utilize foraminiferal aDNA as a paleoenivronmental proxy. This study supported the existence of extremely diverse foraminiferal assemblages. The richness of the foraminiferal community revealed by the molecular record was much higher than that in the fossil record (Pawłowska et al., 2014), mainly due to the detection of nonfossilized monothalamous taxa. The molecular data correlated well with environmental changes and revealed even small changes that were not clearly indicated by other proxy records. The combination of aDNA studies with the analysis of microfossils and sedimentary proxies provides a powerful means to reconstruct past environments more comprehensively.

## 2. Study area

Storfjorden is located in southeastern Svalbard between the islands of Spitsbergen, Edgeøya and Barentsøya (Fig. 1). Storfjorden is ~190 km long and its main basin is ~190 m deep. Two narrow and shallow passages (Heleysundet and Freemansundet) connect northern Storfjorden to the Barents Sea. To the south, a 120-m-deep sill separates the main basin from the Storfjordrenna. Storfjordrenna is 245 m long, with a depth varying from 150 m to 420 m.

The water masses in Storfjorden are composed primarily of exogenous Atlantic and Arctic waters as well as mixed waters that have formed locally. The Atlantic Water is transported northwards by Norwegian Atlantic Current (NwAC), which branches off in the Barents Sea into the West Spitsbergen Current and North Cape Current (Loeng et al. 1991; Blindheim and Østerhus, 2005) (Fig. 1a). In addition to AW, cold and less saline Arctic Water (ArW) is transported to the Barents Sea via East Spitsbergen Current and Bear Island Current (Hopkins, 1991) (Fig. 1a). Arctic water (ArW) from the Arctic Ocean and the Barents Sea

enters Storfjorden via two passages to the northeast and continues along the inner shelf of Svalbard as a Coastal Current (Fig. 1b). AW is characterized by temperatures > 3 °C and salinity > 34.95, while the temperature and salinity of ArW are < 0 °C and 34.3-34.8, respectively. The presence of locally formed water masses is a result of the interactions between AW, ArW and melt water. Skogseth et al. (2005) listed six local water masses: melt water (MW), polar front water (PW), East Spitsbergen water (ESW), brine-enriched shelf water (BSW), Storfjorden surface water (SSW), and modified Atlantic water (MAW). BSW is formed due to the release of a large amounts of brines during polynya events and the intensive formation of sea ice (Haarpainter et al., 2001; Skogseth et al., 2004, 2005) and is characterized by salinities exceeding 34.8 and temperatures below -1.5 °C (Skogseth et al., 2005).

The sedimentary environment in Storfjorden is classified as a low-energy, high-accumulation environment, which is characteristic of inner fjords. The area is sheltered from along-shelf bottom currents and is affected by high terrigenous inputs; therefore deposition prevails over sediment removal by bottom currents (Winklemann and Knies, 2005). The primary productivity is high and strongly depends on the sea-ice formation as well as the seasonal duration of the marginal ice zone (Winkelman and Knies, 2005).

## 3. Materials and methods

### 3.1 Marine sediment core

The 55-cm-long sediment core ST_1.5 was taken with a gravity corer in Storfjorden retrieved with the R/V *Oceania* in August 2014. The sampling station was located at 76° 53,181' N and 19° 27,559' E at a depth of 153 m (Fig. 1). The salinity and temperature of the water column at the coring station was measured with a Mini CTD Sensordata SD 204 at intervals of 1 s (Fig. 2). The core was stored at 4°C and shipped to the Institute of Oceanology PAS for further analyses.

In the laboratory, the core was extruded and cut into 1-cm slices. During cutting, sterile subsamples for ancient DNA (aDNA) analyses were taken at 4 cm intervals. To avoid external and/or cross-contamination the thin layers of sediment that were in contact with under- or overlying sediments were removed using a sterile spatula. Samples for aDNA analyses were kept frozen at -20°C. Samples for other proxy analyses were taken every 2 cm.

### 3.2 Chronology

The chronology of the marine sediment core is based on high-precision accelerator mass spectrometry (AMS) $^{14}$C dating performed on five bivalve shells retrieved from the sediment layers at 2.5, 5.5, 14.5, 43.5, and 52.5 cm core depth and on the foraminifera *Nonionellina labradorica* from the 46.5 cm core depth. The bivalve shells were identified to the highest possible taxonomic level and processed on the 1.5 SDH-Pelletron Model "Compact Carbon AMS" in the Poznań Radiocarbon Laboratory, Poznań, Poland. Dating of foraminiferal tests was performed at the National Ocean Sciences AMS (NOSAMS) laboratory in the Woods Hole Oceanographic Institution, Woods Hole, MA, USA. The dates were converted into calibrated ages using the calibration program CALIB Rev. 7.1.0 Beta (Stuiver and Reimer, 1993) and the Marine13 calibration dataset (Reimer et al., 2013). A reservoir age correction (ΔR) of 105 ± 24 was applied (Mangerud et al., 2006). The calibrated results are reported in units of thousand calibrated years BP (cal ka BP) (Table 1).

## 3.3 Grain size analysis

The samples for grain size analyses were freeze-dried and milled. The measurements were performed using a Mastersizer 2000 particle laser analyzer coupled to a Hydro MU device (Malvern, UK). The samples were treated with ultrasound to avoid aggregation. The raw data were analyzed using GRADISTAT v.8.0 software (Blott and Pye, 2001). The mean 0-63-µm grain size [φ] was calculated via the logarithmic method of moments. The sediment fraction >500 µm was used to reconstruct an ice rafted debris (IRD) record. The grains were counted under a stereomicroscope and the amount of IRD is reported as the concentration (i.e., the number of grains per gram of dry sediment) [grains g$^{-1}$] and the flux [grains cm$^{-2}$ y$^{-1}$].

## 3.4 Benthic foraminifera assemblages

Prior to the analysis of testate benthic foraminifera, samples were wet sieved through a meshes with 500-µm and 100-µm openings and dried at 60°C. Samples with large quantities of tests were divided using a microsplitter. At least 300 specimens of benthic foraminifera were isolated from each sample and collected on micropaleontological slides. Benthic foraminifera specimens were counted and identified to the lowest possible taxonomic level. The quantity of foraminifera is presented as the concentration (i.e., the number of individuals per gram of dry sediment) [ind. g$^{-1}$] and the flux [ind. cm$^{-2}$ y$^{-1}$]. Foraminifera species were grouped according to their ecological tolerances. Four groups of indicators were distinguished: AW/frontal zone indicators, ArW indicators, bottom current indicators and

glaciomarine species (Majewski et al., 2009). The morphologically similar species *Islandiella norcrossi* and *Islandiella helenae* are reported as *Islandiella* spp.

**3.5 Stable isotope analysis**

Carbon and oxygen stable isotope analyses were performed on *Cibicidoides lobatulus* tests selected from 27 sediment layers. Ca. 10 to 12 specimens were collected from each sample and subjected to ultrasonic cleaning. The measurements were performed on a Finnigan MAT 253 mass spectrometer coupled to a Kiel IV carbonate preparation device at the University of Florida. The resulting values are expressed in standard δ notation relative to Vienna Pee Dee Belemnite (VPDB).

**3.6 Ancient DNA analysis**

The total DNA was extracted from approximately 10 g of sediment using a Power Max Soil DNA extraction kit (MoBio). The foraminiferal SSU rDNA fragments containing the 37f hypervariable region were PCR amplified using primers tagged with unique sequences of five nucleotides appended to their 5' ends (denoted by Xs), namely, the foraminifera-specific forward primer s14F1 (5'-XXXXXCGGACACACTGAGGATTGACAG-3') and the reverse primer s15 (5'-XXXXXCCTATCACATAATCATGAAAG-3'). The diatom DNA fragment located in the V4 region was amplified with the forward DIV4for (5'-XXXXXXXXGCGGTAATTCCAGCTCCAATAG-3') and reverse DIV4rev3 (5′-XXXXXXXXCTCTGACAATGGAATACGAATA-3') primers tagged with a unique combination of eight nucleotides (denoted by Xs) attached at each primer's 5'-end. The amplicons were purified using the High Pure PCR Cleanup Micro Kit (Roche) and quantified using a Qubit 2.0 fluorometer. Samples were pooled in equimolar quantities, and the sequence library was prepared using a TruSeq library-preparation kit (Illumina). The samples were then loaded into a MiSeq instrument for a paired-end run of 2*150 cycles (foraminifera) and 2*250 cycles (diatoms). The processing of the HTS sequence data was performed according to procedures described by Lejzerowicz et al. (2013) and Pawłowska et al. (2014). The post-sequencing data processing was performed with the use of the SLIM web app (Dufresne et al., 2019) and included demultiplexing the libraries, joining the paired-end reads, chimera removal, operational taxonomic units (OTUs) clustering, and taxonomic assignment. Sequences were clustered into OTUs using the Swarm module (Mahe et al. 2014), and each OTU was assigned to the highest possible taxonomic level using vsearch (Rognes et al., 2016) against a local database and then reassigned using BLAST (Altschul et al., 1990). The results

1  are presented in OTU-to-sample tables and transformed in terms of the number of sequences,

2  number of OTUs and percentage (%) of sequences.

## 4. Results

### 4.1 Chronology

In total, six radiocarbon dates were obtained, all of which were recorded in

chronological order. The uppermost layer contained modern, post-bomb carbon indicating a

post-1960 age (Table 1). Samples from the 2.5 cm and 5.5 cm core depths were not calibrated

because they revealed ages invalid for the selected calibration curve. The age model was,

therefore, based on the four remaining dates using a linear interpolation. The age of the

bottom of the core was estimated to be approximately ~ 7.9 cal ka BP (Fig. 3). However, the

extremely low temporal resolution between ~ 7.9 cal ka BP and ~ 4 cal ka BP precluded

making any general conclusion about that interval. Therefore, this study focuses only on the

last ~ 4 cal ka BP (the Neoglacial).

### 4.2 Sediment grainsize

The sediment was classified as medium to coarse silt throughout the core. The

sediment accumulation rate (SAR) prior to ~ 2.7 cal ka BP was 0.002 cm y$^{-1}$. The

approximately 10-fold increase in SAR was noted at ~ 2.7 cal ka BP, where it increased to

0.023 cm y$^{-1}$. During the last 1.5 cal ka BP, SAR decreased to 0.01 cm y$^{-1}$ (Fig. 4). The

amount of IRD was the highest prior to ~ 2.7 cal ka BP, reaching up to 83 grains g$^{-1}$. After ~

2.7 cal ka BP, the amount of IRD was relatively stable and did not exceed 18 grains g$^{-1}$. The

IRD flux decreased slightly over time to 0.37 grains g$^{-1}$ cm$^{-1}$, except for one peak reaching 0.8

grains g$^{-1}$ cm$^{-1}$ at ~ 2.6 cal ka BP (Fig. 4).

The mean grain size of the 0-63-µm fraction had its highest value (5.8 φ) at ~ 2.7 cal

27  ka BP (Fig. 4). After ~ 2.4 cal ka BP a slight but continuous reduction in the mean 0-63-µm

grain size was noted. The minimum grain size (6.23 φ) was recorded at the top of the core

(Fig. 4).

### 4.3 Stable isotopes

32  The two δ$^{18}$O data points values prior to ~ 2.7 cal ka BP changed slightly

33  betweenrecorded values of 3.55‰ and 3.69‰ vs. VPDB. Between ~ 2.7 and ~ 1.5 cal ka BP,

34  δ$^{18}$O showed the strongest variation, with values ranging from 3.28‰ to 3.77‰ vs. VPDB.

After ~ 1.5 cal ka BP, $\delta^{18}$O became slightly lighter (3.43‰ - 3.64‰ vs. VPDB), except for one peak noted in the uppermost layer of the core, where $\delta^{18}$O reached 3.87‰ vs. VPDB (Fig. 4).

In the period prior to ~ 2.7 cal ka BP, recorded values of $\delta^{13}$C displayed relatively light values ranging fromreached 0.92‰ to and 1.12‰ vs. VPDB. Slightly heavier $\delta^{13}$C (up to 1.46‰ vs. VPDB) was observed between ~ 2.7 and ~ 1.5 cal ka BP. The gradual decrease was recorded from ~ 1.5 cal ka BP to the present, reaching 0.81‰ vs. VPDB at the top of the core (Fig. 4).

## 4.4 Benthic foraminifera assemblages

A total of 8647 fossil foraminifera specimens belonging to 47 species were identified (Supplement 1; Supplementary Fig. 1). The number of foraminifera individuals in a sample varied from 156 to 2610 ind. $g^{-1}$, and the lowest abundances were observed prior to ~ 2.7 cal ka BP (Fig. 4). A short-term decrease in foraminifera abundance was observed between 2.2 and 1.7 cal ka BP, with values reaching as low as 304 ind. $g^{-1}$. The abundance maxima were noted at 2.3, 1.5, and 0.6 ka BP, with values reaching 2524, 2584, and 2610 ind. $g^{-1}$, respectively. The foraminiferal flux was low and relatively stable throughout the core, with values that did not exceed 1 ind $cm^{-2}$ $y^{-1}$, except for two peaks at 2.3 and 1.5 ka BP, when the flux reached 2.2 ind $cm^{-2}$ $y^{-1}$ for both peaks (Fig. 4).

The most abundant species was *Cassidulina reniforme*, with densities reaching up to 900 ind $g^{-1}$. The other species that constituted the majority of the foraminiferal assemblage were *Bucella frigida*, *Cibicidoides lobatulus*, *Elphidium excavatum*, *Islandiella* spp, *Melonis barleeanum*, and *Nonionellina labradorica*. The abundances of the dominant species followed a general trend, with maxima at ~ 2.3 cal ka BP and after ~ 1.7 cal ka BP and minima prior to ~ 2.7 cal ka BP and between 2.3 and 1.7 cal ka BP. (Fig. 5).

The foraminiferal assemblage prior to ~ 2.7 cal ka BP was dominated by indicators of AW inflow and/or frontal zones and glaciomarine taxa (Fig. 5). The most abundant species were *Nonionellina labradorica* and *Melonis barleeanum*, as well as *Cassidulina reniforme* and *Elphidium excavatum*, which together accounted for up to 60% of the foraminiferal abundance (Fig. 5). After ~ 2.7 cal ka BP, there were AW/frontal zone indicator peaks recorded at 2.4 and 1.8 cal ka BP, where the percentages increased to 33% and 28% of the total abundance, respectively. The period between ~ 2.4 cal ka BP and ~ 1.8 cal ka BP was characterized by an increase in the percentage of sea-ice indicators (*B. frigida* and *Islandiella* spp), which accounted for up to 25% of the total foraminiferal abundance. Additionally, a

short-term peak in the glaciomarine taxa, reaching up to 49% of the foraminiferal assemblage, was recorded between 2.5 and 2.1 cal ka BP. Also, the overall maximum of glaciomarine species abundance was recorded between 1.7 and 0.5 cal ka BP, ranging from 25% to 43% of foraminiferal assemblage (Fig. 5). A decrease in the relative abundance of glaciomarine species was observed after ~ 0.5 cal ka BP and was followed by an increase in the AW/frontal zone indicators and a single peak in the percentage of bottom current indicators, which reached 42% and 19%, respectively (Fig. 5).

**4.5 Foraminiferal aDNA sequences**

A total of 1,499,889 foraminiferal DNA sequences were clustered into 263 OTUs, and 20 remained unassigned. The remaining OTUs were assigned to Globigerinida (5 OTUs), Robertinida (1 OTU), Rotaliida (49 OTUs), Textulariida (18 OTUs), Monothalamea (163 OTUs), and Miliolida (7 OTUs). The majority of sequences belonged to Monothalamea (60%) and Rotaliida (31%) (Supplement 2; Supplementary Fig. 2). Herein, we focus on Monothalamea, which is the dominant component of the foraminiferal aDNA record.

The most important components of the monothalamous assemblage were *Micrometula* sp., *Cylindrogullmia* sp., *Hippocrepinella hirudinea*, *Ovammina* sp., *Nemogullmia* sp., *Tinogullmia* sp., *Cedhagenia saltatus*, undetermined allogromiids belonging to clades A and Y (herein called "allogromiids"), and sequences belonging to taxa known exclusively from environmental sequencing (herein called "environmental clades"). Herein, the term "allogromiid" refers to monothalamous foraminifera with organic or predominantly organic test walls (Gooday, 2002). Morphological and molecular evidence indicate that 'allogromiids' are not a coherent taxonomic group but are scattered between several monothalamous clades (Pawlowski et al., 2002). "Clade" refers to phylogenetic clades defined by molecular data. The clade is traditionally defined as a group of organisms that includes a common ancestor and all the descendants. The sequences belonging to allogromiids were present throughout the core, accounting for 16–31.7% of all the foraminiferal sequences. The exceptions were the intervals from ~ 4.0 to 2.4 cal ka BP, and ~ 1.7 cal ka BP, when the contribution of allogromiid sequences decreased to less than 10% (Fig. 6). The majority of the allogromiids belonged to clade Y, which made up to 100% of the allogromiid sequences. Only at 1.6–1.7 cal ka BP and 2.4–2.6 cal ka BP, most of allogromiid sequences belonged to clade A. Additionally, allogromiids belonging to clade I were noted at ~ 2.4 cal ka BP, where they made up 0.88% of allogromiid sequences (Fig. 7).

The periods prior to ~ 2.4 cal ka BP and ~ 1.7 cal ka BP were marked by the disappearance of sequences belonging to *C. saltatus*, *Nemogullmia* sp., and the environmental clades, followed by an increase in the percentages of sequences belonging to *Micrometula* sp., *Ovammina* sp., *Tinogullmia* sp., *Shepheardella* sp. and *Cylindrogullmia* sp. (Fig. 6).

**4.6 Diatom aDNA sequences**

A total of 824,697 diatom DNA sequences were clustered into 221 OTUs (Supplement 3; Supplementary Figure 3). The most abundantly sequenced diatom taxa were *Thalassiosira* spp, which made up 61.1% of diatom sequences. Other abundantly sequenced taxa were *Chaetoceros* sp. and *T. antarctica*, which made up 8.5% and 11.5% of sequences, respectively. The sequences of *Thalassiosira* sp were most abundant between ~ 2.2 cal ka BP and ~ 1.9 cal ka BP, accounting for up to 85% of all diatom sequences. The lowest percentage (14%) of *Thalassiosira* sp. was recorded at ~ 0.4 cal ka BP. Sequences assigned to *T. antarctica* were recorded throughout the core and their percentages were the highest at ~ 3.3 and ~ 2.6 cal ka BP, reaching up to 13% and 19%, respectively (Fig. 8). Sequences of *T. hispida* were also noted throughout the core and constituted 4.7% of diatom sequences in the uppermost layer. In the remaining samples, *T. hispida* sequences did not exceed 1%. The percentage of sequences of *Chaetoceros* sp. decreased downcore, from 76% at the surface to less than 1% at the bottom of the core (Fig. 8). *Navicula* sp. constituted an important part of the diatom assemblage at ~3.3 cal ka BP and ~1.9 cal ka BP, accounting for up to 25.5% and 10% of all diatom sequences, respectively. In the remaining samples, the abundance of *Navicula* sp. did not exceed 5% (Fig. 8).

**5. Discussion**

The ST_1.5 age model is based on the linear interpolation between the four AMS[14]C dates; thus, the age control of the core should be treated with caution. However , the timing of major environmental changes revealed by the ST_1.5 multiproxy record is in agreement with other records from the region (e.g., Sarnthein et al., 2003; Risebrobakken et al. 2010; Berben et al. 2017). Moreover, the major pulses of AW that were recorded ~ 2.3 and 1.7 cal ka BP correlated well with winter and summer SST maxima recorded in the 23258-2 core (Sarnthein et al., 2003).

**5.1 The period from 4 cal ka BP to 2.7 cal ka BP**

Prior to ~ 2.7 cal ka BP, the ST_1.5 sedimentary record displayed relatively higher

IRD delivery and a relatively lower 0-63-µm sediment fraction than in the following period

(Fig. 4). These results are in agreement with the record from Storfjordrenna (Łącka et al.,

2015), where peaks in IRD were noted during the Neoglacial and were attributed to increased

iceberg rafting due to fluctuations in the glacial fronts (e.g. Forwick et al., 2010).The coarser

0-63 µm fraction may suggest the winnowing of fine grained sediment, however,

foraminiferal fauna showed no clear response to sediment removal.

The foraminiferal flux and abundance prior to 2.7 cal ka BP reached their lowest

values (Fig. 4). Previous studies reported a decrease in the concentration of benthic

foraminifera in Storfjorden at that time, which was attributed to the presence of extensive ice

cover (Rasmussen and Thomsen, 2015; Knies et al. 2017). The dominant components of  the

ST_1.5 foraminiferal assemblage were *C. reniforme*. and  *M. barleeanum* (Fig. 5). The

presence of *C. reniforme* and *M. barleeanum* is associated with cooled and salty AW (e.g.,

Hald and Steinsund, 1996; Jernas et al., 2013). Moreover, these species are also associated

with the presence of phytodetritus, which may be related to the delivery of fresh organic

matter observed in frontal zones and/or near the sea-ice edge (Jennings et al., 2004). The

presence of sea-ice may be indicated also by the relatively light foraminiferal $\delta^{13}$C (Fig. 4), as

well as the highest percentage of the sea-ice species *Thalassiosira antarctica* (cf Ikävalko,

2004; Fig. 8). However, the low sampling resolution during that period precluded us from

making a general conclusion, and the latter assumptions should be confirmed by further

studies.

**5.2 The period from 2.7 cal ka BP to 0.5 cal ka BP. Episodes of AW inflow at ~ 2.3 and**

**1.7 cal ka BP.**

After ~ 2.7 cal ka BP, the increase in SAR was followed by a decrease in the mean

grain size of the 0-63-µm fraction and in the IRD delivery (Fig. 4). The 10-fold increase in

SAR most likely resulted from the intensive supply of turbid meltwater from advancing

glaciers and the consequent intensive sedimentation. Moreover, the accumulation of fine

sediment may also be enhanced by the slowdown of the bottom currents, indicated by the

finer 0-63-µm sediment fraction (Fig. 4). On the other hand, a decrease in IRD delivery may

suggest that the central Storfjorden was not impacted by iceberg rafting at that time. In

contrast, Rasmussen and Thomsen (2015) suggested glacial advance, followed by intensive

ice rafting and meltwater delivery to Storfjorden at that time. According to Knies et al.

(2017), the inner Storfjorden was covered by densely packed sea ice between ~ 2.8 and 0.5 cal

ka BP. Therefore, the decreasing IRD in the ST_1.5 core may result from the presence of a sea-ice cover that reduced iceberg rafting while the majority of coarse-grained material settled in the proximity to the glacial fronts. Similar conclusions have been stated by Forwick and Vorren (2009) and Forwick et al. (2010), who assumed that the enhanced formation of sea ice along the West Spitsbergen coast trapped icebergs inside the Isfjorden system.

The foraminiferal fauna in central Storfjorden revealed more than a 10-fold increase in flux and abundance followed by short-term fluctuations after ~ 2.7 cal ka BP (Fig. 4). The latter may suggest favorable conditions for foraminiferal growth. The major peaks in the total foraminiferal abundance (Fig. 4) followed by the peaks in the percentage of AW foraminiferal indicators (Fig. 5) were noted ~ 2.3 cal ka BP and ~ 1.7 cal ka BP. These peaks were associated with the occurrence of sequences of *T. hispida* (Fig. 8), a diatom species characteristic of subpolar and temperate regions (Katsuki et al., 2009). The timing of the changes described above is in accordance with the findings of Sarntheim et al. (2003), who reported two intervals of the remarkably warmer sea surface on the western continental margin of the Barents Sea at ~ 2.2 and ~ 1.6 cal ka BP, which was attributed to short-term pulses of warm AW advection. Other records also indicated AW inflow to the western and northern Barents Sea as well as to the western Spitsbergen continental margin during mid-late Holocene (e.g., Risebrobakken et al., 2010; Müller et al., 2012; Groot et al., 2014; Berben et al., 2014; 2017). During the mid- Holocene, AW inflow to the Barents Sea was relatively stable. The environmental conditions became more unstable in the late Holocene, with periodic cooling of surface waters, the presence of AW and/or chilled AW near the bottom, and more extensive seasonal sea ice cover (Risebrobakken et al., 2010; Berben et al., 2014; Groot et al. 2014). The timing of these changes differed between the study settings: in the western Barents Sea, it was ~ 1.1/1.5 cal ka BP (Berben et al. 2014; Groot et al., 2014), while in the southwestern Barents Sea, the change in environmental conditions was recorded ~ 2.5 cal ka BP (Risebrobakken et al., 2014). In contrast, the northern Barents Sea experienced surface water cooling and more extensive sea-ice cover prior to 2.7 cal ka BP. The increasing influence of AW was observed after 2.7 cal ka BP (Berben et al., 2017). Our foraminiferal and diatom aDNA records confirm the presence of AW intrusions in Storfjorden after ~ 2.7 cal ka BP, that may have caused an episodic breakup of sea-ice cover and permitted primary production and the development of benthic biota, including foraminifera.

The pulses of AW inflow at 2.3 cal ka BP and 1.7 cal ka BP were marked by the maxima of the foraminiferal flux (Fig. 4) and by peaks in the abundance of species associated with highly productive environments, such as *M. barleeanum* and *N. labradorica* (Fig. 5).

Moreover, the presence of diatom aDNA sequences throughout the core (Fig. 8) may suggest continuous primary production. However, the responses of the benthic foraminifera assemblage to the pulses of AW at ~ 2.3 cal ka BP and ~ 1.7 cal ka BP are slightly different. The dominant components of foraminiferal assemblage at ~ 2.3 cal ka BP were *M. barleeanum* and *E. excavatum*, while at ~ 1.7 cal ka BP, *N. labradorica* and *C. reniforme* were dominant (Fig. 5). The major difference in environmental conditions between these two "AW episodes" was noticeably coarser 0-63 µm sediment fraction noted at ~ 2.3 cal ka BP, what may indicate more intensive winnowing of fine sediment grains,, which would have created favorable conditions for the development of opportunistic species, such as *E. excavatum*. In contrast, the interval between 2.3 and 1.7 cal ka BP featured variable $\delta^{13}$C and $\delta^{18}$O followed by a decrease in the foraminiferal flux and abundance (Fig. 4). The foraminiferal assemblage at this time was dominated by glaciomarine and sea-ice taxa (Fig. 5), which indicate more severe environmental conditions with extensive ice cover and suppressed productivity.

The alternate cooling and warming periods described above were also reflected in the aDNA record of monothalamous foraminifera. During the periods with more severe environmental conditions (i.e., time intervals of 2.2–1.9 cal ka BP and 1.3–0.4 cal ka BP), the monothalamous foraminifera was dominated by allogromiids belonging to clade Y, *Nemogullmia* sp., *C. saltatus* and monothalamids belonging to so called "environmental clades" (Fig. 6). A considerable portion of the allogromiid sequences in the ST_1.5 core belong to clade Y (Fig. 7), which is primarily composed of taxa known only from environmental sequencing that have previously been noted in modern sediments in the Spitsbergen fjords (Pawłowska et al., *unpubl.*). Clade Y has also been abundantly sequenced in the coastal areas off Scotland, characterized by high levels of environmental disturbances (Pawlowski et al., 2014); this might suggest its high tolerance to environmental stress. *C. saltatus* was found by Gooday et al. (2011) in the Black Sea and its occurrence in areas with high levels of pollution suggests that it is an opportunistic species with a high tolerance for environmental disturbances. In addition, so called "environmental clades" are composed of monothalamous taxa known exclusively from environmental sequencing (Lecroq et al., 2011).. The abovementioned taxa nearly disappeared during the episodes of enhanced AW inflow at ~ 2.4 cal ka BP and ~ 1.7 cal ka BP, and the monothalamous assemblage was dominated at that time by *Micrometula* sp., *Ovammina* sp., *Shepheardella* sp., *Tinogullmia* sp., *Cylindrogullmia* sp., and allogromiids belonging to clade A (Fig. 6; Fig. 7). All these taxa have recently been observed in the fjords of Svalbard and Novaya Zemlya (e.g. Gooday et al.,

2005; Majewski et al., 2005; Sabbattini et al., 2007; Pawłowska et al., 2014; Korsun & Hald, 1998; Korsun et al., 1995). *Cylindrogullmia* and *Micrometula* are dependent on the presence of fresh phytodetritus (Alve, 2010). *Ovammina* sp. feeds on diatoms and other forms of microalgae (Goldstein & Alve, 2011). Similarly, the presence of *Tinogullmia* is largely controlled by the presence of organic material on the seafloor. High concentrations of *Tinogullmia* have been found in coastal (Cornelius & Gooday, 2004) and deep-sea regions (Gooday, 1993) within phytodetrital aggregates.

The taxa that dominated the monothalamous assemblage during warm intervals seem to be responsive to the delivery of organic matter and may flourish during phytoplankton blooms associated with the settling of organic matter (e.g., Alve, 2010; Sabbattini et al., 2012, 2013). The pulses of AW inflow may be associated with phytoplankton blooms stimulated by sea-ice melting and with the organic matter supply to the bottom (cf. Łącka et al., 2019). The continuous aDNA record of the sea-ice diatom *T. antarctica* (Fig. 8) suggests the presence of at least seasonal ice cover in the study area. On the other hand, the episodes of AW inflow were associated with the occurrence of the open-water taxa *T. hispida* (Fig. 8). The occurrence of sequences of both these taxa suggests the formation of ice cover during winter-spring, followed by ice-free summers. A similar scenario was proposed by Berben et al. (2017), who suggested increased AW to the eastern Svalbard and partial summer sea ice occurrence after 2.7 cal ka BP. According to the record of Łącka et al. (2019) from Storfjordrenna, the sea-ice melting induced the production of brines that may launch convective mixing and nutrient resupply from the bottom, which stimulated primary production.

Conversely, the colder phases of the Neoglacial were characterized by heavy and densely packed sea ice resulting in limited productivity (Knies et al., 2017). The presence of *T. anatrctica* sequences and the disappearance of *T. hispida* (Fig. 8) may suggest that primary production was associated with sea-ice. Furthermore, the monothalamous assemblage was less diverse and was dominated by more opportunistic taxa, which may indicate a reduced supply of organic matter to the bottom.

**5.3 The period after 0.5 cal ka BP.**

Modern-like conditions were established in Storfjorden at ~ 0.5 cal ka BP (Knies et al., 2017). The ST_1.5 record displayed a decrease in SAR compared to the preceding period, a decreasing 0-63 µm fraction and low IRD delivery (Fig. 4), which may indicate reduced glacial impact. Moreover, the peak of heavy $\delta^{18}O$ recorded on the core top (Fig. 4) may

suggest the presence of AW or slightly increased salinity. Similarly, Berben et al. (2014) recorded $\delta^{18}O$ values that suggested a minor increase in salinity, while foraminiferal fauna showed slightly lower salinities in the western Barents Sea at that time. The latter is in accordance with records from the Fram Strait (e.g. Werner et al., 2013) and the western Spitsbergen shelf (Cabedo-Sanz and Belt, 2016), which suggest episodes of freshening of the surface water masses associated with alternating sea ice increases and ice-free conditions in the late Holocene. Additionally, the records of Rasmussen and Thomsen (2014) and Knies et al., (2017) from Storfjorden indicated seasonally variable sea-ice cover. Moreover, the majority of diatom aDNA sequences found in the ST_1.5 record after ~ 0.5 cal ka BP belonged to *Chaetoceros* sp. (Fig. 8), a taxa that is observed in surface waters and is almost entirely absent under sea ice (Różańska et al., 2008). High abundances of *Chaetoceros* are often associated with highly productive surface waters (Cremer, 1999). Rigual-Hernández et al. (2017) also noted increased abundance of *Chaetoceros* sp. and enhanced algal productivity in Storfjorden after 2.0 cal ka BP, what was associated to the vicinity of the Arctic Front. However, the aDNA record of the monothalamous foraminifera at ~ 0.4 cal ka BP displayed relatively high percentages of taxa that dominated during the colder intervals of the Neoglacial (Fig. 6). This may be related to the recovery from the Little Ice Age, and consequently, from the temporarily deteriorated environmental conditions (D'Andrea et al., 2012). However, due to the low resolution during the LIA, a detailed interpretation is not possible. Therefore, further studies are required to confirm the latter conclusion.

**5.4 Paleoceanographic implications**

Our record revealed a two-phase Neoglacial, with a major shift in environmental conditions at ~ 2.7 cal ka BP. According to the ST_1.5 proxy records, the Neoglacial in Storfjorden was not a constantly cold period, but comprised alternating short-term cooling and warming periods, associated with variability in sea-ice coverage and productivity. The Neoglacial cooling was documented in various proxy reconstructions from the Nordic Seas (e.g., Jennings et al., 2002; Moros et al., 2004; Consolaro et al., 2018). However, there is growing evidence of shifts in environmental conditions in the Nordic Seas region in the Neoglacial, whose timings are in accordance with our record.. Alkenone record from the Norwegian Sea revealed a significant drop in sea surface temperature at 2.7 cal ka BP (Calvo et al., 2002). Risebrobakken et al. (2010) recorded a change in oceanographic conditions in the SW Barents Sea ca. 2.5 cal ka BP. The episodes of reduced surface and subsurface salinity

were recorded after 2.5 cal ka BP, what was attributed to the expansion of coastal waters and the occurrence of more sea-ice (Risebrobakken et al., 2010). Berben et al. (2017) recorded a shift ~2.7 cal ka BP, from the marginal ice zone to Arctic frontal conditions in the eastern Barents Sea. They observed continuous cooling trend from ~ 5.9 cal ka BP to 2.7 cal ka BP, with increased seasonal sea ice with less open water conditions, lower temperatures and decreased AW influence. Whereas, after 2.7 cal ka BP, the influence of AW was variable, but generally generally increasing. The period was characterized by low insolation, associated with surface cooling and enhanced formation of sea ice/reduced sea ice melt (Berben et al., 2017).

Moreover, our evidence of the presence of AW in Storfjorden during the Neoglacial supported previous suggestions that AW inflow during the late Holocene was strong enough to reach also the eastern coasts of Svalbard (e.g., Łącka et al., 2015). Episodic increases of the AW during the late Holocene were also observed in the northern Barents Sea (Duplessy et al., 2001; Lubinski et al., 2001), the eastern Barents Sea (Berben et al., 2014) and the Svalbard margin (Jernas et al., 2013; Werner et al., 2013). Sarnthein et al. (2003) postulated pulses of AW inflow to the western Barents Sea shelf at 2.2 and 1.6 cal ka BP. According to Perner et al. (2015), the Neoglacial delivery of chilled AW to the Nordic Seas culminated between 2.3 and 1.4 cal ka BP. These results are in accordance with the timing of major AW inflows revealed by our record.

## 6. Conclusions

The ST_1.5 multiproxy record revealed that the environmental variability in Storfjorden during the Neoglacial was controlled primarily by the interplay between AW and ArW and sea-ice cover variability. The molecular record supports and complements sedimentary and microfossil records, which indicate that major changes in the environmental conditions in Storfjorden occurred at ~ 2.7 cal ka BP. The general cooling in the early phase of the Neoglacial initiated conditions for the formation of extensive sea-ice cover. The latter part of the Neoglacial (after ~ 2.7 cal ka BP) was characterized by alternating short-term cooling and warming periods. Warming was associated with pulsed inflows of AW and sea-ice melting, which may stimulate phytoplankton blooms and organic matter supply to the bottom. The cold phases were characterized by heavy and densely packed sea ice resulting in limited productivity.

Moreover, the aDNA diatom record supports the conclusion that primary production took place continuously during the Neoglacial, regardless of the sea-ice conditions. The early

phase of the Neoglacial was characterized by the presence of diatom taxa associated with sea ice, whereas the present-day diatom assemblage was dominated by *Chaetoceros* spp, a taxa characteristic of open water.

The aDNA record of monothalamous foraminifera is in agreement with the microfossil record and revealed the timing of the major pulses of AW at 2.3 and 1.7 cal ka BP. The AW inflow was marked by an increase in the percentage of sequences of monothalamous taxa associated with the presence of fresh phytodetritus. The monothalamous assemblage during cold intervals was less diverse and was dominated by monothalamous foraminifera known only from environmental sequencing.

**Author contributions**

MZ and Jan P designed the study. Joanna P, MŁ and MZ collected the sediment core. MŁ and MK performed the sedimentological and micropaleontological analyses. Joanna P performed the molecular analyses and prepared the manuscript with contributions from all co-authors.

**Acknowledgements**

The study was supported by the National Science Centre grants no. 2015/19/D/ST10/00244 and 2016/21/B/ST10/02308, and Swiss National Science Foundation grant no. 31003A_179125. The Authors would like to thank anonymous Reviewers for constructive comments that helped to improve the manuscript.

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

isotopes in benthic foraminiferal tests, and the flux and abundance of foraminifera are

presented.

**Figure 5:** The absolute abundance (expressed as the number of individuals per gram of dry

sediment) and the percentage of the dominant benthic foraminifera.

**Figure 6:** The dominant components of the monothalamous assemblages. The abundance is

expressed as the percentage of the monothalamous sequences and the most abundantly

sequenced taxa are presented. The trend (2-point average) is indicated with a dashed line.

**Figure 7:** The percentage share of certain clades in the allogromiid sequences.

**Figure 8:** The percentage of sequences of dominant diatom taxa vs. time. The trend (2-point

average) is indicated with the dashed line.

**Table captions**

**Table 1:** Raw and calibrated AMS[14]C dates used in the age model.

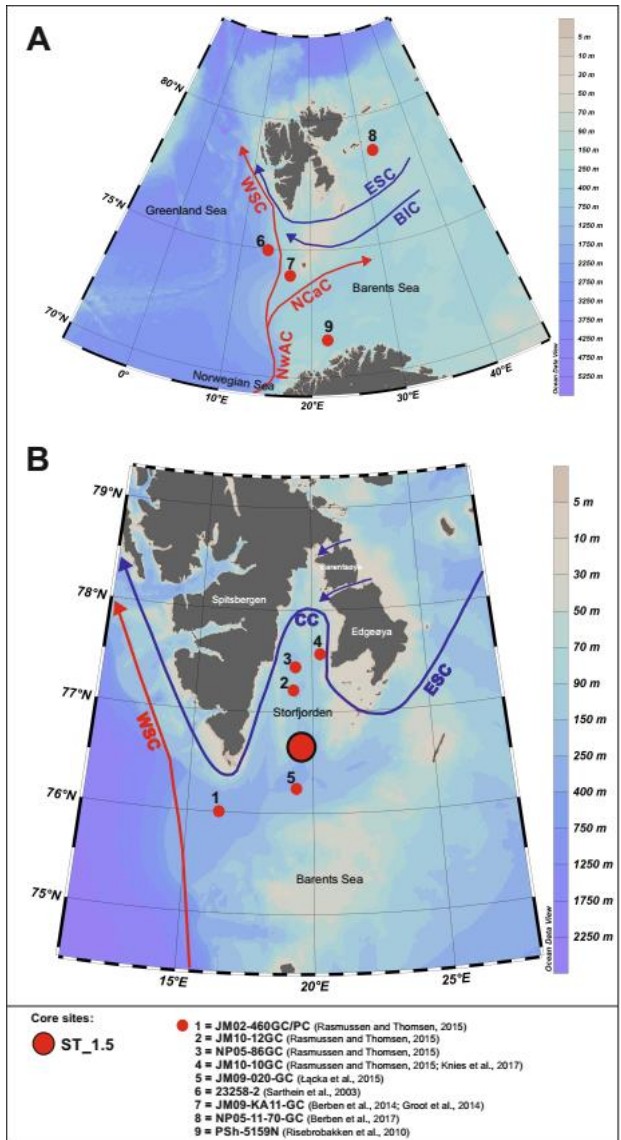

**Figure 1:** The modern oceanography of the study area (A) and the location of the studied core ST_1.5 (B) and
the other cores discussed in this paper (A,B). Abbreviations of main surface currents: WSC – West Spitsbergen
Current, NwAC – ~~North~~ Norwegian Atlantic Current, NCaC – North Cape Current, ESC – East Spitsbergen
Current, BIC – Bear Island Current, CC – Coastal Current.

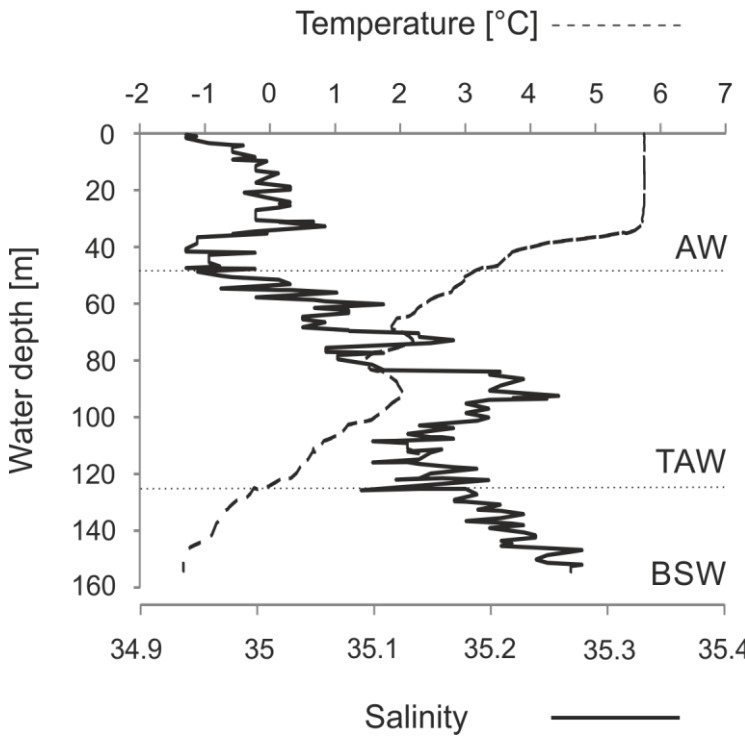

2  **Figure 2:** Temperature and salinity profile from the sampling station. Temperature is marked with a dashed line,
3  and salinity is marked with a black line. Abbreviations: AW – Atlantic Water, TAW – Transformed Atlantic
4  Water, BSW – Brine-enriched Shelf Water.
5

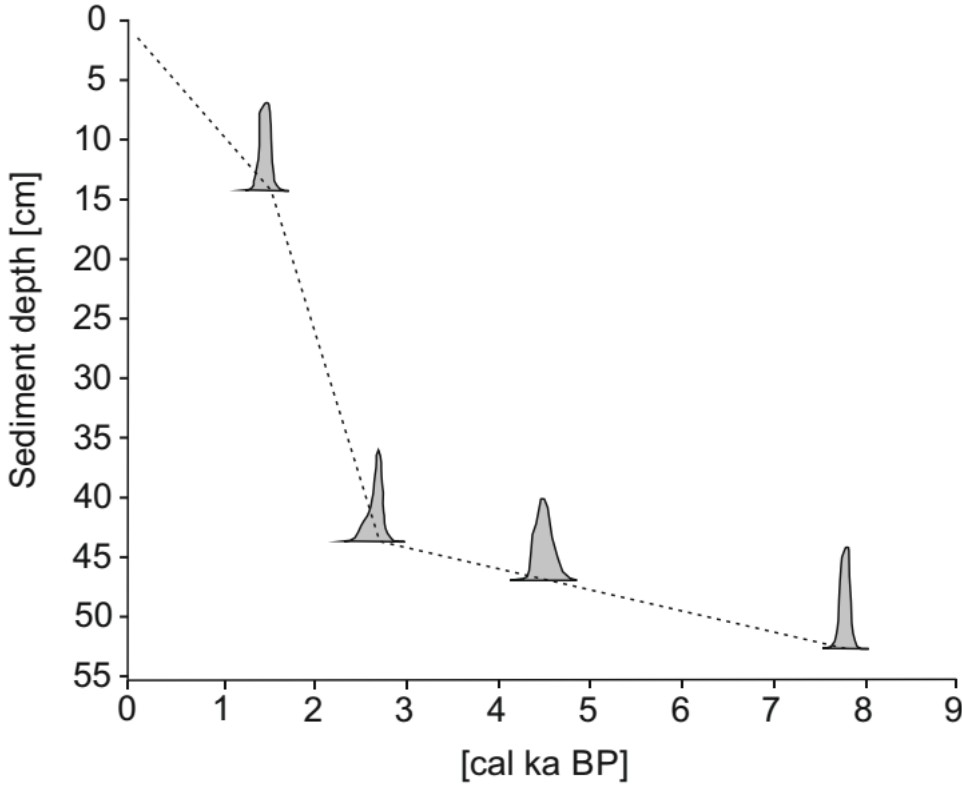

8  **Figure 3:** Age–depth model of the ST_1.5 core. The grey silhouettes show probability distribution of calendar
9  dates that were obtained by calibration of individual [14]C dates used for the age model. The dotted line shows the
10 age-depth model derived from a linear interpolation between the dates.

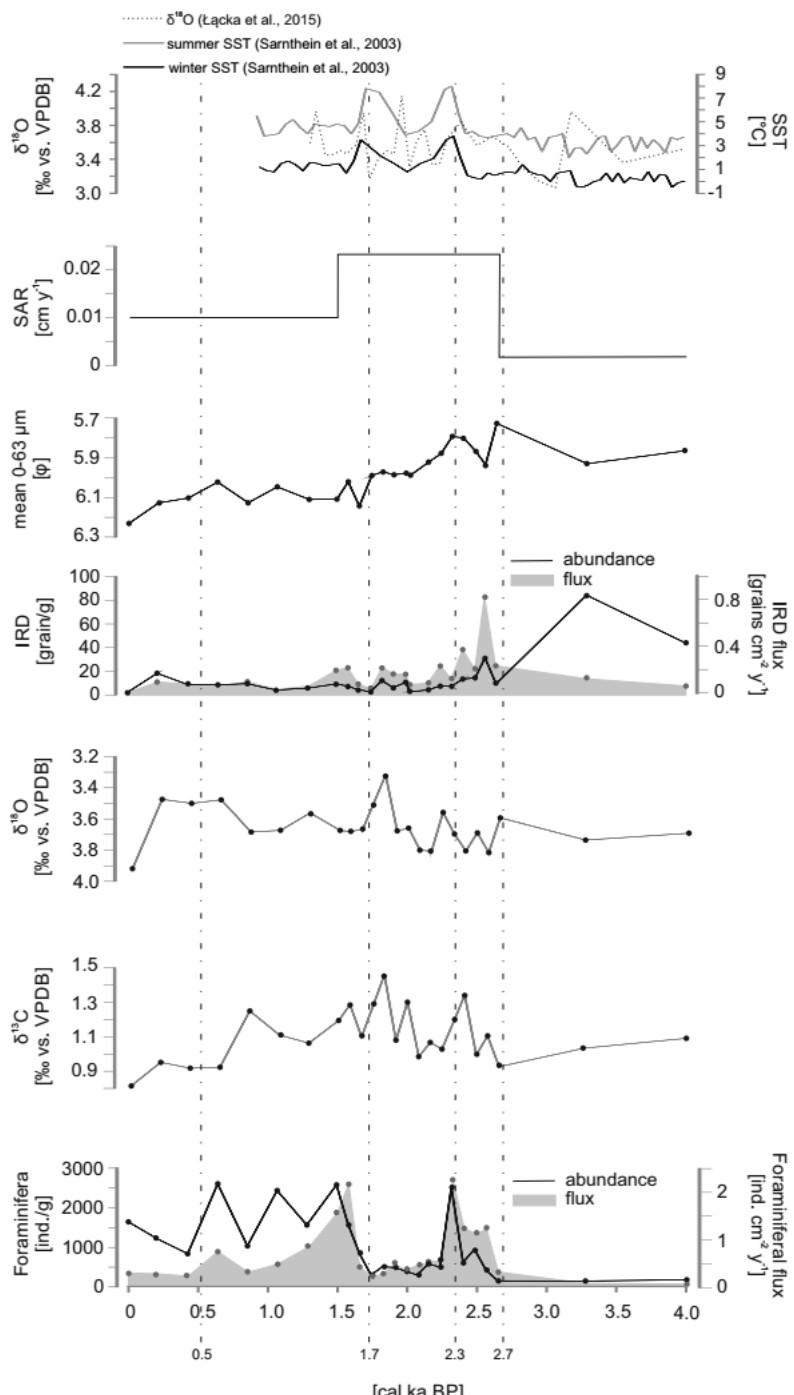

**Figure 4:** Sedimentological and micropaleontological data plotted versus age. The sediment accumulation rate
(SAR), mean grain size of the 0-63-μm fraction, ice-rafted debris (IRD) flux and number of grains per gram of
sediment, oxygen ($\delta^{18}$O) and carbon ($\delta^{13}$C) stable isotopes in benthic foraminiferal tests, and the flux and
abundance of foraminifera are presented.

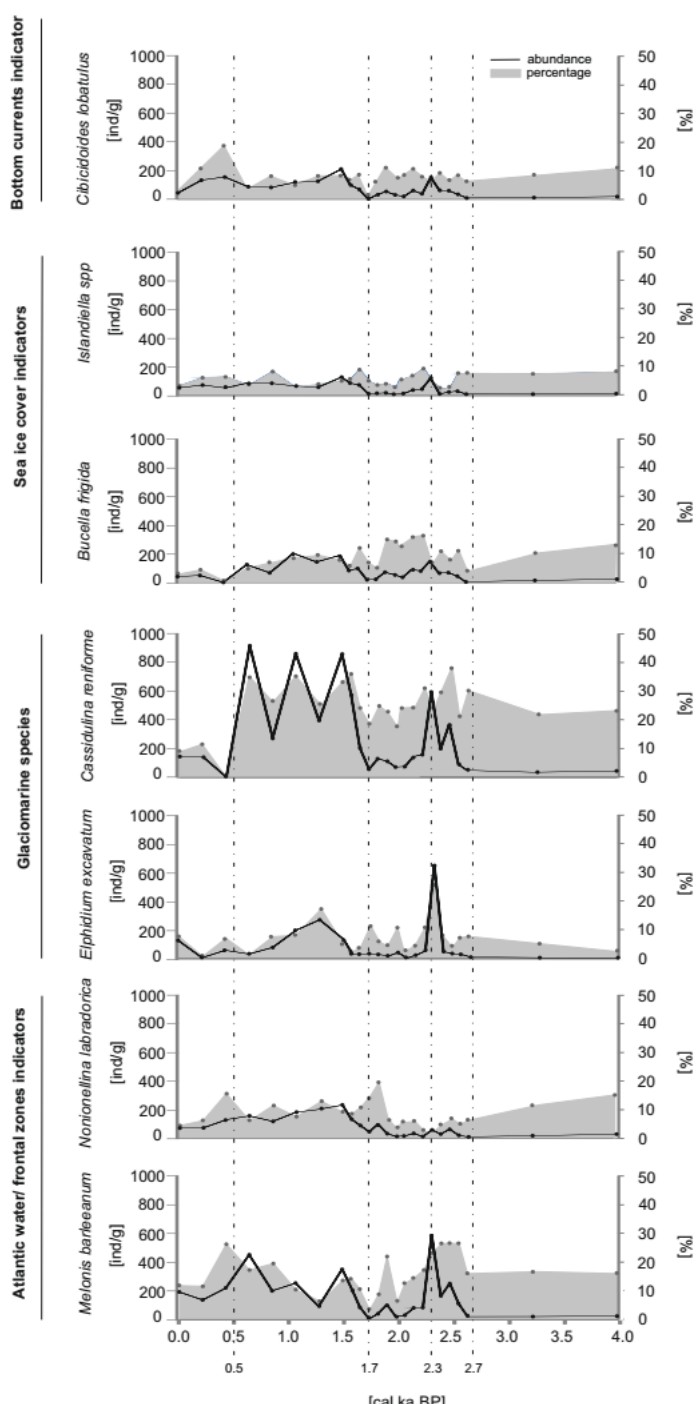

**Figure 5:** The absolute abundance (expressed as the number of individuals per gram of dry sediment) and the
percentage of the dominant benthic foraminifera.

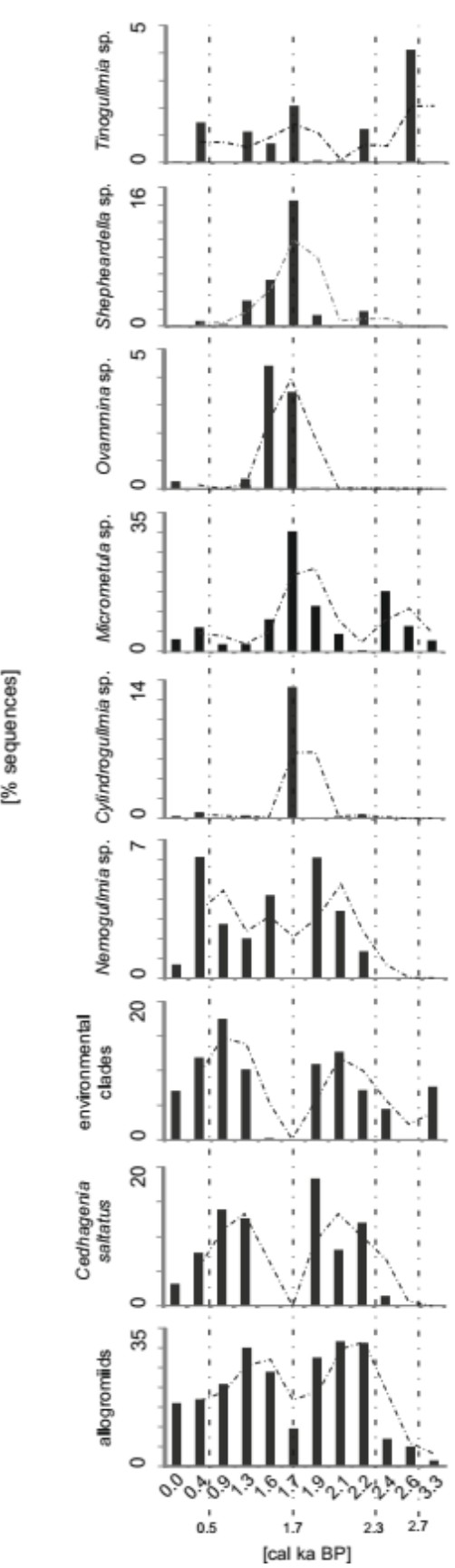

**Figure 6:** The dominant components of the monothalamous assemblages. The abundance is expressed as the percentage of the monothalamous sequences and the most abundantly sequenced taxa are presented. The trend (2-point average) is indicated with a dashed line.

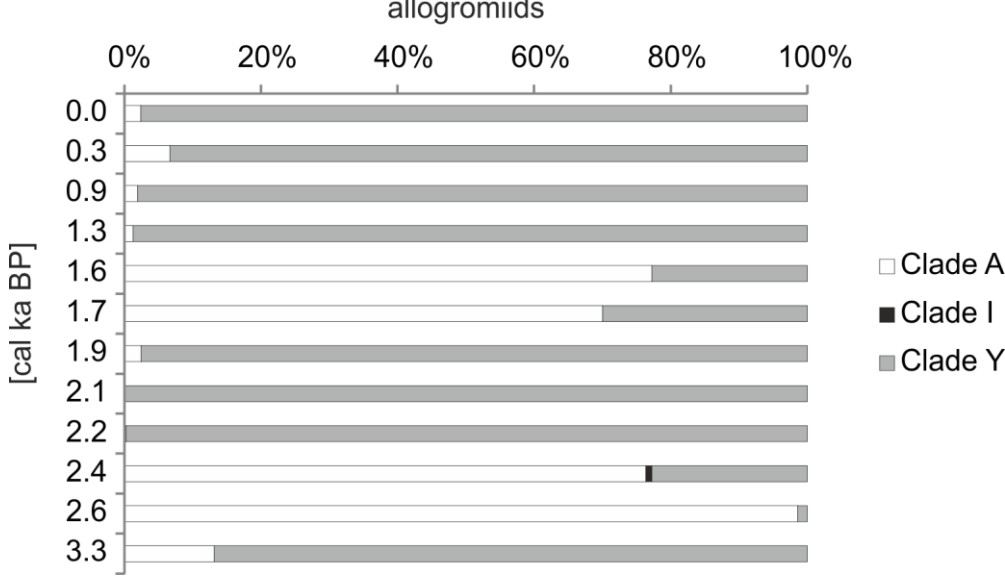

2    **Figure 7:** The percentage share of certain clades in the allogromiid sequences.

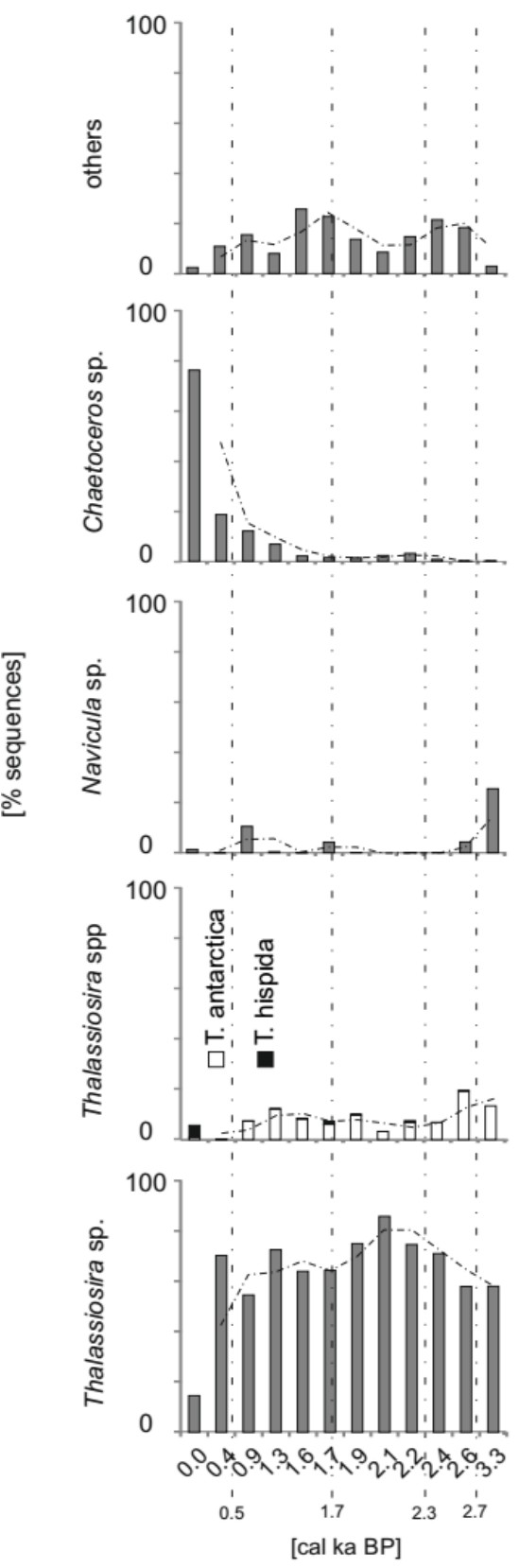

2
3 **Figure 8:** The percentage of sequences of dominant diatom taxa vs. time. The trend (2-point average) is
4 indicated with a dashed line.

**Table 1:** Raw and calibrated AMS[14]C dates used in the age model. B stands for bivlave shells, while F stands for benthic foraminifera tests.

| Core depth [cm] | Material | Raw AMS [14]C | Cal. a BP ± 2σ | Cal. a BP used in age model |
|---|---|---|---|---|
| 2.5 | *Nuculana pernula* (**B**) | 107.38 ± 0.33 pMC | - | - |
| 5.5 | *Yoldiella lenticula* (**B**) | 290 ± 30 BP | - | - |
| 14.5 | *Turitella erosa* (**B**) | 2020 ± 30 BP | 1356-1555 | 1500 |
| 43.5 | *Yoldiella solituda* (**B**) | 3010 ± 50 BP | 2484-2787 | 2700 |
| 46.5 | *Nonionellina labradorica* (**F**) | 4490± 40 BP | 4400-4701 | 4500 |
| 52.5 | *Yoldiella lenticula* (**B**) | 7545 ± 35 BP | 7803-7989 | 7890 |

# Multiproxy evidence of the Neoglacial expansion of Atlantic Water to eastern Svalbard

Joanna Pawłowska[1*], Magdalena Łącka[1], Małgorzata Kucharska[1], Jan Pawlowski[1,2], Marek Zajączkowski[1]

[1]Institute of Oceanology Polish Academy of Sciences, Sopot, 81-712, Poland

[2]Department of Genetics and Evolution, University of Geneva, Geneva, CH 1211, Switzerland

*Correspondence to*: Joanna Pawłowska (pawlowska@iopan.pl)

**Abstract.** The main goal of this study is to reconstruct the paleoceanographic development of Storfjorden during the Neoglacial (~ 4 cal ka BP). Storfjorden is one of the most important brine factories in the European Arctic and is responsible for deep water production. Moreover, it is a climate-sensitive area influenced by two contrasting water masses: warm and saline Atlantic Water (AW) and cold and fresh Arctic Water (ArW). Herein, a multiproxy approach was applied to provide evidence for existing interactions between the inflow of AW and sea-ice coverage, which are the major drivers of environmental changes in Storfjorden. The sedimentary and microfossil records indicate that a major reorganization of oceanographic conditions in Storfjorden occurred at ~ 2.7 cal ka BP. The cold conditions and the less-pronounced presence of AW in Storfjorden during the early phase of the Neoglacial were the prerequisite conditions for the formation of extensive sea-ice cover. The period after ~ 2.7 cal ka BP was characterized by alternating short-term cooling and warming intervals. Warming was associated with pulsed inflows of AW and sea-ice melting that stimulated phytoplankton blooms and organic matter supply to the bottom. The cold phases were characterized by heavy and densely packed sea ice resulting in decreased productivity. The ancient environmental DNA (aDNA) records of foraminifera and diatoms support the occurrence of the major pulses of AW (~2.3 and ~1.7 cal ka BP) and the variations in sea-ice cover. The episodes of enhanced AW inflow were marked by an increase in the percentage of DNA sequences of monothalamous foraminifera associated with the presence of fresh phytodetritus. Cold and less productive intervals were marked by an increased proportion of monothalamous taxa

known only from environmental sequencing. The diatom aDNA record indicates that primary production was continuous during the Neoglacial, regardless of the sea-ice conditions. However, the colder periods were characterized by the presence of diatom taxa associated with sea ice, whereas the present-day diatom assemblage is dominated by open-water taxa.

## 1. Introduction

The northward flow of Atlantic Water (AW) is one of the major contributors of heat to the Arctic Ocean (Polyakov et al., 2017). Recent oceanographic data indicate a warming trend due to an increased inflow of AW towards the Arctic Ocean (Rudels et al., 2015, Polyakov et al., 2017). AW has been present along the western margin of Svalbard for at least the last 12,000 years (e.g., Werner et al., 2011; Rasmussen et al., 2014). One of the major intrusions of AW occurred during the early Holocene (10.8 – 6.8 cal ka BP). A distinct cooling and freshening of the west Spitsbergen shelf bottom water masses occurred during the mid-late Holocene (6.8 – 1 cal ka BP) and was accompanied by glacier readvances in Svalbard, leading to the present-day conditions (Ślubowska-Woldengen et al., 2007; Telesiński et al., 2018). The paleoceanographic conditions in the Svalbard margins correlate closely to the sea surface temperature (SST) variations in the Nordic Seas and confirm that the Svalbard area is highly sensitive to fluctuations in the inflow of AW (Ślubowska-Woldengen et al., 2007; Werner et al., 2013). Conversely, until the 1990s eastern Svalbard was recognized as an area exclusively influenced by the East Spitsbergen Current (ESC), which carries cold, less saline Arctic Water (ArW) from the Barents Sea (e.g., Quadfasel et al., 1988; Piechura et al., 1996). However, recent studies have revealed that the oceanography of the area is much more complicated (e.g., Skogseth et al., 2007; Geyer et al., 2010). Oceanographic data obtained from conductivity–temperature sensors attached to *Delphinapterus leucas* show a substantial contribution of AW to Storfjorden, East Spitsbergen (Lydersen et al., 2002). Recently, Hansen et al. (2011) suggested the presence of AW in Storfjorden during the early Holocene warming (11 – 6.8 cal ka BP), which was further confirmed by the foraminiferal and sedimentary records of Łącka et al. (2015).

The latter part of the Holocene, the so-called Neoglacial cooling (~ 4 cal ka BP), in the European Arctic is characterized by a declined summer insolation at northern latitudes (Berger, 1978) that correlates to a decline in summer SST (e.g., Andersen et al., 2004; Risebrobakken et al., 2010; Rasmussen et al., 2014; Ivanova et al., 2019). The cooling of the surface waters and the limited AW inflow towards the Nordic Seas led to the formation of an extended sea-ice cover in West Spitsbergen margin (Müller et al., 2012). In addition, the

southwestern and eastern shelf of Spitsbergen experienced a strengthening of the East Spitsbergen Current and/or Jan Mayen Current leading to an intensification of ArW inflow and the formation of extensive sea-ice cover (e.g., Sarnthein et al., 2003; Berben et al., 2014). Therefore, the Neoglacial is usually considered a generally cold period (e.g., Consolaro et al., 2018). However, the records from Storfjorden and the Barents Sea suggest that the Neoglacial was a period of variable oceanographic conditions with strong temperature and salinity gradients (Martrat et al., 2003; Risebrobakken et al. 2010; Sarnthein et al., 2003; Łącka et al., 2015; 2019). In addition, there is evidence of episodic intensifications of the warm AW inflow towards western Svalbard at that time (e.g.; Rasmussen et al., 2012).

According to Nilsen et al. (2008), the critical parameter controlling the fjord–shelf exchange is the density difference between the fjord water masses and the AW. The local winter ice production and the formation of brine-enriched waters determine the density of local water masses, which is a key factor that enables AW to penetrate into fjords during the spring and summer. Moreover, the production of brine-enriched waters and the associated deep-water overflow are key contributors to large-scale ocean circulation (Killworth, 1983). In this respect, Storfjorden is especially important because it is one of the few areas where brine-enriched waters have been frequently observed (Haarpainter et al., 2001). In recent decades, reduced brine formation has occurred during the periods with the most intensive AW advection to Storfjorden and less sea-ice formation in the Barents Sea, while intense brine formation re-establishes during periods of recurrent cooling (Årthun et al., 2011;Rasmussen and Thomsen, 2014).

The aim of this study is to reconstruct the paleoceanographic development of Storfjorden during the Neoglacial at multicentennial resolution. We assumed that the periodic intensification of the AW inflow to the West Spitsbergen shelf during the Neoglacial resulted in the appearance of AW also in eastern Spitsbergen, similar to the conditions in the early Holocene (e.g., Łącka et al., 2015), affecting the density and extent of sea-ice cover in the area. A multiproxy approach comprising sedimentary, microfossil and molecular records was applied to provide evidence for the interactions between the inflow of AW and sea-ice coverage in Storfjorden. The ancient environmental DNA (aDNA) analysis targeted diatoms and nonfossilized monothalamous foraminifera. Both these of groups are hardly preserved in fossil records from the Svalbard fjords (Pawłowska et al., 2014) and shelf areas (Zimmermann et al., 2019 and references therein). Recent studies have demonstrated that analyses of genetic material obtained directly from environmental samples (so-called environmental DNA) are an efficient method for performing biodiversity surveys across time and space (Thomsen and

Willerslev, 2015). The content of environmental DNA samples may be analyzed by DNA metabarcoding, which consists of high-throughput sequencing of taxonomically informative DNA fragments called metabarcodes. The identification of short, species-specific DNA fragments (so called "barcodes") allows us to obtain species-level assignments of modern and ancient DNA sequences (Herbert et al., 2003). The further demonstration that DNA can be preserved in the environment across geological timescales opened new avenues for palaeoclimatic and palaeoceanographic studies. Recent studies have demonstrated the preservation of DNA in marine sediments for tens to hundreds thousands of years. An aDNA approach was successfully applied to trace the Holocene history of dinoflagellates, haptophytes (e.g., Coolen et al., 2009, 2013; Boere et al., 2009) and foraminifera in deep sea (Lejzerowicz et al., 2013) and coastal areas (Pawłowska et al., 2014; 2016). The study of Pawłowska et al. (2016) was the first attempt to utilize foraminiferal aDNA as a paleoenivronmental proxy. This study supported the existence of extremely diverse foraminiferal assemblages. The richness of the foraminiferal community revealed by the molecular record was much higher than that in the fossil record (Pawłowska et al., 2014), mainly due to the detection of nonfossilized monothalamous taxa. The molecular data correlated well with environmental changes and revealed even small changes that were not clearly indicated by other proxy records. The combination of aDNA studies with the analysis of microfossils and sedimentary proxies provides a powerful means to reconstruct past environments more comprehensively.

**2. Study area**

Storfjorden is located in southeastern Svalbard between the islands of Spitsbergen, Edgeøya and Barentsøya (Fig. 1). Storfjorden is ~190 km long and its main basin is ~190 m deep. Two narrow and shallow passages (Heleysundet and Freemansundet) connect northern Storfjorden to the Barents Sea. To the south, a 120-m-deep sill separates the main basin from the Storfjordrenna. Storfjordrenna is 245 m long, with a depth varying from 150 m to 420 m.

The water masses in Storfjorden are composed primarily of exogenous Atlantic and Arctic waters as well as mixed waters that have formed locally. The Atlantic Water is transported northwards by Norwegian Atlantic Current (NwAC), which branches off in the Barents Sea into the West Spitsbergen Current and North Cape Current (Loeng et al. 1991; Blindheim and Østerhus, 2005) (Fig. 1a). In addition to AW, cold and less saline Arctic Water (ArW) is transported to the Barents Sea via East Spitsbergen Current and Bear Island Current (Hopkins, 1991) (Fig. 1a). Arctic water (ArW) from the Arctic Ocean and the Barents Sea

enters Storfjorden via two passages to the northeast and continues along the inner shelf of Svalbard as a Coastal Current (Fig. 1b). AW is characterized by temperatures > 3 °C and salinity > 34.95, while the temperature and salinity of ArW are < 0 °C and 34.3-34.8, respectively. The presence of locally formed water masses is a result of the interactions between AW, ArW and melt water. Skogseth et al. (2005) listed six local water masses: melt water (MW), polar front water (PW), East Spitsbergen water (ESW), brine-enriched shelf water (BSW), Storfjorden surface water (SSW), and modified Atlantic water (MAW). BSW is formed due to the release of a large amounts of brines during polynya events and the intensive formation of sea ice (Haarpainter et al., 2001; Skogseth et al., 2004, 2005) and is characterized by salinities exceeding 34.8 and temperatures below -1.5 °C (Skogseth et al., 2005).

The sedimentary environment in Storfjorden is classified as a low-energy, high-accumulation environment, which is characteristic of inner fjords. The area is sheltered from along-shelf bottom currents and is affected by high terrigenous inputs; therefore deposition prevails over sediment removal by bottom currents (Winklemann and Knies, 2005). The primary productivity is high and strongly depends on the sea-ice formation as well as the seasonal duration of the marginal ice zone (Winkelman and Knies, 2005).

## 3. Materials and methods

### 3.1 Marine sediment core

The 55-cm-long sediment core ST_1.5 was taken with a gravity corer in Storfjorden retrieved with the R/V *Oceania* in August 2014. The sampling station was located at 76° 53,181' N and 19° 27,559' E at a depth of 153 m (Fig. 1). The salinity and temperature of the water column at the coring station was measured with a Mini CTD Sensordata SD 204 at intervals of 1 s (Fig. 2). The core was stored at 4°C and shipped to the Institute of Oceanology PAS for further analyses.

In the laboratory, the core was extruded and cut into 1-cm slices. During cutting, sterile subsamples for ancient DNA (aDNA) analyses were taken at 4 cm intervals. To avoid external and/or cross-contamination the thin layers of sediment that were in contact with under- or overlying sediments were removed using a sterile spatula. Samples for aDNA analyses were kept frozen at -20°C. Samples for other proxy analyses were taken every 2 cm.

### 3.2 Chronology

The chronology of the marine sediment core is based on high-precision accelerator mass spectrometry (AMS) [14]C dating performed on five bivalve shells retrieved from the sediment layers at 2.5, 5.5, 14.5, 43.5, and 52.5 cm core depth and on the foraminifera *Nonionellina labradorica* from the 46.5 cm core depth. The bivalve shells were identified to the highest possible taxonomic level and processed on the 1.5 SDH-Pelletron Model "Compact Carbon AMS" in the Poznań Radiocarbon Laboratory, Poznań, Poland. Dating of foraminiferal tests was performed at the National Ocean Sciences AMS (NOSAMS) laboratory in the Woods Hole Oceanographic Institution, Woods Hole, MA, USA. The dates were converted into calibrated ages using the calibration program CALIB Rev. 7.1.0 Beta (Stuiver and Reimer, 1993) and the Marine13 calibration dataset (Reimer et al., 2013). A reservoir age correction ($\Delta R$) of $105 \pm 24$ was applied (Mangerud et al., 2006). The calibrated results are reported in units of thousand calibrated years BP (cal ka BP) (Table 1).

**3.3 Grain size analysis**

The samples for grain size analyses were freeze-dried and milled. The measurements were performed using a Mastersizer 2000 particle laser analyzer coupled to a Hydro MU device (Malvern, UK). The samples were treated with ultrasound to avoid aggregation. The raw data were analyzed using GRADISTAT v.8.0 software (Blott and Pye, 2001). The mean 0-63-µm grain size [$\varphi$] was calculated via the logarithmic method of moments. The sediment fraction >500 µm was used to reconstruct an ice rafted debris (IRD) record. The grains were counted under a stereomicroscope and the amount of IRD is reported as the concentration (i.e., the number of grains per gram of dry sediment) [grains $g^{-1}$] and the flux [grains $cm^{-2}$ $y^{-1}$].

**3.4 Benthic foraminifera assemblages**

Prior to the analysis of testate benthic foraminifera, samples were wet sieved through a meshes with 500-µm and 100-µm openings and dried at 60°C. Samples with large quantities of tests were divided using a microsplitter. At least 300 specimens of benthic foraminifera were isolated from each sample and collected on micropaleontological slides. Benthic foraminifera specimens were counted and identified to the lowest possible taxonomic level. The quantity of foraminifera is presented as the concentration (i.e., the number of individuals per gram of dry sediment) [ind. $g^{-1}$] and the flux [ind. $cm^{-2}$ $y^{-1}$]. Foraminifera species were grouped according to their ecological tolerances. Four groups of indicators were distinguished: AW/frontal zone indicators, ArW indicators, bottom current indicators and

glaciomarine species (Majewski et al., 2009). The morphologically similar species *Islandiella norcrossi* and *Islandiella helenae* are reported as *Islandiella* spp.

**3.5 Stable isotope analysis**

Carbon and oxygen stable isotope analyses were performed on *Cibicidoides lobatulus* tests selected from 27 sediment layers. Ca. 10 to 12 specimens were collected from each sample and subjected to ultrasonic cleaning. The measurements were performed on a Finnigan MAT 253 mass spectrometer coupled to a Kiel IV carbonate preparation device at the University of Florida. The resulting values are expressed in standard δ notation relative to Vienna Pee Dee Belemnite (VPDB).

**3.6 Ancient DNA analysis**

The total DNA was extracted from approximately 10 g of sediment using a Power Max Soil DNA extraction kit (MoBio). The foraminiferal SSU rDNA fragments containing the 37f hypervariable region were PCR amplified using primers tagged with unique sequences of five nucleotides appended to their 5' ends (denoted by Xs), namely, the foraminifera-specific forward primer s14F1 (5'-XXXXXCGGACACACTGAGGATTGACAG-3') and the reverse primer s15 (5'-XXXXXCCTATCACATAATCATGAAAG-3'). The diatom DNA fragment located in the V4 region was amplified with the forward DIV4for (5'-XXXXXXXXGCGGTAATTCCAGCTCCAATAG-3') and reverse DIV4rev3 (5′-XXXXXXXXCTCTGACAATGGAATACGAATA-3') primers tagged with a unique combination of eight nucleotides (denoted by Xs) attached at each primer's 5'-end. The amplicons were purified using the High Pure PCR Cleanup Micro Kit (Roche) and quantified using a Qubit 2.0 fluorometer. Samples were pooled in equimolar quantities, and the sequence library was prepared using a TruSeq library-preparation kit (Illumina). The samples were then loaded into a MiSeq instrument for a paired-end run of 2*150 cycles (foraminifera) and 2*250 cycles (diatoms). The processing of the HTS sequence data was performed according to procedures described by Lejzerowicz et al. (2013) and Pawłowska et al. (2014). The post-sequencing data processing was performed with the use of the SLIM web app (Dufresne et al., 2019) and included demultiplexing the libraries, joining the paired-end reads, chimera removal, operational taxonomic units (OTUs) clustering, and taxonomic assignment. Sequences were clustered into OTUs using the Swarm module (Mahe et al. 2014), and each OTU was assigned to the highest possible taxonomic level using vsearch (Rognes et al., 2016) against a local database and then reassigned using BLAST (Altschul et al., 1990). The results

1 are presented in OTU-to-sample tables and transformed in terms of the number of sequences,

2 number of OTUs and percentage (%) of sequences.

4 **4. Results**

**4.1 Chronology**

In total, six radiocarbon dates were obtained, all of which were recorded in

chronological order. The uppermost layer contained modern, post-bomb carbon indicating a

post-1960 age (Table 1). Samples from the 2.5 cm and 5.5 cm core depths were not calibrated

because they revealed ages invalid for the selected calibration curve. The age model was,

therefore, based on the four remaining dates using a linear interpolation. The age of the

bottom of the core was estimated to be approximately ~ 7.9 cal ka BP (Fig. 3). However, the

extremely low temporal resolution between ~ 7.9 cal ka BP and ~ 4 cal ka BP precluded

making any general conclusion about that interval. Therefore, this study focuses only on the

last ~ 4 cal ka BP (the Neoglacial).

**4.2 Sediment grainsize**

The sediment was classified as medium to coarse silt throughout the core. The

sediment accumulation rate (SAR) prior to ~ 2.7 cal ka BP was 0.002 cm $y^{-1}$. The

approximately 10-fold increase in SAR was noted at ~ 2.7 cal ka BP, where it increased to

0.023 cm $y^{-1}$. During the last 1.5 cal ka BP, SAR decreased to 0.01 cm $y^{-1}$ (Fig. 4). The

amount of IRD was the highest prior to ~ 2.7 cal ka BP, reaching up to 83 grains $g^{-1}$. After ~

2.7 cal ka BP, the amount of IRD was relatively stable and did not exceed 18 grains $g^{-1}$. The

IRD flux decreased slightly over time to 0.37 grains $g^{-1}$ $cm^{-1}$, except for one peak reaching 0.8

grains $g^{-1}$ $cm^{-1}$ at ~ 2.6 cal ka BP (Fig. 4).

The mean grain size of the 0-63-µm fraction had its highest value (5.8 φ) at ~ 2.7 cal

27 ka BP (Fig. 4). After ~ 2.4 cal ka BP a slight but continuous reduction in the mean 0-63-µm

grain size was noted. The minimum grain size (6.23 φ) was recorded at the top of the core

(Fig. 4).

**4.3 Stable isotopes**

The two $\delta^{18}$O data points prior to ~ 2.7 cal ka BP recorded values of 3.55‰ and

3.69‰ vs. VPDB. Between ~ 2.7 and ~ 1.5 cal ka BP, $\delta^{18}$O showed the strongest variation,

with values ranging from 3.28‰ to 3.77‰ vs. VPDB. After ~ 1.5 cal ka BP, $\delta^{18}$O became

slightly lighter (3.43‰ - 3.64‰ vs. VPDB), except for one peak noted in the uppermost layer of the core, where $\delta^{18}O$ reached 3.87‰ vs. VPDB (Fig. 4).

In the period prior to ~ 2.7 cal ka BP, recorded values of $\delta^{13}C$ reached 0.92‰ and 1.12‰ vs. VPDB. Slightly heavier $\delta^{13}C$ (up to 1.46‰ vs. VPDB) was observed between ~ 2.7 and ~ 1.5 cal ka BP. The gradual decrease was recorded from ~ 1.5 cal ka BP to the present, reaching 0.81‰ vs. VPDB at the top of the core (Fig. 4).

**4.4 Benthic foraminifera assemblages**

A total of 8647 fossil foraminifera specimens belonging to 47 species were identified (Supplement 1; Supplementary Fig. 1). The number of foraminifera individuals in a sample varied from 156 to 2610 ind. $g^{-1}$, and the lowest abundances were observed prior to ~ 2.7 cal ka BP (Fig. 4). A short-term decrease in foraminifera abundance was observed between 2.2 and 1.7 cal ka BP, with values reaching as low as 304 ind. $g^{-1}$. The abundance maxima were noted at 2.3, 1.5, and 0.6 ka BP, with values reaching 2524, 2584, and 2610 ind. $g^{-1}$, respectively. The foraminiferal flux was low and relatively stable throughout the core, with values that did not exceed 1 ind $cm^{-2}$ $y^{-1}$, except for two peaks at 2.3 and 1.5 ka BP, when the flux reached 2.2 ind $cm^{-2}$ $y^{-1}$ for both peaks (Fig. 4).

The most abundant species was *Cassidulina reniforme*, with densities reaching up to 900 ind $g^{-1}$. The other species that constituted the majority of the foraminiferal assemblage were *Bucella frigida*, *Cibicidoides lobatulus*, *Elphidium excavatum*, *Islandiella* spp, *Melonis barleeanum*, and *Nonionellina labradorica*. The abundances of the dominant species followed a general trend, with maxima at ~ 2.3 cal ka BP and after ~ 1.7 cal ka BP and minima prior to ~ 2.7 cal ka BP and between 2.3 and 1.7 cal ka BP. (Fig. 5).

The foraminiferal assemblage prior to ~ 2.7 cal ka BP was dominated by indicators of AW inflow and/or frontal zones and glaciomarine taxa (Fig. 5). The most abundant species were *Nonionellina labradorica* and *Melonis barleeanum*, as well as *Cassidulina reniforme* and *Elphidium excavatum*, which together accounted for up to 60% of the foraminiferal abundance (Fig. 5). After ~ 2.7 cal ka BP, there were AW/frontal zone indicator peaks recorded at 2.4 and 1.8 cal ka BP, where the percentages increased to 33% and 28% of the total abundance, respectively. The period between ~ 2.4 cal ka BP and ~ 1.8 cal ka BP was characterized by an increase in the percentage of sea-ice indicators (*B. frigida* and *Islandiella* spp), which accounted for up to 25% of the total foraminiferal abundance. Additionally, a short-term peak in the glaciomarine taxa, reaching up to 49% of the foraminiferal assemblage, was recorded between 2.5 and 2.1 cal ka BP. Also, the overall maximum of glaciomarine

species abundance was recorded between 1.7 and 0.5 cal ka BP, ranging from 25% to 43% of foraminiferal assemblage (Fig. 5). A decrease in the relative abundance of glaciomarine species was observed after ~ 0.5 cal ka BP and was followed by an increase in the AW/frontal zone indicators and a single peak in the percentage of bottom current indicators, which reached 42% and 19%, respectively (Fig. 5).

**4.5 Foraminiferal aDNA sequences**

A total of 1,499,889 foraminiferal DNA sequences were clustered into 263 OTUs, and 20 remained unassigned. The remaining OTUs were assigned to Globigerinida (5 OTUs), Robertinida (1 OTU), Rotaliida (49 OTUs), Textulariida (18 OTUs), Monothalamea (163 OTUs), and Miliolida (7 OTUs). The majority of sequences belonged to Monothalamea (60%) and Rotaliida (31%) (Supplement 2; Supplementary Fig. 2). Herein, we focus on Monothalamea, which is the dominant component of the foraminiferal aDNA record.

The most important components of the monothalamous assemblage were *Micrometula* sp., *Cylindrogullmia* sp., *Hippocrepinella hirudinea*, *Ovammina* sp., *Nemogullmia* sp., *Tinogullmia* sp., *Cedhagenia saltatus*, undetermined allogromiids belonging to clades A and Y (herein called "allogromiids"), and sequences belonging to taxa known exclusively from environmental sequencing (herein called "environmental clades"). Herein, the term "allogromiid" refers to monothalamous foraminifera with organic or predominantly organic test walls (Gooday, 2002). Morphological and molecular evidence indicate that 'allogromiids' are not a coherent taxonomic group but are scattered between several monothalamous clades (Pawlowski et al., 2002). "Clade" refers to phylogenetic clades defined by molecular data. The clade is traditionally defined as a group of organisms that includes a common ancestor and all the descendants. The sequences belonging to allogromiids were present throughout the core, accounting for 16–31.7% of all the foraminiferal sequences. The exceptions were the intervals from ~ 4.0 to 2.4 cal ka BP, and ~ 1.7 cal ka BP, when the contribution of allogromiid sequences decreased to less than 10% (Fig. 6). The majority of the allogromiids belonged to clade Y, which made up to 100% of the allogromiid sequences. Only at 1.6–1.7 cal ka BP and 2.4–2.6 cal ka BP, most of allogromiid sequences belonged to clade A. Additionally, allogromiids belonging to clade I were noted at ~ 2.4 cal ka BP, where they made up 0.88% of allogromiid sequences (Fig. 7).

The periods prior to ~ 2.4 cal ka BP and ~ 1.7 cal ka BP were marked by the disappearance of sequences belonging to *C. saltatus*, *Nemogullmia* sp., and the environmental

clades, followed by an increase in the percentages of sequences belonging to *Micrometula* sp.,

*Ovammina* sp., *Tinogullmia* sp., *Shepheardella* sp. and *Cylindrogullmia* sp. (Fig. 6).

**4.6 Diatom aDNA sequences**

A total of 824,697 diatom DNA sequences were clustered into 221 OTUs (Supplement 3; Supplementary Figure 3). The most abundantly sequenced diatom taxa were *Thalassiosira* spp, which made up 61.1% of diatom sequences. Other abundantly sequenced taxa were *Chaetoceros* sp. and *T. antarctica*, which made up 8.5% and 11.5% of sequences, respectively. The sequences of *Thalassiosira* sp were most abundant between ~ 2.2 cal ka BP and ~ 1.9 cal ka BP, accounting for up to 85% of all diatom sequences. The lowest percentage (14%) of *Thalassiosira* sp. was recorded at ~ 0.4 cal ka BP. Sequences assigned to *T. antarctica* were recorded throughout the core and their percentages were the highest at ~ 3.3 and ~ 2.6 cal ka BP, reaching up to 13% and 19%, respectively (Fig. 8). Sequences of *T. hispida* were also noted throughout the core and constituted 4.7% of diatom sequences in the uppermost layer. In the remaining samples, *T. hispida* sequences did not exceed 1%. The percentage of sequences of *Chaetoceros* sp. decreased downcore, from 76% at the surface to less than 1% at the bottom of the core (Fig. 8). *Navicula* sp. constituted an important part of the diatom assemblage at ~3.3 cal ka BP and ~1.9 cal ka BP, accounting for up to 25.5% and 10% of all diatom sequences, respectively. In the remaining samples, the abundance of *Navicula* sp. did not exceed 5% (Fig. 8).

**5. Discussion**

The ST_1.5 age model is based on the linear interpolation between the four AMS$^{14}$C dates; thus, the age control of the core should be treated with caution. However , the timing of major environmental changes revealed by the ST_1.5 multiproxy record is in agreement with other records from the region (e.g., Sarnthein et al., 2003; Risebrobakken et al. 2010; Berben et al. 2017). Moreover, the major pulses of AW that were recorded ~ 2.3 and 1.7 cal ka BP correlated well with winter and summer SST maxima recorded in the 23258-2 core (Sarnthein et al., 2003).

**5.1 The period from 4 cal ka BP to 2.7 cal ka BP**

Prior to ~ 2.7 cal ka BP, the ST_1.5 sedimentary record displayed relatively higher IRD delivery and a relatively lower 0-63-µm sediment fraction than in the following period (Fig. 4). These results are in agreement with the record from Storfjordrenna (Łącka et al.,

2015), where peaks in IRD were noted during the Neoglacial and were attributed to increased iceberg rafting due to fluctuations in the glacial fronts (e.g. Forwick et al., 2010).The coarser 0-63 µm fraction may suggest the winnowing of fine grained sediment, however, foraminiferal fauna showed no clear response to sediment removal.

The foraminiferal flux and abundance prior to 2.7 cal ka BP reached their lowest values (Fig. 4). Previous studies reported a decrease in the concentration of benthic foraminifera in Storfjorden at that time, which was attributed to the presence of extensive ice cover (Rasmussen and Thomsen, 2015; Knies et al. 2017). The dominant components of the ST_1.5 foraminiferal assemblage were *C. reniforme*. and *M. barleeanum* (Fig. 5). The presence of *C. reniforme* and *M. barleeanum* is associated with cooled and salty AW (e.g., Hald and Steinsund, 1996; Jernas et al., 2013). Moreover, these species are also associated with the presence of phytodetritus, which may be related to the delivery of fresh organic matter observed in frontal zones and/or near the sea-ice edge (Jennings et al., 2004). The presence of sea-ice may be indicated also by the relatively light foraminiferal $\delta^{13}$C (Fig. 4), as well as the highest percentage of the sea-ice species *Thalassiosira antarctica* (cf Ikävalko, 2004; Fig. 8). However, the low sampling resolution during that period precluded us from making a general conclusion, and the latter assumptions should be confirmed by further studies.

**5.2 The period from 2.7 cal ka BP to 0.5 cal ka BP. Episodes of AW inflow at ~ 2.3 and 1.7 cal ka BP.**

After ~ 2.7 cal ka BP, the increase in SAR was followed by a decrease in the mean grain size of the 0-63-µm fraction and in the IRD delivery (Fig. 4). The 10-fold increase in SAR most likely resulted from the intensive supply of turbid meltwater from advancing glaciers and the consequent intensive sedimentation. Moreover, the accumulation of fine sediment may also be enhanced by the slowdown of the bottom currents, indicated by the finer 0-63-µm sediment fraction (Fig. 4). On the other hand, a decrease in IRD delivery may suggest that the central Storfjorden was not impacted by iceberg rafting at that time. In contrast, Rasmussen and Thomsen (2015) suggested glacial advance, followed by intensive ice rafting and meltwater delivery to Storfjorden at that time. According to Knies et al. (2017), the inner Storfjorden was covered by densely packed sea ice between ~ 2.8 and 0.5 cal ka BP. Therefore, the decreasing IRD in the ST_1.5 core may result from the presence of a sea-ice cover that reduced iceberg rafting while the majority of coarse-grained material settled in the proximity to the glacial fronts. Similar conclusions have been stated by Forwick and

Vorren (2009) and Forwick et al. (2010), who assumed that the enhanced formation of sea ice along the West Spitsbergen coast trapped icebergs inside the Isfjorden system.

The foraminiferal fauna in central Storfjorden revealed more than a 10-fold increase in flux and abundance followed by short-term fluctuations after ~ 2.7 cal ka BP (Fig. 4). The latter may suggest favorable conditions for foraminiferal growth. The major peaks in the total foraminiferal abundance (Fig. 4) followed by the peaks in the percentage of AW foraminiferal indicators (Fig. 5) were noted ~ 2.3 cal ka BP and ~ 1.7 cal ka BP. These peaks were associated with the occurrence of sequences of *T. hispida* (Fig. 8), a diatom species characteristic of subpolar and temperate regions (Katsuki et al., 2009). The timing of the changes described above is in accordance with the findings of Sarntheim et al. (2003), who reported two intervals of the remarkably warmer sea surface on the western continental margin of the Barents Sea at ~ 2.2 and ~ 1.6 cal ka BP, which was attributed to short-term pulses of warm AW advection. Other records also indicated AW inflow to the western and northern Barents Sea as well as to the western Spitsbergen continental margin during mid-late Holocene (e.g., Risebrobakken et al., 2010; Müller et al., 2012; Groot et al., 2014; Berben et al., 2014; 2017). During the mid- Holocene, AW inflow to the Barents Sea was relatively stable. The environmental conditions became more unstable in the late Holocene, with periodic cooling of surface waters, the presence of AW and/or chilled AW near the bottom, and more extensive seasonal sea ice cover (Risebrobakken et al., 2010; Berben et al., 2014; Groot et al. 2014). The timing of these changes differed between the study settings: in the western Barents Sea, it was ~ 1.1/1.5 cal ka BP (Berben et al. 2014; Groot et al., 2014), while in the southwestern Barents Sea, the change in environmental conditions was recorded ~ 2.5 cal ka BP (Risebrobakken et al., 2014). In contrast, the northern Barents Sea experienced surface water cooling and more extensive sea-ice cover prior to 2.7 cal ka BP. The increasing influence of AW was observed after 2.7 cal ka BP (Berben et al., 2017). Our foraminiferal and diatom aDNA records confirm the presence of AW intrusions in Storfjorden after ~ 2.7 cal ka BP, that may have caused an episodic breakup of sea-ice cover and permitted primary production and the development of benthic biota, including foraminifera.

The pulses of AW inflow at 2.3 cal ka BP and 1.7 cal ka BP were marked by the maxima of the foraminiferal flux (Fig. 4) and by peaks in the abundance of species associated with highly productive environments, such as *M. barleeanum* and *N. labradorica* (Fig. 5). Moreover, the presence of diatom aDNA sequences throughout the core (Fig. 8) may suggest continuous primary production. However, the responses of the benthic foraminifera assemblage to the pulses of AW at ~ 2.3 cal ka BP and ~ 1.7 cal ka BP are slightly different.

The dominant components of foraminiferal assemblage at ~ 2.3 cal ka BP were *M. barleeanum* and *E. excavatum*, while at ~ 1.7 cal ka BP, *N. labradorica* and *C. reniforme* were dominant (Fig. 5). The major difference in environmental conditions between these two "AW episodes" was noticeably coarser 0-63 μm sediment fraction noted at ~ 2.3 cal ka BP, what may indicate more intensive winnowing of fine sediment grains,, which would have created favorable conditions for the development of opportunistic species, such as *E. excavatum*. In contrast, the interval between 2.3 and 1.7 cal ka BP featured variable $\delta^{13}$C and $\delta^{18}$O followed by a decrease in the foraminiferal flux and abundance (Fig. 4). The foraminiferal assemblage at this time was dominated by glaciomarine and sea-ice taxa (Fig. 5), which indicate more severe environmental conditions with extensive ice cover and suppressed productivity.

The alternate cooling and warming periods described above were also reflected in the aDNA record of monothalamous foraminifera. During the periods with more severe environmental conditions (i.e., time intervals of 2.2–1.9 cal ka BP and 1.3–0.4 cal ka BP), the monothalamous foraminifera was dominated by allogromiids belonging to clade Y, *Nemogullmia* sp., *C. saltatus* and monothalamids belonging to so called "environmental clades" (Fig. 6). A considerable portion of the allogromiid sequences in the ST_1.5 core belong to clade Y (Fig. 7), which is primarily composed of taxa known only from environmental sequencing that have previously been noted in modern sediments in the Spitsbergen fjords (Pawłowska et al., *unpubl.*). Clade Y has also been abundantly sequenced in the coastal areas off Scotland, characterized by high levels of environmental disturbances (Pawlowski et al., 2014); this might suggest its high tolerance to environmental stress. *C. saltatus* was found by Gooday et al. (2011) in the Black Sea and its occurrence in areas with high levels of pollution suggests that it is an opportunistic species with a high tolerance for environmental disturbances. In addition, so called "environmental clades" are composed of monothalamous taxa known exclusively from environmental sequencing (Lecroq et al., 2011).. The abovementioned taxa nearly disappeared during the episodes of enhanced AW inflow at ~ 2.4 cal ka BP and ~ 1.7 cal ka BP, and the monothalamous assemblage was dominated at that time by *Micrometula* sp., *Ovammina* sp., *Shepheardella* sp., *Tinogullmia* sp., *Cylindrogullmia* sp., and allogromiids belonging to clade A (Fig. 6; Fig. 7). All these taxa have recently been observed in the fjords of Svalbard and Novaya Zemlya (e.g. Gooday et al., 2005; Majewski et al., 2005; Sabbattini et al., 2007; Pawłowska et al., 2014; Korsun & Hald, 1998; Korsun et al., 1995). *Cylindrogullmia* and *Micrometula* are dependent on the presence of fresh phytodetritus (Alve, 2010). *Ovammina* sp. feeds on diatoms and other forms of

microalgae (Goldstein & Alve, 2011). Similarly, the presence of *Tinogullmia* is largely controlled by the presence of organic material on the seafloor. High concentrations of *Tinogullmia* have been found in coastal (Cornelius & Gooday, 2004) and deep-sea regions (Gooday, 1993) within phytodetrital aggregates.

The taxa that dominated the monothalamous assemblage during warm intervals seem to be responsive to the delivery of organic matter and may flourish during phytoplankton blooms associated with the settling of organic matter (e.g., Alve, 2010; Sabbattini et al., 2012, 2013). The pulses of AW inflow may be associated with phytoplankton blooms stimulated by sea-ice melting and with the organic matter supply to the bottom (cf. Łącka et al., 2019). The continuous aDNA record of the sea-ice diatom *T. antarctica* (Fig. 8) suggests the presence of at least seasonal ice cover in the study area. On the other hand, the episodes of AW inflow were associated with the occurrence of the open-water taxa *T. hispida* (Fig. 8). The occurrence of sequences of both these taxa suggests the formation of ice cover during winter-spring, followed by ice-free summers. A similar scenario was proposed by Berben et al. (2017), who suggested increased AW to the eastern Svalbard and partial summer sea ice occurrence after 2.7 cal ka BP. According to the record of Łącka et al. (2019) from Storfjordrenna, the sea-ice melting induced the production of brines that may launch convective mixing and nutrient resupply from the bottom, which stimulated primary production.

Conversely, the colder phases of the Neoglacial were characterized by heavy and densely packed sea ice resulting in limited productivity (Knies et al., 2017). The presence of *T. anatrctica* sequences and the disappearance of *T. hispida* (Fig. 8) may suggest that primary production was associated with sea-ice. Furthermore, the monothalamous assemblage was less diverse and was dominated by more opportunistic taxa, which may indicate a reduced supply of organic matter to the bottom.

**5.3 The period after 0.5 cal ka BP.**

Modern-like conditions were established in Storfjorden at ~ 0.5 cal ka BP (Knies et al., 2017). The ST_1.5 record displayed a decrease in SAR compared to the preceding period, a decreasing 0-63 μm fraction and low IRD delivery (Fig. 4), which may indicate reduced glacial impact. Moreover, the peak of heavy $\delta^{18}$O recorded on the core top (Fig. 4) may suggest the presence of AW or slightly increased salinity. Similarly, Berben et al. (2014) recorded $\delta^{18}$O values that suggested a minor increase in salinity, while foraminiferal fauna showed slightly lower salinities in the western Barents Sea at that time. The latter is in

accordance with records from the Fram Strait (e.g. Werner et al., 2013) and the western Spitsbergen shelf (Cabedo-Sanz and Belt, 2016), which suggest episodes of freshening of the surface water masses associated with alternating sea ice increases and ice-free conditions in the late Holocene. Additionally, the records of Rasmussen and Thomsen (2014) and Knies et al., (2017) from Storfjorden indicated seasonally variable sea-ice cover. Moreover, the majority of diatom aDNA sequences found in the ST_1.5 record after ~ 0.5 cal ka BP belonged to *Chaetoceros* sp. (Fig. 8), a taxa that is observed in surface waters and is almost entirely absent under sea ice (Różańska et al., 2008). High abundances of *Chaetoceros* are often associated with highly productive surface waters (Cremer, 1999). Rigual-Hernández et al. (2017) also noted increased abundance of *Chaetoceros* sp. and enhanced algal productivity in Storfjorden after 2.0 cal ka BP, what was associated to the vicinity of the Arctic Front. However, the aDNA record of the monothalamous foraminifera at ~ 0.4 cal ka BP displayed relatively high percentages of taxa that dominated during the colder intervals of the Neoglacial (Fig. 6). This may be related to the recovery from the Little Ice Age, and consequently, from the temporarily deteriorated environmental conditions (D'Andrea et al., 2012). However, due to the low resolution during the LIA, a detailed interpretation is not possible. Therefore, further studies are required to confirm the latter conclusion.

**5.4 Paleoceanographic implications**

Our record revealed a two-phase Neoglacial, with a major shift in environmental conditions at ~ 2.7 cal ka BP. According to the ST_1.5 proxy records, the Neoglacial in Storfjorden was not a constantly cold period, but comprised alternating short-term cooling and warming periods, associated with variability in sea-ice coverage and productivity. The Neoglacial cooling was documented in various proxy reconstructions from the Nordic Seas (e.g., Jennings et al., 2002; Moros et al., 2004; Consolaro et al., 2018). However, there is growing evidence of shifts in environmental conditions in the Nordic Seas region in the Neoglacial, whose timings are in accordance with our record.. Alkenone record from the Norwegian Sea revealed a significant drop in sea surface temperature at 2.7 cal ka BP (Calvo et al., 2002). Risebrobakken et al. (2010) recorded a change in oceanographic conditions in the SW Barents Sea ca. 2.5 cal ka BP. The episodes of reduced surface and subsurface salinity were recorded after 2.5 cal ka BP, what was attributed to the expansion of coastal waters and the occurrence of more sea-ice (Risebrobakken et al., 2010). Berben et al. (2017) recorded a shift ~2.7 cal ka BP, from the marginal ice zone to Arctic frontal conditions in the eastern

Barents Sea. They observed continuous cooling trend from ~ 5.9 cal ka BP to 2.7 cal ka BP, with increased seasonal sea ice with less open water conditions, lower temperatures and decreased AW influence. Whereas, after 2.7 cal ka BP, the influence of AW was variable, but generally generally increasing. The period was characterized by low insolation, associated with surface cooling and enhanced formation of sea ice/reduced sea ice melt (Berben et al., 2017).

Moreover, our evidence of the presence of AW in Storfjorden during the Neoglacial supported previous suggestions that AW inflow during the late Holocene was strong enough to reach also the eastern coasts of Svalbard (e.g., Łącka et al., 2015). Episodic increases of the AW during the late Holocene were also observed in the northern Barents Sea (Duplessy et al., 2001; Lubinski et al., 2001), the eastern Barents Sea (Berben et al., 2014) and the Svalbard margin (Jernas et al., 2013; Werner et al., 2013). Sarnthein et al. (2003) postulated pulses of AW inflow to the western Barents Sea shelf at 2.2 and 1.6 cal ka BP. According to Perner et al. (2015), the Neoglacial delivery of chilled AW to the Nordic Seas culminated between 2.3 and 1.4 cal ka BP. These results are in accordance with the timing of major AW inflows revealed by our record.

## 6. Conclusions

The ST_1.5 multiproxy record revealed that the environmental variability in Storfjorden during the Neoglacial was controlled primarily by the interplay between AW and ArW and sea-ice cover variability. The molecular record supports and complements sedimentary and microfossil records, which indicate that major changes in the environmental conditions in Storfjorden occurred at ~ 2.7 cal ka BP. The general cooling in the early phase of the Neoglacial initiated conditions for the formation of extensive sea-ice cover. The latter part of the Neoglacial (after ~ 2.7 cal ka BP) was characterized by alternating short-term cooling and warming periods. Warming was associated with pulsed inflows of AW and sea-ice melting, which may stimulate phytoplankton blooms and organic matter supply to the bottom. The cold phases were characterized by heavy and densely packed sea ice resulting in limited productivity.

Moreover, the aDNA diatom record supports the conclusion that primary production took place continuously during the Neoglacial, regardless of the sea-ice conditions. The early phase of the Neoglacial was characterized by the presence of diatom taxa associated with sea ice, whereas the present-day diatom assemblage was dominated by *Chaetoceros* spp, a taxa characteristic of open water.

The aDNA record of monothalamous foraminifera is in agreement with the microfossil record and revealed the timing of the major pulses of AW at 2.3 and 1.7 cal ka BP. The AW inflow was marked by an increase in the percentage of sequences of monothalamous taxa associated with the presence of fresh phytodetritus. The monothalamous assemblage during cold intervals was less diverse and was dominated by monothalamous foraminifera known only from environmental sequencing.

**Author contributions**

MZ and Jan P designed the study. Joanna P, MŁ and MZ collected the sediment core. MŁ and MK performed the sedimentological and micropaleontological analyses. Joanna P performed the molecular analyses and prepared the manuscript with contributions from all co-authors.

**Acknowledgements**

The study was supported by the National Science Centre grants no. 2015/19/D/ST10/00244 and 2016/21/B/ST10/02308, and Swiss National Science Foundation grant no. 31003A_179125. The Authors would like to thank anonymous Reviewers for constructive comments that helped to improve the manuscript.

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

**Figures captions**

**Figure 1:** The modern oceanography of the study area (A) and the location of the studied core

ST_1.5 (B) and the other cores discussed in this paper (A,B). Abbreviations of the main

surface currents: WSC – West Spitsbergen Current, NwAC – Norwegian Atlantic Current,

NCaC – North Cape Current, ESC – East Spitsbergen Current, BIC – Bear Island Current, CC

– Coastal Current.

**Figure 2:** Temperature and salinity profile from the core location. Temperature is marked

with a dashed line, and salinity is marked with a black line. Abbreviations: AW – Atlantic

Water, TAW – Transformed Atlantic Water, BSW – Brine-enriched Shelf Water.

**Figure 3:** Age–depth model of the ST_1.5 core. The gray silhouettes show the probability

distribution of the calendar dates that were obtained by the calibration of the individual [14]C

dates used for the age model. The dotted line shows the age-depth model derived from linear

interpolation between the dates.

**Figure 4:** Sedimentological and micropaleontological data plotted versus age. The sediment

accumulation rate (SAR), mean grain size of the 0-63-µm fraction, ice-rafted debris (IRD)

flux and number of grains per gram of sediment, oxygen ($\delta^{18}O$) and carbon ($\delta^{13}C$) stable

isotopes in benthic foraminiferal tests, and the flux and abundance of foraminifera are

presented.

**Figure 5:** The absolute abundance (expressed as the number of individuals per gram of dry

sediment) and the percentage of the dominant benthic foraminifera.

**Figure 6:** The dominant components of the monothalamous assemblages. The abundance is

expressed as the percentage of the monothalamous sequences and the most abundantly

sequenced taxa are presented. The trend (2-point average) is indicated with a dashed line.

**Figure 7:** The percentage share of certain clades in the allogromiid sequences.

**Figure 8:** The percentage of sequences of dominant diatom taxa vs. time. The trend (2-point

average) is indicated with the dashed line.

**Table captions**

**Table 1:** Raw and calibrated AMS$^{14}$C dates used in the age model.

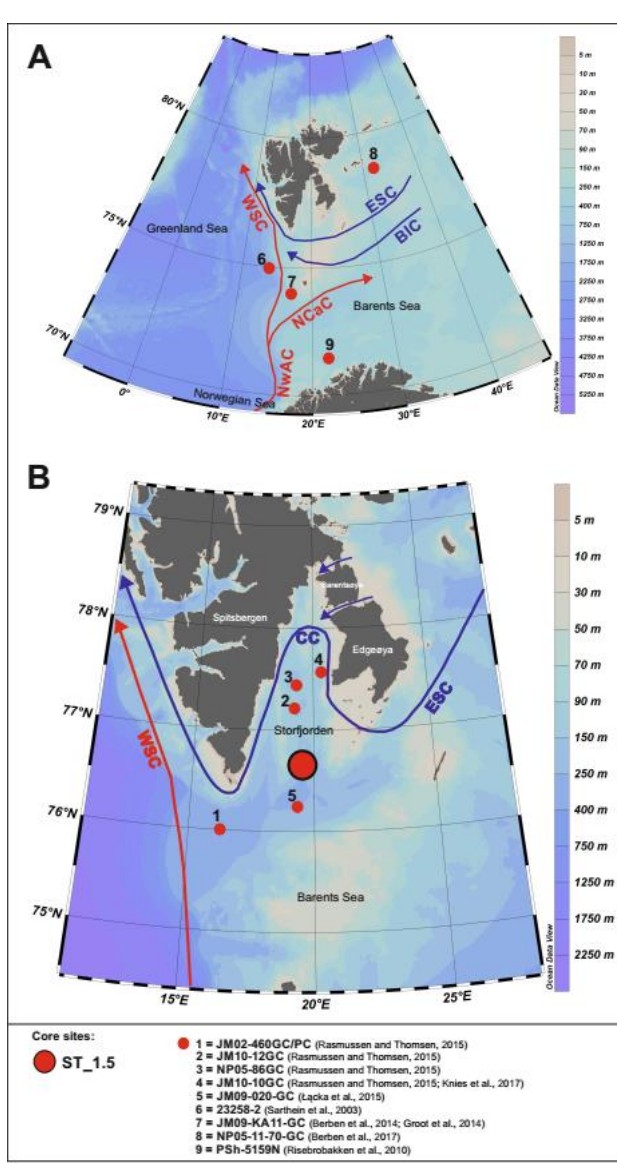

9 **Figure 1:** The modern oceanography of the study area (A) and the location of the studied core ST_1.5 (B) and
10 the other cores discussed in this paper (A,B). Abbreviations of main surface currents: WSC – West Spitsbergen
11 Current, NwAC – Norwegian Atlantic Current, NCaC – North Cape Current, ESC – East Spitsbergen Current,
12 BIC – Bear Island Current, CC – Coastal Current.

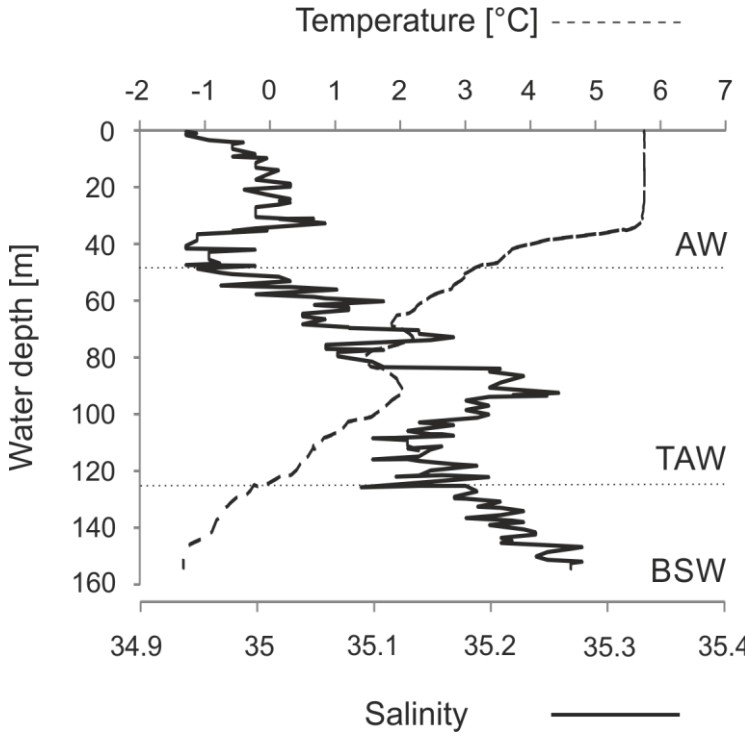

Figure 2: Temperature and salinity profile from the sampling station. Temperature is marked with a dashed line, and salinity is marked with a black line. Abbreviations: AW – Atlantic Water, TAW – Transformed Atlantic Water, BSW – Brine-enriched Shelf Water.

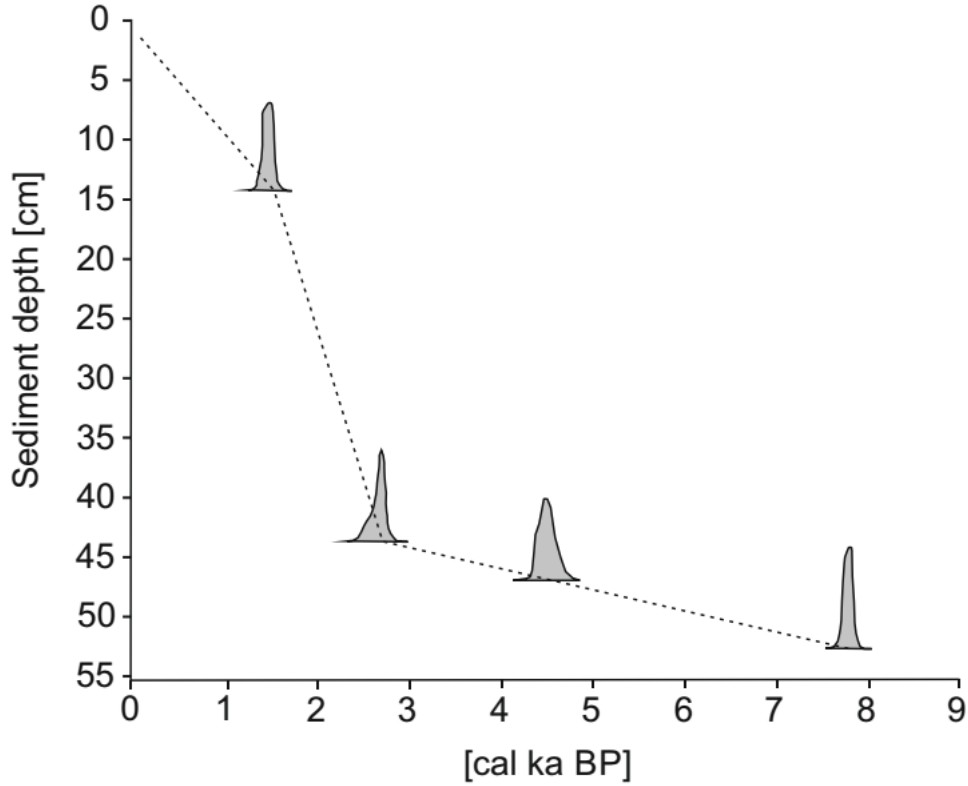

**Figure 3:** Age–depth model of the ST_1.5 core. The grey silhouettes show probability distribution of calendar dates that were obtained by calibration of individual [14]C dates used for the age model. The dotted line shows the age-depth model derived from a linear interpolation between the dates.

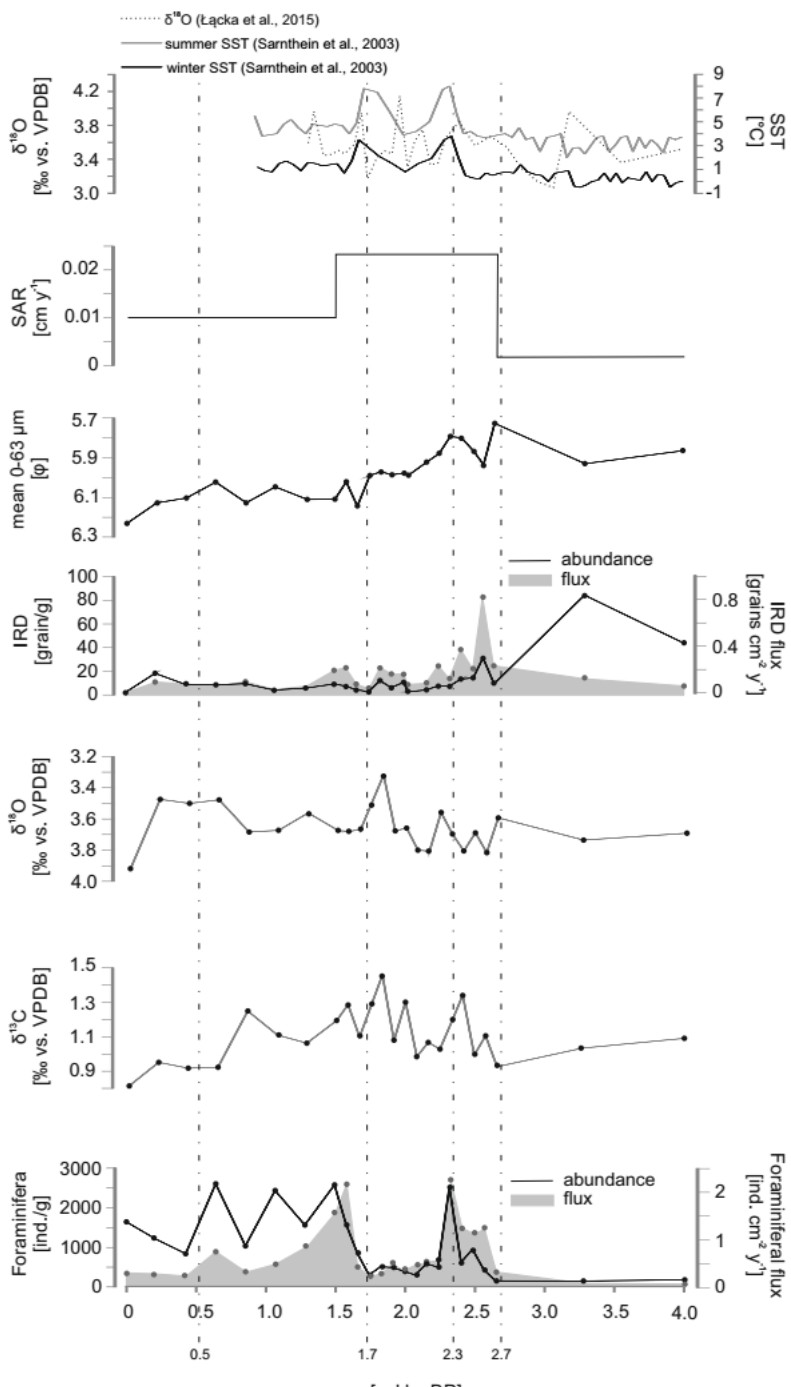

**Figure 4:** Sedimentological and micropaleontological data plotted versus age. The sediment accumulation rate
(SAR), mean grain size of the 0-63-μm fraction, ice-rafted debris (IRD) flux and number of grains per gram of
sediment, oxygen ($\delta^{18}$O) and carbon ($\delta^{13}$C) stable isotopes in benthic foraminiferal tests, and the flux and
abundance of foraminifera are presented.

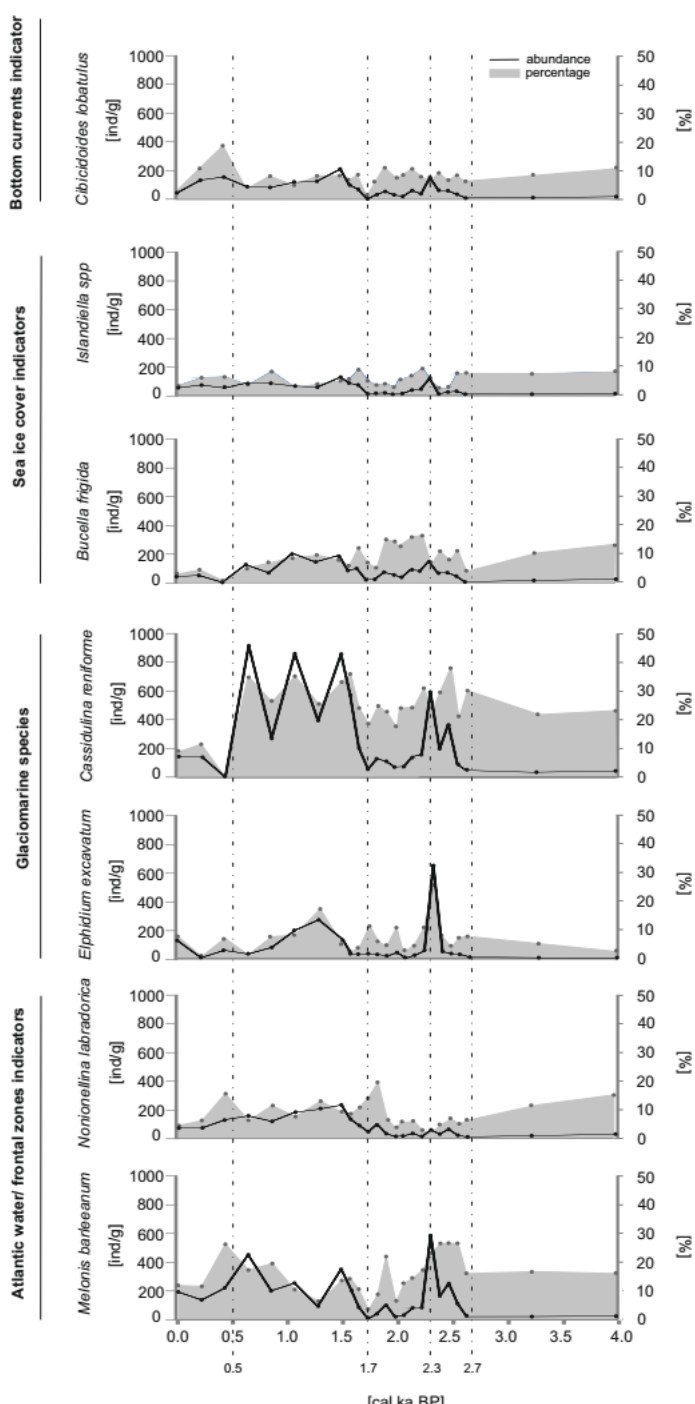

2 **Figure 5:** The absolute abundance (expressed as the number of individuals per gram of dry sediment) and the
3 percentage of the dominant benthic foraminifera.

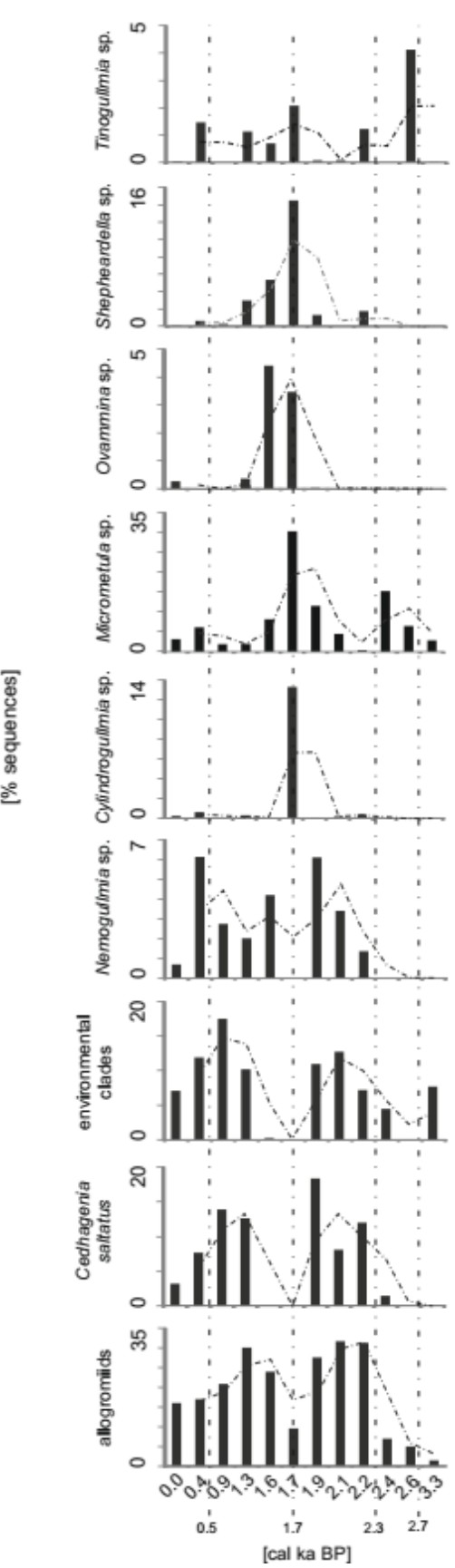

**Figure 6:** The dominant components of the monothalamous assemblages. The abundance is expressed as the percentage of the monothalamous sequences and the most abundantly sequenced taxa are presented. The trend (2-point average) is indicated with a dashed line.

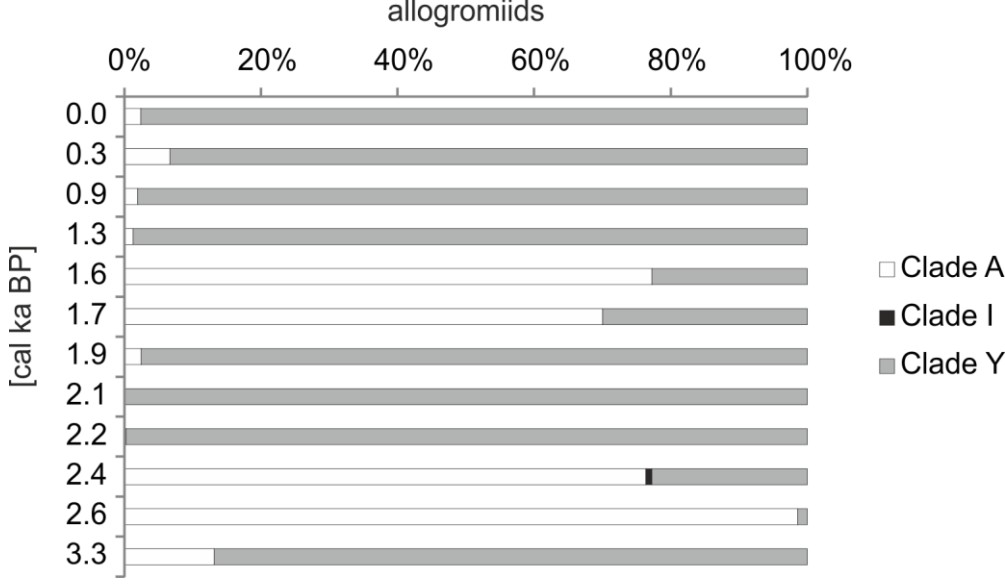

2    **Figure 7:** The percentage share of certain clades in the allogromiid sequences.

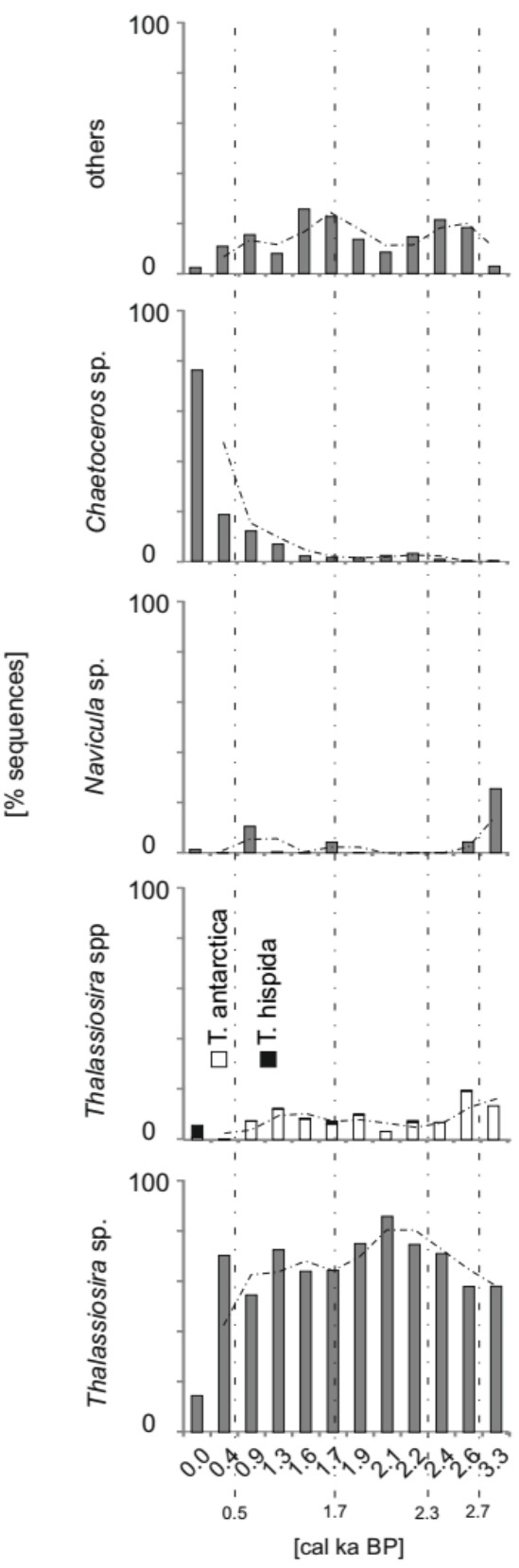

2
3 **Figure 8:** The percentage of sequences of dominant diatom taxa vs. time. The trend (2-point average) is
4 indicated with a dashed line.

**Table 1:** Raw and calibrated AMS$^{14}$C dates used in the age model. B stands for bivlave shells, while F stands for benthic foraminifera tests.

| Core depth [cm] | Material | Raw AMS $^{14}$C | Cal. a BP ± 2σ | Cal. a BP used in age model |
|---|---|---|---|---|
| 2.5 | *Nuculana pernula* (**B**) | 107.38 ± 0.33 pMC | - | - |
| 5.5 | *Yoldiella lenticula* (**B**) | 290 ± 30 BP | - | - |
| 14.5 | *Turitella erosa* (**B**) | 2020 ± 30 BP | 1356-1555 | 1500 |
| 43.5 | *Yoldiella solituda* (**B**) | 3010 ± 50 BP | 2484-2787 | 2700 |
| 46.5 | *Nonionellina labradorica* (**F**) | 4490± 40 BP | 4400-4701 | 4500 |
| 52.5 | *Yoldiella lenticula* (**B**) | 7545 ± 35 BP | 7803-7989 | 7890 |