# Peer review of "Multiproxy evidence of the Neoglacial expansion of Atlantic Water to eastern Svalbard"

_Climate of the Past, 2019_

## Referee Comment (RC1) · Anonymous Referee #1 · 10 Apr 2019

This paper presents an interesting multiproxy dataset to document the paleoceanography near Svalbard and compares traditional sedimentary and microfossil proxies with a novel approach involving ancient environmental DNA. As such, the dataset certainly deserves publishing, but I have some comments/reservations about the age model and the discussion of the results. The discussion has some writing-technical issues. In several cases the own results are presented, without clear arguments supporting the interpretation (e.g. P12, L9–11 & L28–30; P15, L12–15) but rather followed by a literature review. The own results need to be better used to document the paleoceanographic/ environmental signal that is gained from this new site and data, before comparing to the literature. Figures integrating the own results with key records from previous studies is also advised.

Major comments

First of all, the raw data needs to be made publicly available and/or presented with the manuscript. Needed are tables that list unique sample labels and relevant metadata such as core coordinates, sampling depths, measured data for each proxy (sedimentology, foraminifer assemblage data, stable isotopes and aDNA), etc.

Age model. The ages used for the age model seem arbitrary. What is the argument to choose 1500, 2700 and 7890 yr BP? Those ages are not the average of the 2sigma calibrated yrs BP. The most up-to-date radiocarbon calibration (Calib 7.1) was not used. Why? There is 9 cm sediment between 2700 and 7890 cal yr BP (43.5–52.5 cm), or a sedimentation rate of 0.0017 cm/yr assuming a constant sedimentation rate. Have you considered the possibility of a hiatus? Are there changes in the sedimentology/lithology? Additional dating could help solve this issue. Using your proxies to support the age model (P10, L23), make your environmental interpretation become circular. You need to separate the age model from the environmental proxies.

Methods. This type of study (aDNA) is still very new in paleoceanography and more details about the aDNA method would be useful. For example, a short account of the bioinformatics (how were sequences translate to OTUs) would be advisable, rather than referring to other papers. How did you determine that the aDNA was in fact ancient?

Discussion. You write in the results section (P7, L18-20): "However, the extremely low time resolution between 9 cal ka BP and 4 cal ka BP precluded making any general conclusion about that interval. Therefore, the manuscript focuses only on the last 4 cal ka BP (the Neoglacial)." It is not clear where the 9 and 4 cal ka BP come from? The only "certain" ages are 7890 and 2700 cal yr BP (but see my comments above)

[Figure]

measured in samples that are only 9 cm away from each other, and thus showing an extremely low time resolution. With only 2 samples analysed in this interval, this is clearly not sufficient to warrant the lengthy discussion (P10–12) on the interval prior to 2700 yr BP. While the fossil assemblages and aDNA may give valuable information about the environment, it is not possible to say something meaningful with regard to timing of events in this interval. That would require analysis of additional samples and 14C dates (but preferably a record with a higher sedimentation rate).

Higher current speeds (i.e. P.11, L5) can strongly influence paleoceanographic records. What is the effect of bottom water currents on the microfossil and aDNA records here? Could this bias your interpretation?

Do the foram assemblages, and diatom and foram DNA assemblage data show a change supporting the interpreted shift from polynya conditions to densely packed sea ice environment at 2700 cal yr BP?

The AW pulses at 2.3 and 1.7 cal kyr BP show an opposite pattern in foraminifer flux and abundance (Fig. 3, lower two panels): low at 1.7, while high at 2.3 cal kyr BP. Why are these such different patterns to AW pulses? How does this compare to the aDNA records?

You claim an increase in fresh phytodetritus and/or phytoplankton blooms (e.g. P16, L4), but do you actually document this? It seems this is being inferred from the foram assemblages. More cautious wording is advised here.

How does the aDNA signal reflect sea ice cover? You refer to the genera Navicula and Thalassiosira as occurring in sea ice, but these genera also occur elsewhere. For example, Thalassiosira is very diverse in temperate regions (Hoppenrath et al. 2007, Eur. J. Phycol.). Did you identify Thalassiosira species that occur in sea ice, or does the aDNA data not allow to classify to species level?

Several studies in the region are mentioned in the discussion (e.g. Sarnthein et al.

2003, Rasmussen and Thomson 2014, Knies et al. 2017), some of which apparently show comparable signals. This should be discussed in more detail (i.e. what is comparable), and preferably supported by a clear figure showing the key-proxies from those studies that show similarities with the own records.

Minor comments

P5 – sampling. The core was sampled every cm and at 5 cm for aDNA. Were all other proxies also analysed at 5 cm or at 1 cm? A list/table with raw data would help answer this question.

P5, L8&11. aDNA sampling interval at 5 cm – repetition. It would be more informative to have a list of the sample depths.

P6, L16. Please list these 27 levels. And provide raw data.

P6, L22. What is the primer length?

P8, L23. Specifiy "certain species".

P9, L23. Please specify the being and end of the time intervals.

P10, L21 (and throughout). Please remove ST_1.5. You analysed only one core in this study, so that does not have to be repeated.

P11, L17. Codominant – be careful with this term, as it means that the species/groups are equally dominant. Is that always the case?

P12, L9–11. What does this mean in terms of environment/paleocenaography?

P12, L28–30. As above. It would help to put P13, LL4–8 first in the paragraph.

P13, L12–14. What data that you present do you base this interpretation on?

P13, L15. Which diatom aDNA sequences? Could these be transported (currents) rather being than reflection of local production?

P14, L2. . . . are not [a] coherent. . .

P14, L9. This is speculation.

P14, L24–34. It is not clear what the conclusion is from this list of examples.

P15, L12–16. it is not clear what are own results and what comes from literature.

P15, L16. The IP. . . (capital)

P15, L25. Can you identify the LIA in your record?

P16, L4. Did you actually prove phytoplankton blooms occurred or rather that benthic forams responded to changes in environment and productivity?

---

## Referee Comment (RC2) · Anonymous Referee #2 · 26 Apr 2019

The authors present new study on multicentennial environmental reconstruction of eastern Svalbard region over the last ca. 4000, the so-called Neoglacial. Well established proxies (sedimentary, geochemical and microfossils) along with very novel molecular approach (foraminifera and diatom derived ancient DNA) were studied in marine sediment core in order to deliver the broad database for the paleo-interpretations. The study area was already investigated in number of studies, however here the authors tested new molecular proxy, which seems to well support and improve the interpretations based on standard tools. In my opinion, the study is well worth to be

published after some, rather minor improvements, particular of the discussion chapter. Please follow the detailed comments below:

Detailed comments:

Abstract: Perhaps it could be more pronounced why the authors choose Storfjorden for the area of study and what is the specific importance of the region.

Introduction: Page 3, line 27. Wouldn't be enough to refer only to the published study of Pawłowska?

Study area: Page 4, lines 27-29. The location of the studied sediment core seems to be rather off the Storfjorden, in the trough, thus I wonder if the study area descriptions, including low energy and high SAR environment, are still applying?

Do you know what is the thickness of AW branch that enters the core location, does it affect the bottom environment directly, do you have modern bottom temperature and salinity data?

Sampling: With a relatively short sediment core, the aDNA sampling resolution could be higher.

Fossil foraminifera: It should be mention somewhere what was the resolution of fossil foraminiferal analysis, I assume it was every 1 cm.

Page 6, line 15. Please provide full name of the species as it is mentioned here for the first time.

Do you have any possible explanation for the low time resolution between 7890 and 2700 cal BP? Strong bottom currents or possible sediment slide?

Page 8, lines 33-34 to page 9, line 1. The mentioned three percentage values, what are they refer to, it is not clear from the sentence, consider rewording.

Foraminiferal aDNA:

The authors focus only on soft walled monothalamea group with regard to molecular record. Do the authors plan to relate the fossil and the molecular records of hard walled foraminifera as well? Perhaps the agglutinated taxa which are also difficult to stay preserved could be investigated molecularly.

Discussion: Overall, I would like to suggest including 'chronological' headlines into the discussion chapter e.g. 'Interval prior to 2.7 ka BP' , 'Episodes of enhanced AW inflow' et. al. to make it easier for the reader to follow.

Page 10, lines 25-31. It would be highly recommended to provide summary figure that would visualize the correlation between your results and the cited studies.

Page 11, line 4-7. Can the strong currents provide also unfavorable conditions for benthic foraminifera and explain generally very low fauna abundance? Or this is related exclusively with heavy sea ice cover? Is there any detectable response from current velocity indicators like C. lobatulus?

Line 2-5. Might be that IRD and higher mean grain size can also source from extensive transport of shore sea ice?

Page 12, line 6-8. Is it possible to detect the past occurrence of dense brines transport to the bottom in the foraminiferal isotopic signatures measured by the authors?

Line 24-25. Yet, no clear response from C. lobatulus.

Page 13, line 12-14. Here, the authors explain brines as a source of water mixing and nutrient supply, with a positive effect of foraminiferal fauna, whereas for the interval prior to 2.7 cal ka BP, brine formation is presented as a hazardous factor, which seems to be a bit confusing.

Line 29-31. Was the strong bottom current activity reflected also in the changes in grain size fraction?

Page 15, line 25. The authors mentioned LIA but what about the other prominent
climatic events that occurred during the last 2 ka. Can the results be related to them as well, if not, can the authors discuss the possible reason for the lack of larger scale climatic signals, e.g. perhaps local variability. The discussion could improve from a bit broader overview of other Svalbard records, that also underly the AW inflow.

General comment, can the authors observe a relation of the reconstructed higher bottom current activities and the diversity of fragile, soft organic-walled monothalamids?

Figures:

Fig. 3. Could you perhaps mark the sampling points on the graphs. It seems as for the interval 4 ka BP to 2.7 ka BP there are very few sampling points, thus there is almost no detectable variability in the data. Would it be reasonable to consider sediment turbation and homogenization of the signals in such a small thickness of sediment?

The dash lines indicating intervals are very useful, you could probably apply them also to figure 4 and 5 and 7 so it is easier to compare the data.

Fig. 4. I would suggest to change scale down to 30% in order to have better overview for the potential variability, except C. reniforme.

Fig. 5. The age scale is bit too compacted, please consider stretching it.

Fig. 6. 'Clade I' was not mentioned in the result chapter, does it stand for 'environmental clade' (page 9, line 21)?

―――――――――――――――――

---

## Author Comment (AC1) · 21 Jun 2019

We would like to thank the Referee for constructive review, that will help us to improve the manuscript. Written below are our responses to the Referee's comments. The comments were reproduced and are followed by our responses. Based on the comments, we propose the changes of the manuscript. The revised version of the manuscript will be prepared based on the decision of the Editor. Anonymous Referee #1

This paper presents an interesting multiproxy dataset to document the paleoceanography near Svalbard and compares traditional sedimentary and microfossil proxies with a novel approach involving ancient environmental DNA. As such, the dataset certainly deserves publishing, but I have some comments/reservations about the age model and the discussion of the results. The discussion has some writing-technical issues. In several cases the own results are presented, without clear arguments supporting the interpretation (e.g. P12, L9–11 & L28–30; P15, L12–15) but rather followed by a literature review. The own results need to be better used to document the paleoceanographic/ environmental signal that is gained from this new site and data, before comparing to the literature. Figures integrating the own results with key records from previous studies is also advised.

Major comments

Referee's comment: First of all, the raw data needs to be made publicly available and/or presented with the manuscript. Needed are tables that list unique sample labels and relevant metadata such as core coordinates, sampling depths, measured data for each proxy (sedimentology, foraminifer assemblage data, stable isotopes and aDNA), etc.

Response: According to the Reviewer's suggestion, the raw data will be provided as electronic supplementary material.

Referee's comment: Age model. The ages used for the age model seem arbitrary. What is the argument to choose 1500, 2700 and 7890 yr BP? Those ages are not the average of the 2 sigma calibrated yrs BP. The most up-to-date radiocarbon calibration (Calib 7.1) was not used. Why?

Response: The calibration was refined with the use of the latest version of Calib program. However, the calibration dataset (Marine 13, Reimer et al. 2013) remained the same, thus the obtained results of calibration have not changed. The dates used in the age model marked the tops of probability curves on the probability distribution plot provided by Calib 7.1 program (see Fig. 2).

Referee's comment: There is 9 cm sediment between 2700 and 7890 cal yr BP (43.5–52.5 cm), or a sedimentation rate of 0.0017 cm/yr assuming a constant sedimentation rate. Have you considered the possibility of a hiatus? Are there changes in the sedimentology/lithology? Additional dating could help solve this issue. Using your proxies to support the age model (P10, L23), make your environmental interpretation become circular. You need to separate the age model from the environmental proxies.

Response: We agree with the Reviewer that additional dating would improve the age model. According to the linear age model, the beginning of the Neoglacial was recorded at 46 cm sediment depth. Therefore, we decided to provide an additional radiocarbon date from this layer. The dating of foraminiferal tests revealed the age of 4.5 cal ka BP, which confirms our previous age estimation. We also agree that environmental proxies should be separated from the age model, therefore, we decided to remove the sentence considering our proxy record from the mentioned above paragraph. The low sediment accumulation rate recorded for the period from 7890 to 2700 yr BP was most likely a result of glacial retreat and consequent low delivery of sedimentary material. SAR recorded in the studied core was consistent with the results obtained by ŁÄĚcka et al. (2015) in Storfjordrenna for this time period. On the other hand, Knies et al. (2017) and Rasmussen and Thomsen (2015) recorded higher accumulation rates in the inner Storfjorden. However, their studied cores were located relatively close to the shore, and, in our opinion, were more affected by sedimentary material delivery.

Referee's comment: Methods. This type of study (aDNA) is still very new in paleoceanography and more details about the aDNA method would be useful. For example, a short account of the bioinformatics (how were sequences translate to OTUs) would be advisable, rather than referring to other papers. How did you determine that the aDNA was in fact ancient?

Response: We have followed the Reviewers suggestion and added a broader description of post-sequencing data analysis. The added text is as follows: The post-sequencing data processing was performed with the use of SLIM web app (Dufresne

et al., 2019) and included demultiplexing the libraries, joining the paired-end reads, chimera removal, Operational Taxonomic Units (OTUs) clustering, and taxonomic assignment. Sequences were clustered into OTUs using Swarm module (Mahe et al. 2014) and each OTU was assigned to the highest possible taxonomic level using vsearch (Rognes et al., 2016) against a local database and then reassigned using BLAST (Altschul et al., 1990). In order to ensure that obtained results represent ancient DNA, we have kept stringent precautions at each step of the analysis, from sampling to laboratory analysis. These include samples storage and processing in a sterile environment, using physically isolated work area at each step of the analysis and providing negative (blank) controls during DNA extraction, PCR amplification, and quantification. The DNA extraction was performed in the laboratory free from foraminiferal and diatom DNA in the Institute of Oceanology PAN, while PCR amplification and DNA sequencing were performed in laboratories adapted for work with ancient environmental DNA at the University of Geneva.

Referee's comment: Discussion. You write in the results section (P7, L18-20): "However, the extremely low time resolution between 9 cal ka BP and 4 cal ka BP precluded making any general conclusion about that interval. Therefore, the manuscript focuses only on the last 4 cal ka BP (the Neoglacial)." It is not clear where the 9 and 4 cal ka BP come from? The only "certain" ages are 7890 and 2700 cal yr BP (but see my comments above) measured in samples that are only 9 cm away from each other, and thus showing an extremely low time resolution. With only 2 samples analysed in this interval, this is clearly not sufficient to warrant the lengthy discussion (P10–12) on the interval prior to 2700 yr BP. While the fossil assemblages and aDNA may give valuable information about the environment, it is not possible to say something meaningful with regard to timing of events in this interval. That would require analysis of additional samples and 14C dates (but preferably a record with a higher sedimentation rate).

Response: The date 9 000 results from the linear interpolation of accumulation rate based on SAR calculated for the period prior to 7890 cal ka BP. We agree that it is an

oversimplification, therefore we have decided to keep the date 7890 cal ka BP as the oldest certain age. As mentioned above, we have decided to provide additional radio-carbon date. The obtained date was in accordance with the existing age model and confirmed that the onset of the Neoglacial was recorded at 46 cm sediment depth. We agree that the Discussion about the period prior to $\sim$ 2.7 cal ka BP is disproportionately long compared to the low number of samples in this interval. Therefore, we decided to shorten this part of the Discussion. Now the text is as follows: During the period prior to $\sim$ 2.7 cal ka BP, the ST_1.5 sedimentary record displayed elevated and variable IRD delivery and coarsening of the 0-63-$\mu$m sediment fraction (Fig. 4). These results are in agreement with the record from Storfjordrenna (ŁÄĚcka et al., 2015), where peaks in IRD were noted during the Neoglacial and were attributed to increased iceberg rafting due to fluctuations in the glacial fronts (e.g. Forwick et al., 2010). Coarser 0-63 $\mu$m may suggest winnowing of fine grained sediment, however, foraminiferal fauna showed no clear response for sediment removal. The ST_1.5 foraminiferal assemblage was domi-nated by glacier-proximal fauna (primarily C. reniforme) and indicators of frontal zones (primarily M. barleeanum; Fig. 5). The presence of C. reniforme and M. barleeanus is linked to cooled and salty AW (e.g., Hald and Steinsund, 1996; Jernas et al., 2013). Moreover, these species are also associated with the presence of phytodetritus, which may be related to the delivery of fresh organic matter observed in frontal zones and/or near the sea-ice edge (Jennings et al., 2004). Relatively light $\delta$13C (Fig. 4), followed by the maximum percentage of sea-ice species Thalassiosira antarctica (cf Ikävalko, 2004; Fig. 8) may indicate primary production associated with the presence of sea-ice and/or periodic inflow of ArW The typical response of a foraminiferal community to high trophic resources is an increase in diversity and standing stock (Wollenburg and Kuhnt, 2000). According to our data, the foraminiferal community showed no clear signs of in-creased productivity, as the abundance and flux of foraminifera were low prior to $\sim$ 2.7 cal ka BP (Fig. 4). Similarly, Rasmussen and Thomsen (2015) noted a decrease in concentration of benthic foraminifera in Storfjorden at that time, which was attributed to the more extensive seasonal sea-ice cover. Also, Knies et al. (2017) suggested a variable sea-ice cover extent and a fluctuating sea-ice margin in Storfjorden prior to ∼ 2.8 cal ka BP. In contrast, our data may suggest the presence of high-energy environment during the interval prior to ∼ 2.7 cal ka BP, what may be the major factor limiting the development of the foraminiferal community. However, low sampling resolution during that period precluded making any general conclusion and the latter assumption should be confirmed by further studies.

Referee's comment: Higher current speeds (i.e. P.11, L5) can strongly influence paleoceanographic records. What is the effect of bottom water currents on the microfossil and aDNA records here? Could this bias your interpretation?

Response: The change in the grain size in the 0-63 $\mu$m fraction may suggest selective removal of sediment due to the winnowing of fine sediments. However, there was no clear response in fossil foraminifera. Foraminiferal flux and abundance were extremely low at that time and the assemblage was strongly dominated by C. reniforme and M. barleeanum, taxa that are associated with the delivery of fresh phytodetritus. Relatively light $\delta$13C, followed by increased % of aDNA sequences of Thalassiosira antarctica may suggest that primary production was associated with the presence of sea ice at that time. Despite potentially high food supply, foraminiferal standing stock remained low, which may result from higher bottom currents speed and winnowing that limited foraminiferal community development. On the other hand, the flux and abundance of C. lobatulus, which is considered a bottom currents indicator, remained relatively low and stable during the Neoglacial. The major peak in abundance was recorded at ∼0.4 cal ka BP, flowed by minor peaks at ∼ 2.3 and ∼ 1.5 cal ka BP. Our observations are consistent with the record of ŁÄĚcka et al. (2015) from Storfjordrenna. They observed an increase in the mean grain size (> 63 $\mu$m) during the late Holocene (i.e., after 3.6 cal ka BP), what may indicate more vigorous bottom currents and winnowing of fine-grained sediment. However, it was not followed by the increase in C. lobatulus abundance. In the case of monothalamous foraminifera, no bottom currents indicators were identified so far. The knowledge about monothalamids' ecology and environmental tolerance is

incomplete, and using them as a proxy is still limited. Therefore, no clear information about bottom currents activity can be inferred from aDNA record.

Referee's comment: Do the foram assemblages, and diatom and foram DNA assemblage data show a change supporting the interpreted shift from polynya conditions to densely packed sea ice environment at 2700 cal yr BP?

Response: As explained in the Discussion, our record contradicts other interpretations suggesting that Storfjorden was covered by densely packed sea-ice after ∼ 2.7 cal ka BP (cf. Knies et al. 2017). We proposed an alternative scenario that assumed pulsed inflows of AW after ∼ 2.7 cal ka BP, which caused a periodic breakup of sea ice cover and allowed primary productivity. These pulses were recorded in the abundance and taxonomic composition of fossil foraminifera assemblages as well as in shifts in monothalamous foraminifera inferred from aDNA. Moreover, the presence of diatom aDNA during the entire Neoglacial suggested continuous primary production (see P13, L9 – P14, L34). Referee's comment: The AW pulses at 2.3 and 1.7 cal kyr BP show an opposite pattern in foraminifer flux and abundance (Fig. 3, lower two panels): low at 1.7, while high at 2.3 cal kyr BP. Why are these such different patterns to AW pulses? How does this compare to the aDNA records? Response: Indeed, the response of the foraminiferal community showed differences between ∼ 2.3 cal ka BP and ∼ 1.7 cal ka BP. The dominant components of foraminiferal assemblage at ∼ 2.3 cal ka BP were M. barleeanum and E. excavatum, while at ∼ 1.7 cal ka BP, N. labradorica and C. reniforme reached higher percentages. The major difference in environmental conditions between these two "AW episodes" was noticeably coarser 0-63 $\mu$m sediment fraction noted ∼ 2.3 cal ka BP, what may indicate more intensive winnowing and consequent sediment sorting, what creates favorable conditions for development of highly opportunistic species, such as E. excavatum, which reached its' maximum flux and percentage at that time. Moreover, slightly lighter $\delta$18O and $\delta$13C at ∼1.7 cal ka BP suggested a slight difference in AW characteristics. The difference may be supported by the presence of more diverse monothalamous assemblage and the occurrence of

sequences of diatom T. hispida at ∼ 1.7 cal ka BP. The relevant information has been added to the Discussion.

Referee's comment: You claim an increase in fresh phytodetritus and/or phytoplankton blooms (e.g. P16, L4), but do you actually document this? It seems this is being inferred from the foram assemblages. More cautious wording is advised here.

Response: We agree with the Reviewer's comment. The sentence has been modified to "Warming was associated with pulsed inflows of AW and sea-ice melting, which may stimulate phytoplankton blooms and organic matter supply to the bottom".

Referee's comment: How does the aDNA signal reflect sea ice cover? You refer to the genera Navicula and Thalassiosira as occurring in sea ice, but these genera also occur elsewhere. For example, Thalassiosira is very diverse in temperate regions (Hoppenrath et al. 2007, Eur. J. Phycol.). Did you identify Thalassiosira species that occur in sea ice, or does the aDNA data not allow to classify to species level.

Response: We have manually checked the sequence assignment. The majority of diatom sequences were assigned to Thalassiosira sp., and it was not possible to assign them to species level. However, we identified the sequences belonging to Thalassiosira antarctica, which is a sea-ice species. We have modified the paragraph of the Discussion considering the sea-ice diatoms. Now the text is as follows: The record of diatom aDNA supports the latter assumption, as the percentage of sea-ice species Thalassiosira antarctica (cf. Ikävalko, 2004) reached its maximum during this period. Referee's comment: Several studies in the region are mentioned in the discussion (e.g. Sarnthein et al. 2003, Rasmussen and Thomsen 2014, Knies et al. 2017), some of which apparently show comparable signals. This should be discussed in more detail (i.e. what is comparable), and preferably supported by a clear figure showing the key-proxies from those studies that show similarities with the own records. Response: The data showing temperature and isotopic records from GISP2 core (Cuffey and Clow, 1997; Alley, 2000) and Storfjordrenna (ŁĂĔcka et al., 2015), as well as temperature

records of Sarnthein et al., (2003), have been added to the Figure 3. Moreover, more detailed information about comparable signals has been added to the Discussion.

Referee's comment: Minor comments P5 – sampling. The core was sampled ever y cm and at 5 cm for aDNA. Were all other proxies also analysed at 5 cm or at 1 cm? A list/table with raw data would help answer this question. P5, L8&11. aDNA sampling interval at 5 cm – repetition. It would be more informative to have a list of the sample depths. P6, L16. Please list these 27 levels. And provide raw data.

Response: The repetition has been removed from the text. The raw data including sampling resolution will be added to the manuscript as an electronic supplement.

Referee's comment: P6, L22. What is the primer length?

Response: The length of primers is approximately 20 base pairs (bp): the diatom-specific primers are 22 bp long, while foraminifera-specific primers are 19 bp-long. The full sequences of primers are provided in the Material and methods section in the manuscript.

Referee's comment: P8, L23. Specify "certain species".

Response: Herein, by "certain species" we mean dominant species. To avoid confusion, the phrase "certain species" have been removed.

Referee's comment: P9, L23. Please specify the being and end of the time intervals.

Response: The mentioned above time intervals spanned the period from $\sim$ 4 cal ka BP to 2.4 cal ka BP and $\sim$ 1.7 cal ka BP. The relevant information has been added to the text.

Referee's comment: P10, L21 (and throughout). Please remove ST_1.5. You analyzed only one core in this study, so that does not have to be repeated.

Response: The repetitions have been removed from the text.

Referee's comment: P11, L17. Codominant – be careful with this term, as it means that the species/groups are equally dominant. Is that always the case?

Response: Each of the mentioned above foraminifera indicators groups made up to 40% of foraminiferal abundance. However, we have decided to change the word "codominated" to "dominated".

Referee's comment: P12, L9–11. What does this mean in terms of environment/paleocenaography?

Response: Our record displayed an almost 10-fold increase in sediment accumulation rate, accompanied with a decrease in IRD delivery and coarsening of <63 $\mu$m fraction. The increase in SAR resulted most likely from glacial advance observed in Storfjorden at that time (cf. Rasmussen and Thomsen, 2015) and consequent settling of sedimentary material. Sediment accumulation may be also enhanced by the slowdown of bottom currents, as indicated by the decrease in <63 $\mu$m fraction. Moreover, glacial advance is typically followed by more intensive IRD delivery (cf. Rasmussen and Thomsen 2015). However, Storfjorden was covered by densely packed sea ice at that time (Knies et al., 2017) and the majority of icebergs may be trapped in the innermost part of Storfjorden. The relevant explanations have been added to the text.

Referee's comment: P12, L28–30. As above. It would help to put P13, LL4–8 first in the paragraph.

Response: Indeed, placing the information about benthic foraminifera abundance and change in diatom community at the beginning of the paragraph will make our interpretation more clear and easy-to-follow. Therefore, we have modified the paragraph according to the Reviewer's suggestion.

Referee's comment: P13, L12–14. What data that you present do you base this interpretation?

Response: The proposed scenario is based on the alkenone record from Storfjordrenna provided by ŁÄĚcka et al. (article after review)

Referee's comment: P13, L15. Which diatom aDNA sequences? Could these be transported (currents) rather being than reflection of local production?

Response: Herein, we mean diatom sequences in general. Our aim was to pay attention to the continuity of the diatom aDNA record over the Neoglacial. The changes in taxonomic composition were discussed in the other parts of the discussion. We agree that diatoms may be transported by sea currents. However, the record was dominated by one genus (Thalassiosira) and taxonomic composition was relatively stable in the entire record, therefore there are no clear signs of the presence of extraneous taxa.

Referee's comment: P14, L2. . . . are not [a] coherent . . .

Response: The sentence has been corrected.

Referee's comment: P14, L9. This is speculation.

Response: Indeed, Clade Y is still poorly studied, therefore most information about its ecology are assumptions. Therefore, we have decided to remove the latter part of the sentence.

Referee's comment: P14, L24–34. It is not clear what the conclusion is from this list of examples.

Response: The aim of this paragraph was to shortly describe the monothalamous taxa recorded in the studied core and to highlight the relation of listed taxa to the presence of phytodetritus. The general conclusions about the changes in monothalamous assemblages are presented in the following paragraph (P15, L1-11).

Referee's comment: P15, L12–16. it is not clear what are own results and what comes from literature.

Response: There was a mistake in the sentence, the word "and" is unnecessary. Now the text is as follows: The decrease in the percentage of foraminiferal sea-ice indicators

that started after $\sim$ 1.7 cal ka BP suggests a gradually diminishing sea-ice coverage in Storfjorden (Fig. 4). Modern-like conditions were established in Storfjorden $\sim$ 0.5 cal ka BP, with seasonally variable sea-ice cover resulting in intensified but variable polynyal activity (Rasmussen and Thomsen, 2014; Knies et al., 2017).

Referee's comment: P15, L16. The IP . . . (capital)

Response: The sentence has been corrected.

Referee's comment: P15, L25. Can you identify the LIA in your record?

Response: Yes, it is possible to identify LIA in our record, however, it spanned only one sample (at 4 cm sediment depth), therefore we avoided making any general conclusion about the LIA.

Referee's comment: P16, L4. Did you actually prove phytoplankton blooms occurred or rather that benthic forams responded to changes in environment and productivity?

Response: We have based our conclusion both on microfossil and molecular records of benthic foraminifera and on molecular record of diatoms. Indeed, microfossil and aDNA record of benthic forams shows response of foraminiferal community to environmental changes, however, the aDNA record of diatoms may be an indicator of the primary production.

---

## Author Comment (AC2) · 21 Jun 2019

We would like to thank the Referee for constructive review, that will help us to improve the manuscript. Written below are our responses to the Referee's comments. The comments were reproduced and are followed by our responses. Based on the comments, we propose the changes of the manuscript. The revised version of the manuscript will be prepared based on the decision of the Editor.

Anonymous Referee #2 The authors present new study on multicentennial environ-

[Discussion paper]

[Figure]

mental reconstruction of eastern Svalbard region over the last ca. 4000, the so-called Neoglacial. Well established proxies (sedimentary, geochemical and microfossils) along with very novel molecular approach (foraminifera and diatom derived ancient DNA) were studied in marine sediment core in order to deliver the broad database for the paleo-interpretations. The study area was already investigated in number of studies, however here the authors tested new molecular proxy, which seems to well support and improve the interpretations based on standard tools. In my opinion, the study is well worth to be published after some, rather minor improvements, particular of the discussion chapter. Please follow the detailed comments below:

Detailed comments: Referee's comment: Abstract: Perhaps it could be more pronounced why the authors choose Storfjorden for the area of study and what is the specific importance of the region.

Response: We agree with this comment. We have added more information about Storfjorden to the Abstract. The added text is as follows: Storfjorden is one of the most important "brine factory" in the European Arctic, responsible for the deep water production. Moreover, it is a climate-sensitive area, influenced by two contrasting water masses: warm and saline Atlantic Water (AW) and colder and fresher Arctic Water (ArW)

Referee's comment: Introduction: Page 3, line 27. Wouldn't be enough to refer only to the published study of Pawłowska?

Response: The study of Pawłowska et al. (2014) considers only foraminifera. Unfortunately, the results of diatom analysis from sediment cores have not been published yet, therefore, it was necessary to refer to personal communication.

Referee's comment: Study area: Page 4, lines 27-29. The location of the studied sediment core seems to be rather off the Storfjorden, in the trough, thus I wonder if the study area descriptions, including low energy and high SAR environment, are still applying?

Response: The core is located in the central Storfjorden, off the through. The study of Winkelmann and Knies (2005), where the sedimentary environment in Storfjorden was described, covers also central and outer parts of Storfjorden.

Referee's comment: Do you know what is the thickness of AW branch that enters the core location, does it affect the bottom environment directly, do you have modern bottom temperature and salinity data?

Response: The temperature and salinity profile from the coring site has been added to the manuscript. During the August 2014, AW occupied the uppermost 47 m, while the intermediate layer was dominated by TAW. In the near bottom layer, BSW was observed.

Referee's comment: Sampling: With a relatively short sediment core, the aDNA sampling resolution could be higher.

Response: Material for aDNA analysis have been taken before the core was dated, therefore, we have decided to sample the core with fixed 5-cm interval. Indeed, the sampling resolution could have been higher. Unfortunately, we have no more material suitable for aDNA analysis to provide a higher resolution record.

Referee's comment: Fossil foraminifera: It should be mention somewhere what was the resolution of fossil foraminiferal analysis, I assume it was every 1 cm.

Response: Fossil foraminifera were analyzed every 2 cm. The appendix with raw data, including sampling resolution, will be provided as electronic supplementary material.

Referee's comment: Page 6, line 15. Please provide full name of the species as it is mentioned here for the first time.

Response: The full name has been added to the sentence.

Referee's comment: Do you have any possible explanation for the low time resolution between 7890 and 2700 cal BP? Strong bottom currents or possible sediment slide?

Response: The low sediment accumulation rate recorded for the period from 7890 to 2700 yr BP was most likely a result of glacial retreat and consequent low delivery of sedimentary material. SAR recorded in the studied core was consistent with the results obtained by ŁÄĚcka et al. (2015) in Storfjordrenna for this time period. On the other hand, Knies et al. (2017) and Rasmussen and Thomsen (2015) recorded higher accumulation rates in the central and inner Storfjorden. However, their studied cores were located relatively close to the shore, therefore, were more affected by sedimentary material delivery.

Referee's comment: Page 8, lines 33-34 to page 9, line 1. The mentioned three percentage values, what are they refer to, it is not clear from the sentence, consider rewording.

Response: The sentence has been corrected as follows: After ∼ 2.7 cal ka BP, there were AW/frontal zone indicator peaks recorded at 2.4 and 1.8 cal ka BP, where the percentages increased to 33%, and 28% of the total abundance.

Referee's comment: Foraminiferal aDNA: The authors focus only on soft walled monothalamea group with regard to molecular record. Do the authors plan to relate the fossil and the molecular records of hard walled foraminifera as well? Perhaps the agglutinated taxa which are also difficult to stay preserved could be investigated molecularly.

Response: The relation between the molecular and fossil record has been already studied (see Pawłowska et al., 2014; Geobiology) and it was not our intention to duplicate these results. In our study, we decided to focus on monothalamous foraminifera, as they are the dominant component of aDNA record and may provide the most valuable environmental information.

Referee's comment: Discussion: Overall, I would like to suggest including 'chronological' headlines into the discussion chapter e.g. 'Interval prior to 2.7 ka BP' , 'Episodes of enhanced AW inflow' et. al. to make it easier for the reader to follow.

Response: We agree with this comment, headlines have been added to the Discussion

Referee's comment: Page 10, lines 25-31. It would be highly recommended to provide summary figure that would visualize the correlation between your results and the cited studies.

Response: The data showing temperature and isotopic records from GISP2 core (Cuffey and Clow, 1997; Alley, 2000) and Storfjordrenna (ŁÄĔcka et al., 2015), as well as temperature records of Sarnthein et al., (2003), have been added to the Figure 3.

Referee's comment: Page 11, line 4-7. Can the strong currents provide also unfavorable conditions for benthic foraminifera and explain generally very low fauna abundance? Or this is related exclusively with heavy sea ice cover? Is there any detectable response from current velocity indicators like C. lobatulus?

Response: The percentage of C. lobatulus remained relatively stable during the Neoglacial, except for the peak $\sim$ 0.4 cal ca BP and minor peaks at $\sim$ 2.3 and $\sim$ 1.5 cal ka BP. Therefore, we were not able to make any unequivocal conclusions. Moreover, the low number of samples in the interval prior to $\sim$ 2.7 cal ka BP is not sufficient to warrant the lengthy discussion and does not allow to make any general conclusions. Therefore, we decided to shorten the part of the Discussion considering this time interval.

Referee's comment: Line 2-5. Might be that IRD and higher mean grain size can also source from extensive transport of shore sea ice?

Response: Indeed, the sea-ice rafting may be an important source of ice-rafted debris. However, the sampling station was located relatively distant from the shore, therefore, the terrestrial impact was rather minor.

Referee's comment: Page 12, line 6-8. Is it possible to detect the past occurrence of dense brines transport to the bottom in the foraminiferal isotopic signatures measured by the authors?

[Figure]

Response: The 18O and 13C values prior to ∼ 2.7 cal ka BP were relatively stable. However, for this period isotopes were measured in 3 sediment layers, which may affect the result. Therefore, we have added to the mentioned above paragraph conclusion that the potential influence of brines on foraminiferal abundance has to be confirmed by other studies.

Referee's comment: Line 24-25. Yet, no clear response from C. lobatulus.

Response: Indeed, as mentioned above, the percentage of C. lobatulus was rather stable during the Neoglacial. Our observations are consistent with the record of ŁÄĔcka et al. (2015) from Storfjordrenna. They observed an increase in the mean grain size (> 63 $\mu$m) during the late Holocene (i.e., after 3.6 cal ka BP), which was not followed by the increase in C. lobatulus abundance.

Referee's comment: Page 13, line 12-14. Here, the authors explain brines as a source of water mixing and nutrient supply, with a positive effect of foraminiferal fauna, whereas for the interval prior to 2.7 cal ka BP, brine formation is presented as a hazardous factor, which seems to be a bit confusing.

Response: As mentioned above, the low number of samples in the interval prior to 2.7 cal ka BP precluded making any general conclusion. The Discussion considering the influence of sea-ice on foraminifera during that interval has been modified. Now the text is as follows: The ST_1.5 foraminiferal assemblage was dominated by glacier-proximal fauna (primarily C. reniforme) and indicators of frontal zones (primarily M. barleeanum; Fig. 5). The presence of C. reniforme and M. barleeanus is linked to cooled and salty AW (e.g., Hald and Steinsund, 1996; Jernas et al., 2013). Moreover, these species are also associated with the presence of phytodetritus, which may be related to the delivery of fresh organic matter observed in frontal zones and/or near the sea-ice edge (Jennings et al., 2004). Relatively light $\delta$13C (Fig. 4), followed by the maximum percentage of sea-ice species Thalassiosira antarctica (cf Ikävalko, 2004; Fig. 8) may indicate primary production associated with the presence of sea-ice and/or

periodic inflow of ArW The typical response of a foraminiferal community to high trophic resources is an increase in diversity and standing stock (Wollenburg and Kuhnt, 2000). According to our data, the foraminiferal community showed no clear signs of increased productivity, as the abundance and flux of foraminifera were low prior to $\sim$ 2.7 cal ka BP (Fig. 4). Similarly, Rasmussen and Thomsen (2015) noted a decrease in concentration of benthic foraminifera in Storfjorden at that time, which was attributed to the more extensive seasonal sea-ice cover. Also, Knies et al. (2017) suggested a variable sea-ice cover extent and a fluctuating sea-ice margin in Storfjorden prior to $\sim$ 2.8 cal ka BP. In contrast, our data may suggest the presence of high-energy environment during the interval prior to $\sim$ 2.7 cal ka BP, what may be the major factor limiting the development of the foraminiferal community. However, low sampling resolution during that period precluded making any general conclusion and the latter assumption should be confirmed by further studies.

Referee's comment: Line 29-31. Was the strong bottom current activity reflected also in the changes in grain size fraction?

Response: Indeed, there were slight peaks in the 0-63 $\mu$m that coincided with the increase in C. lobatulus. The relevant information has been added to the Discussion.

Referee's comment: Page 15, line 25. The authors mentioned LIA but what about the other prominent climatic events that occurred during the last 2 ka. Can the results be related to them as well, if not, can the authors discuss the possible reason for the lack of larger scale climatic signals, e.g. perhaps local variability. The discussion could improve from a bit broader overview of other Svalbard records, that also underly the AW inflow.

Response: We have followed the Reviewer's suggestion and added a paragraph considering other records from the Nordic Seas. The added text is as follows: Our record revealed two-phase Neoglacial, with a major shift in environmental conditions at $\sim$ 2.7 cal ka BP. According to the ST_1.5 record, the Neoglacial in Storfjorden was not a

constantly cold period, but comprised alternate, short-term cooling and warming periods, associated with variability in sea-ice coverage and productivity. There is various evidence of a shift in environmental conditions in the Nordic Seas region in mid-Neoglacial. Alkenone record from the Norwegian Sea revealed a significant drop of sea surface temperature at 2.7 cal ka BP (Calvo et al., 2002). Risebrobakken et al. (2010) recorded a change in oceanographic conditions in the SW Barents Sea ca. 2.5 cal ka BP, followed by the episodes of reduced surface and subsurface salinity after 2.5 cal ka BP, what was attributed to the expansion of coastal waters and the occurrence of more sea-ice. Moreover, our evidence of the presence of AW in Storfjorden during the Neoglacial supported previous suggestions that AW inflow during the late Holocene was strong enough to reach also the eastern coasts of Svalbard (e.g., ŁÄĚcka et al., 2015). Moreover, Sarnthein et al. (2003) postulated pulses of AW inflow to the western Barents Sea shelf at 2.2 and 1.6 cal ka BP. According to Perner et al. (2015), the Neoglacial delivery of chilled AW to the Nordic Seas culminated between 2.3 and 1.4 cal ka BP. Also, Rasmussen et al. (2014a) and Jernas et al (2013) recorded slightly warmer and less glacial conditions during the last 2 ka on the western Spitsbergen shelf.

Referee's comment: General comment, can the authors observe a relation of the reconstructed higher bottom current activities and the diversity of fragile, soft organic-walled monothalamids?

Response: The most intensive bottom currents were likely to occur during the interval prior to 2.7 cal ka BP. Unfortunately, the aDNA was analyzed only in one sample during this time interval, therefore, we cannot make any general conclusions. Moreover, the knowledge about monothalamids ecology and environmental tolerance is still scarce and incomplete and no bottom currents indicators have been identified in this group so far.

Referee's comment: Figures: Fig. 3. Could you perhaps mark the sampling points on the graphs. It seems as for the interval 4 ka BP to 2.7 ka BP there are very few

sampling points, thus there is almost no detectable variability in the data. Would it be reasonable to consider sediment turbation and homogenization of the signals in such a small thickness of sediment? The dash lines indicating intervals are very useful, you could probably apply them also to figure 4 and 5 and 7 so it is easier to compare the data.

Response: Sampling points and dashed lines have been added to the graphs, according to the Reviewer's suggestion. Our sedimentary record indicated more vigorous bottom currents and consequent winnowing of fine sediment. Therefore, the homogenization of signal may be related to selective removal of mineral and organic particles, rather than turbation.

Referee's comment: Fig. 4. I would suggest to change scale down to 30% in order to have better over view for the potential variability, except C. reniforme.

Response: There are two taxa that exceeded 30% of foraminiferal assemblage – C. reniforme and E. excavatum. We have decided to use the scale reaching up to 50% to clearly show the differences between the percentages of certain taxa and to highlight the dominance of species such as C. reniforme or M. barleeanum. Therefore, we would prefer to keep the scale in its current form.

Referee's comment: Fig. 5. The age scale is bit too compacted, please consider stretching it.

Response: We have prepared the figure according to the Reviewers comment, however, stretching the scale resulted also in the increase in the distance between the data bars and, in consequence, graph became less clear and the trends were less visible. Therefore, we would prefer to keep the scale in its current form.

Referee's comment: Fig. 6. 'Clade I' was not mentioned in the result chapter, does it stand for 'environmental clade' (page 9, line 21)?

Response: Clade I does not stand for the environmental clade. Allogromiids belonging

to Clade I were noted only in one sample, where they made 0.88% of allogromiid sequences. The information about the occurrence of Clade I have been added to the Results section.

---

## Editor Decision (ED1)

We would like to thank the Referees for a constructive review, that helped us to improve the manuscript. Written below are our responses to the Referee's comments. The comments were reproduced and are followed by our responses (in italics).

Referee #1

The manuscript is in places lengthy, incoherent and difficult to follow. The interpretations are on more than one occasion based on a part of the observations, while other (supporting or contradicting) data is not discussed. In a few cases, too much weight is given to minor changes in the data records resulting in not very convincing interpretations. A scenario is presented based on a paper in preparation, which cannot be verified, and integration with existing literature is incomplete. These are substantial problems, and therefore I cannot give a favorable review. I would still like to see this manuscript published, because the multiproxy data definitely deserve publishing. More work is needed on clearly communicating the interpretation (based on all datasets) as well as a better integration with the existing literature. A final small comment: the paper does not address the central question in the title "does aDNA complement traditional methods".

*We would like thank the Referee for a critical and constructive review that helped us to improve the manuscript. Following the suggestions of the Referee, we have modified our interpretation by removing or shortening parts of Discussion concerning only minor changes in the dataset and by adding broader explanations whenever necessary. Moreover, we have added more references to the latest literature from the Nordic Seas region.*

*In our opinion, the question from the title is addressed in the manuscript, as one of our conclusions is that "the molecular record supports and complements sedimentary and microfossil records (...)". Further in the Conclusions section, we highlight the fact that the diatom and foraminiferal aDNA reflected environmental changes inferred from other proxies. However, we did not perform a direct comparison between the microfossil and molecular records. This issue was broadly discussed in our previous studies (Pawłowska et al., 2014; 2016) and it was not our intention to duplicate this discussion in the current manuscript.*

Points of concern:

Generally, in terms of data interpretation, the sea ice indicating foraminifers B. frigida and Islandiella are present from 2.7 ka BP. The sea ice diatom T. antarctica seems present in all samples (although Fig 8 is not easy to read). The fluctuations in these species' abundance is limited (0–10%), yet the changes are interpreted as major shifts in the paleoceanography. Such changes could be explained be errors in counting/measuring. This should be considered more carefully. Similarly, page 13 L31-34,

there is strong interpretation based on a minor change in grain size and C. lobulatus percentage (a change, not clear from the figure).

*Indeed, the sea ice diatom T. antarctica was present in all analyzed samples, as well as foraminiferal sea ice indicators B. frigida and Islandiella spp (see Fig. 5, Fig. 8 and Supplementary materials). This may indicate that sea ice occurred in Storfjorden at least seasonally during the whole studied period. However, there were some variations in the percentages of sea ice indicators suggesting the formation of more extensive sea ice cover.*
*We agree with the Referee that some statements are too strong considering the magnitude of sea ice indicators fluctuations. Therefore, we have modified parts of the discussion concerning sea-ice indicators and we have made conclusions more moderate.*

The discussion of the period from 4 ka to 2.7 BP has been altered from a previous version, but it remains based on 2 samples in 3 cm of sediment (page 11, table 1). With n=2, it is impossible to discuss variability (IRD delivery, coarsening) nor make a (un)favorable comparison with other records. This is over-interpreting the data and has to be toned down.

*We agree that low sampling resolution during this period precluded making general conclusion (what was stated at the end of the paragraph). The discussion considering the period prior to 2.7 cal ka BP have been further shortened and conclusions became more general.*

The results remain poorly integrated with the study of Knies et al. 2017 (p. 13, L3 onwards) and are not necessarily contradictory as is claimed. Knies et al. 2017 suggest a permanent sea ice cover in Storfjorden for 2.8–0.5 ka BP, but do not exclude the possibility that AW inflow occurred as their IP25 record has too low resolution. The foram assemblage data of this study records a dominantly glaciomarine setting (up to 0.5 ka BP), with some AW and sea ice influence. In my view, this is not necessarily inconsistent with Knies et al. However, the authors present a scenario based on a paper in preparation (Lacka et al.), which cannot be verified.

*We agree with the Referee's comment concerning the interpretation of results of Knies et al. (2017). The broader explanation considering the results of Knies et al. (2017) have been added to the Discussion.*
*The paper of Łącka et al. (2019) is now published. The reference has been added to the manuscript.*

It is also not clear from the lengthy description on p. 14-15 how the foraminiferal DNA data actually supports the interpretation of AW pulses and/or changes in sea ice conditions. The diatom DNA is not

discussed here, while T. antarctica was continuously recorded in the core – suggesting a constant presence of sea ice?

*Monothalamous species found in the aDNA record may be divided into two groups. During the episodes of AW inflow, the increase in percentage of taxa associated with the delivery of fresh phytodetritus. Conversely, colder periods, characterized by the more extensive sea ice cover were characterized by the dominance of more opportunistic monothalamous taxa.*

*Indeed, the continuous record of T. antarctica may indicate the presence of at least seasonal ice cover at the study site. Moreover, the pulses of AW were associated with the occurrence of DNA sequences of T. hispida, an open water species. The occurrence of sequences of both these taxa may suggest the formation of ice cover during winter-spring, followed by ice-free summers. Similar scenario was proposed by Berben et al. (2017), who suggested increased AW to the eastern Svalbard and partial summer sea ice occurrence after 2.7 cal ka BP. According to record of Łącka et al. (2019) from Storfjordrenna, the sea-ice melting induced the production of brines that may launch convective mixing and nutrient resupply from the bottom what stimulated primary production. These conditions supported the development of phytodetritus-dependent monothalamous taxa.*

*Conversely, the colder phases of the Neoglacial were characterized by heavy and densely packed sea ice resulting in limited productivity (Knies et al., 2017). The presence of T. anatrctica sequences and disappearance of T. hispida may suggest that primary production was associated with sea-ice. Furthermore, the monothalamous assemblage was less diverse and was dominated by more opportunistic taxa, what may indicate reduced supply of organic matter to the bottom.*
*All the explanations written above have been included in the discussion.*

Chapter 6.3: It is difficult to call in a major shift at 2.7 ka BP, with the limited data prior to this time (see above). The chapter lists the own observations, next to a few statements from the literature focusing on the AW observations into the region. But it does not present a coherent picture for the Storfjorden area in this time period. Nothing is mentioned about polynyas and sea ice production, and maybe the clearest shift in environmental conditions at 0.5 ka BP (Knies et al. 2017).

*In order to make a discussion more clear and easy-to-follow, we have added additional sub-chapter spanning the period after 0.5 cal ka BP, and thus we extended discussion considering this period. Moreover, we have added broader explanations of the relationship between sea-ice, AW inflow and productivity in the discussion concerning time interval from 2.7 cal ka BP to 0.5 cal ka BP.*

There are minor spelling and grammatical errors throughout the text. Throughout: figure reference

numbers in the text often refer to the wrong figure. Page 12 – it is unclear what is own data and what is from literature.

*We have carefully checked and corrected the references to figures in the text. We have also send the manuscript to professional language editor.*

**Referee #2**

In this manuscript, the authors present a paleoreconstruction of environmental conditions for the last ca. 4000 years. They used a marine sediment core retrieved from Storfjorden, eastern Svalbard and a multitude of proxies. In particular, the environmental DNA record offers new information regarding past oceanographic conditions. This novel approach provides interesting new data that adds to the existing records of the area. Therefore, I feel that this study plays a positive role towards multi-proxy studies and certainly deserves to be published. Nonetheless, I do have some concerns regarding the interpretation of the data.

GENERAL COMMENTS:

Chronology:
I wouldn´t say that the chronology is the worst for this area. However, with respect to strengthen it, I wonder if you tried (and if not, why not?) to measure any additional core depths for AMS 14C dating? In particular, it seems to me, from Figure 4 & 5, that there is actually foraminiferal material between 5.5 and 14.5 cm as well as between 14.5 and 43.5 cm so I wonder why this is not used.

*The age model is based on AMS$^{14}$C dates inferred from bivalve shells and benthic Nonionellina labradorica. In order to obtain another $^{14}$C dates, we will have to use another foraminifera species, as it'd be difficult to obtain sufficient number of N. labradorica specimens from single sediment layer. Considering the large differences that may occur between $^{14}$C dates obtained from foraminifera species, the age model may have high uncertainty.*

Methodology and results:

Although the general approach and method on itself seems valid, one of the shortcomings (and/or missed opportunities) of this study is the low resolution. I mean, the core is 55 cm (which is not too much material) and the key proxy of this study (what adds the new knowledge) is only sampled for every 5 cm (i.e. 12 samples if I calculate correctly?). Nonetheless, in addition to that, the data in this study is only presented/discussed for the most recent 4000 years. (In my calculations this should then be less then 12 data points for fig 6-7-8. I find this quite confusing, so please explain this?). Regarding the sensitivity of cross-contamination when dealing with DNA samples, I can imagine it might be too

late now. However, I do feel it is a shame that not every cm was investigated. In addition to that, even for the other proxies, despite the core was sampled every cm, the results only shows data for every 2 cm. Something which btw should be better explained in the material and method section.

*The aDNA samples were taken every 5 cm, whereas in the other proxies were analyzed every 2 cm. The relevant explanation have been added to the Materials and methods section.*

*Indeed, the study (especially the aDNA record) will benefit from more data points. Unfortunately, there is no more sedimentary material available for molecular analysis. As Referee mentioned, aDNA samples are very sensitive to contamination and should be subsampled and analyzed according to stringent protocol. At this point, the risk of cross-contamination between the samples and/or the external contamination is high, so it is not possible to obtain material for aDNA analyses in a way that will guarantee the credibility of the results.*

Related to this topic, I´m not entirely sure regarding the division/interpretation of the results/discussion. Specifically, the period prior to ca. 2700 years (sub-chapter 6.1) is based on only 2 data points for foram/IRD/isotopes and only 1 for the DNA data. This seems rather low to me to actually base any conclusions/interpretations on. Furthermore, the DNA record is within this time frame (for obvious reasons) not even discussed. I feel this issue should be more acknowledged and explained within the manuscript.

*We agree, that due to low sampling resolution, the interpretation is difficult. It is already stated at the end of the paragraph concerning period prior to 2.7 cal ka BP that low sampling resolution during this period precluded making any general conclusion. Moreover, we have shortened this part of the discussion and made it more general to avoid any over-interpretation.*

Discussion:
How do you explain the selection of cores you compare your data with? In Figure 1, only very local (in the vicinity of the studied core) marine cores are included. Why not geographically broader? For example, Holocene records on the AW pathway? Such as those presented by studies such as Hald et al., 1996; Hald et al., 2007; Berben et al., 2014; Risebrobakken and Berben, 2019…… OR, what about east of Svalbard? Berben et al., 2017 and references within.

*We agree that the comparison with geographically broader records may improve the manuscript. We have added the comparison of our data with recent records from the Nordic Seas. Also, we have added the location of discussed cores to Fig. 1.*

When adding sea-ice cover to the discussion… What about the results published in Berben et al., 2017; Belt et al., 2015. There, the authors present a sea ice reconstruction based on sea ice biomarkers

and also describe a change of environmental conditions at ca. 2700 years ago with episodic periods of increased AW. This seems to correlate well with the finding of this study and thus, worth to compare with.

*Indeed, these results correspond well with our data. We have added the discussion of the results of Belt et al. (2015) and Berben et al. (2017) in comparison to our data.*

Regarding the structure of the discussion… why did you gave the period before 2.7 ka (based on 2 data points) an entire sub-chapter and all the rest of your record another sub-chapter? Why not splitting 6.2 into different sub-chapters? I think this would make it easier to bring across the main messages.

*Following the suggestion of the Referee, we have added additional sub-chapter spanning the period from 0.5 cal ka BP to present.*

Furthermore, I think the statement at Page15 Line30-31 is also quite simplistic considering the fact the "first" period is based on 2 data points. Thereby, I mean that if you have only 2 data points for 1300 years, it is not so surprising the records looks stable. Concluding, based on that, that the variability/alternating periods etc. start only after 2.7 ka is quite short-sighted. Therefore, I suggest to place your discussion into a wider geographical context (like the broader Barents Sea) as previous studies indicated similar things and therefore, might support the conclusions drawn within this manuscript.

*We agree with the Referee's comment. We have extended the paragraph "Paleoceanographic implications" by including other studies from the Nordic Seas region to support our findings.*

References:
Most of the paleo references within this manuscript are ca. 5 years old (or older) even though this area has been investigated in several more recent studies. Therefore, I recommend including the latest literature within this field both in your introduction and discussion. While doing so, keep a wider geographical area, and not just Storfjorden, in mind. *According the suggestion, we have added references to the latest literature to the manuscript.*

Language:
I believe, there is still some room for improvement when it comes to the writing. In particular, grammar mistakes should be avoided. Furthermore, language can be improved and repetition of the same sentence structure should be avoided. With respect to this comment, I made quite some suggestions in the minor comments.

*We would like thank the Referee for the language corrections, we have changed the text according to the suggestions. We have also send a manuscript for a professional language editing.*

SPECIFIC COMMENTS:

*In general, we have followed the Referees suggestions. If necessary, we have added an additional explanations to the Referee's comments.*

Abstract

P1L13: Change "was" by "is"

P1L15: Why brine factory between brackets? Plus should it not be plural?

P1L16-17: Remove comma. Rewrite. "…masses: warm saline Atlantic Water (AW) and cold fresh Arctic Water…"

P1L18: "…evidence for existing interactions between the AW inflow and…"

P1L27: "… a decreased productivity. …"

P1L29: Add an "s" after variation

P1L29-P2L1: Rewrite sentence. Possible make two of them.

1. Introduction

P2L7: Rewrite. "The northwards flow of….(AW), transported by the ???Current, is …"

P2L8: Rewrite. "…indicates a warming…" Btw, a warming of what?

P2L9: Rewrite. "…increased inflow of AW towards the Arctic…"

P2L10: Rewrite. "…western Svalbard margin during, at least, the last…"

P2L13: Add spaces after 6.8 and before 1 (consistent with notation in line 12)

P2L15-18: Although it is a correct statement, I suggest to include more references as the study of Slubowska-Woldengen is not only one. So, rewrite to (e.g. Slubowska-Woldengen et al., 2007…ADD OTHERS FROM THE AREA…)

P2L18: "…fluctuations of AW inflow (e.g. references)"

P2L21: From the Barents Sea? Be more specific where the ESC comes from.

P2L24-25: "…Storfjorden, East Spitsbergen (Lydersen…"

P2L25: "…Recently, Hansen et al. (2011) suggested that AW…"

P2L26: "…BP) something that was…"

P2L27-28: Do you have a reference for this statement? What about paleoreconstruction studies from the Barents Sea?

P2L30: "…Arctic is characterized by a declined summer …"

P2L31: "…1978) that correlates to a decline…"

P2L31-32: Again, add "e.g." before the references. There many more studies from this wider area that indicate the same. In particular, include more recent studies.

P2L32: "…. Waters and a limited…"

P2L33: "towards the Nordic…"

P2L33: When you use Müller et al., 2012 as a reference. I suggest you specify the location of this study (i.e. West Spitsbergen).

P2L33-P3L2: With respect to the comment above, other studies (such as Berben et al., 2014; 2017) indicate similar increased sea ice conditions (! Based on similar sea ice proxies (i.e. IP25, biomarkers)) for the SW and E of Svalbard. These studies are with respect to sea ice references more appropriate than Sarnthein et al., 2003. Nonetheless, the latter is probably more correct with respect to changes in water masses. Although, here the same comment as above: this is only 1 of the many studies of this area indicative of water mass changes throughout the Holocene. So, add "e.g." and other references (!more recent than 2003?).

P3L2: change "has" by "is"

P3L3: "considered to represent a constant cold …."

P3L3: Do you have a reference for the last part of this sentence?

P3L4-5: Similar comment as before. Include more recent references for the wider area of the Barents Sea.

P3L6: Rewrite. "…2015). In addition for that period, there is…"

P3L7: "…of warm AW inflow towards western Svalbard (e.g. …"

P3L10: Remove "the" before fjord and before AW

P3L16-19: Rewrite sentence!

P3L20: "The aim of this study….

P3L21-24: Rewrite sentence! And again, use more than just 1 study to support this assumption. Plenty of studies from the area to back up this statement.

P3L25: …comprising composed…? These two words in a row does not makes any sense. Revise.

2. Study area

In general, see comments with Figure 1. Also refer to this figure within this chapter.

P4L11: Add brackets around Heleysundet and Freemansundet

P4L16: "Arctic waters as well as mixed…"

P4L16-18: Rewrite this sentence. The current that branches off is not longer the WSC anymore!

P4L19: Explain the 2 passages in the north better. See also comment with Figure 1.

P4L20-21: Add the unit of salinity.

P4L25-28: Rewrite/Revise sentence. Plus add unit of salinity

P4L29-30: "…in Storfjorden is classified as a low-energy and high-accumulation environment, which is characteristic …"

P4L30-32: Rewrite sentence. Explain the last part of the sentence better.

P4L33: "..formation as well as the duration…"

3. Materials and methods

3.1 Sampling

I suggest deleting the subheading "3.1 Sampling" or changing it by "3.1. Marine sediment core"

P5L4: Change "during the cruise" by "retrieved with"

P5L5: Be consistent in using a space or not before degrees (See how you do it within the results chapter)

P5L6: remove space before "of"

P5L7: Extruded? I don´t think this is the correct word here.

*The piston extrusion of sediment core is a common method used in the analyses of marine and lacustrine sediments. The ST_1.5 core was extruded, therefore, we assume that the word is correct.*

P5L9: "extraneous on and/or cross-contamination of the thin…"

P5L8-10: Explain better. Rewrite sentence.

P5L11: change "in" by "at"

3.2 Sediment dating

P5L13: Change "sediment dating" to "Chronology"

P5L14: Change "sediment layeres" by "marine sediment core"

P5L15: Add "retrieved" after layers

P5L16: Add "core depth" after cm

P5L16-17: Rewrite "…from 46.5 cm core depth."

P5L18: Change "in" by "at"

P5L20: Change "in" by "at"

P5L21: Remove space before new sentence.

P5L23-24: Explain why you chose this value for delta R. Plus rewrite sentence as it is not entirely correct as you say it. The here given value represents the local reservoir age (DeltaR) that is applied, rather than the difference… I think? Double check!

P5L25: "…BP) (Table 1)."

*The ΔR value was applied after Mangerud et al. (2006), because it is the most recently published dataset of reservoir ages from North Atlantic region. However, it should be noted that the reservoir age is based on a few data points from western Spitsbergen, and the age may be different for the eastern coast. No data is available for the latter region. The specific value of 105 ± 24 was applied after Łącka et al. (2015). The core analyzed by Łącka et al. (2015) is located in the vicinity of the ST_1.5 core. Unfortunately, the authors of other studies from Storfjorden (e.g. Rasmussen et al., 2007; Rasmussen and Thomsen, 2015; Knies et al., 2017) did not include the ΔR value in their manuscripts.*

3.3 Sediment grain size

P5L27: Change subtitle to "Grain size analysis"

P5L28: "…for grain size…"

P5L29: analyzer… Pay attention to consistent use of z/s (language!)

P5L31-32: Add reference for this method

P5L33: "… was used to reconstruct an ice rafted debris (IRD) record. …"

P6L1-2: "… is reported as concentrations (i.e. the number…."

3.4 Fossil foraminifera

P6L5: I suggest being more specific and thus, changing the subtitle to "Benthic foraminiferal assemblages"

P6L6-7: Rewrite sentence. Same comment as before and thus, be more specific. Literally say what your aim is (i.e. to reconstruct benthic foraminiferal assemblages). Also, make "mesh" plural. You used more than one mesh.

P6L11-12: Similar comment as for Grain size. "…presented as concentration (…) and flux (….)."

3.5 Stable isotope analysis

In general, what about errors for these measurements? In terms of reproducibility. Vital effect corrections?

*We directly compare our isotopic data with the $\delta^{18}O$ record of Łącka et al. (2015) and $\delta^{18}O$-based SST record of Sarthein et al. (2003) (see Fig. 4). Łącka et al. (2015) performed isotopic measurements of E. excavatum tests, while Sarntheinet al. (2003) used C. lobatulus for measurements. In both studies no correction was applied, therefore we also did not apply a correction to our results.*

P6L20: Change "From" by "Ca. "

P6L21: Delete space before new sentence. "… performed using a …"

3.6 Ancient DNA analysis

I suggest giving a bit more information about this proxy in general. As it is a more recently developed proxy, this might be useful to the readers.

*Following the Referee's suggestion, we have added more information about aDNA anlysis to the Introduction.*

4. Results:

Generally, the Results (and later also the Discussion) section should be revised with respect to the

writing style. It contains often the same sentence structure. Something that is getting kind of unpleasant to read.

Furthermore, as they are often wrongly placed and/or used the use of comma´s should be revised. There is no harm in starting a new sentence.

Also, as the chronology is not sufficient enough to claim exact ages, make sure you ALWAYS use "~" when referring to timing/ages.

Then, conjugate your verbs correctly: You might have observed/recorded/noted etc. things in the past; however, your results still ARE what they are as of today. So, make sure to conjugate your verbs in the present tense when necessary.

4.1 Sediment age and type:

P7L20: What is meant with sediment type?

P7L21-22: Rewrite: "All dates were recorded in a chronological order…."

P7L22: Delete "depths of"

P7L23: Add "core depth" after 5.5 cm

P7L23: Delete "that were"

P7L24: Place commas after "was" and after "therefore"

P7L24-28: You say here "three remaining dates". However, even though you don´t present data between 4000 and 9000 years ago later in this study, from Figure 3 it is clear that you still used the last date in your age model. In addition, from Figure 3 it also seems you still use the first date to construct your age model. Hence, that looks like you actually use 5 data points (incl. linear interpolation). I think you should explain this better, both here within the text as well as later in Figure 3 and Table 1.

*Indeed, the age model is based on four radiocarbon dates. The text was corrected.*

*As explained in the legend in Figure 3, $^{14}C$ dates are marked with grey silhouttes and the dotted line shows the age-depth model retrieved from the linear interpolation between the dates. The additional explanation has been added to the figure caption.*

*The date from 2.5 cm was not calibrated, therefore, we decided not to include both calibrated and non-calibrated dates in the age model. We only assume that the top of the core represents modern age.*

P7L26: Change "time" by "temporal"

P7L27: Rewrite …precluded the making of…

P7L27-28: Change "the manuscript" by "this study"

P7L29: Comment here and related to many examples later on. When describing your results be careful with the used tense. Your results still "are" what they are, so no need to write in the past tense about them. Hence, change "was" by "is".

P7L29: Rewrite …An approximately…

P7L30: Rewrite ….where it increases…

P7L31: Rewrite ….decreases…

P7L32: Change "was" by "is"

P7L33: Rewrite ...The IRD flux decreased slightly with….

P8L1-2: ….one peak reaching 0.8 grains g-1 cm-1 at ~ 2.6 cal ka BP.

P8L3: … fraction had its highest…

P8L4: …after ~ 2.4 cal ka BP…

NOTE: For chapter 4.2 to 4.5, (and later also for the discussion) the conjugation of verbs should be double checked as I will not any longer do this in the further review.

4.2 Stable isotopes

Revise this chapter based on general comments stated below results.

4.3 Fossil foraminifera

P8L18: I suggest rethinking this sub-title. From the scope of the journal it is quite obvious this study deals with fossil foraminifera. I would be more specific here in what you present (i.e. benthic foraminiferal assemblages?)

P8L19: A total of 8647 specimens? I assume, this is the total for all samples? However, in which way is this relevant for this study?

P8L20-24: Rewrite sentence. Possibly split into two sentences. Make it more clear what you are trying to say.

P8L23-24: …There are a few… foraminifera recorded. In particular, these peaks lay at ~ 2.0 and ~ 1.8 cal ka BP as well as at the sediment….

P8L24: ….37, 37 and 66%, …

P8L23-25: If you refer here to Figure 4. I would make this also more visible within Figure 4. Right now, it is not clear what you are referring to. Or explain better.

P8L29: …. 2524, 2584 and 2610 ind. g-1, respectively. …

Note: No need to repeat the unit in a summary like this. Apply this within the further manuscript as it occurs more often

P8L30-32: Rewrite sentences more clearly. Clarify that the flux reached 2.2 for both peaks.

P9L3: ….at ~2.3 …

P9L5-9: Rewrite sentence. Too long.

P9L9: Delete "After ~ 2.7 cal ka BP, there were"

P9L10: Add "were" between peaks and recorded

P9L11-15: Rewrite sentence. Too long, too many commas.

4.4 Foraminiferal aDNA sequences

P9L21: A total of … Why is this number relevant?

P9L31-34: Rewrite sentence. ! Grammar.

P10L1-9: Pay attention to the repetition of the same sentence structure.

4.5 Diatom aDNA sequences

P10L12: Same comment as before regarding the total.

P10L17: Delete space before start of new sentence

P10L18: … recorded at ~ 0.4 …

Note: revise entire manuscript wrt adding "at" when you refer to a certain moment in time.

P10L24: Delete "." before start of new sentence.

P10L26: Rewrite sentence. No need for a comma in a sentence this short.

6. Discussion

P10L28: You jump from chapter 4 Results to chapter 6 Discussion?

P10L29-30: Rewrite: …a linear interpolation between four AMS 14C dates and thus, the age control….

However, which fourth date is included now? 2.5 cm or 52.5 cm? Clarify better throughout the manuscript.

P10L30-32: What is meant with this statement? Explain!

P10L34: …correlates…

P10L33-P11L2: Refer to figure 4! With respect to this statement: I can see the maxima in core 23258. However, the minimum in this core is not clear to me. In addition, the temperature minimum and maxima in the GISP2 core are not clear to me at all… So, please explain this statement?

Furthermore, what is the reasoning behind the comparison of this record with GISP2? Why comparing it to a Greenland ice core record if you further keep it (geographically speaking) quite local? If comparing it to a Greenland ice core, then why GISP2 and not one of the others?

In addition to this, why did you picked core 23258 for comparison at this point? In your Figure 1, you show the location of several other marine cores, but not the one from Sarnthein et al., 2003. Why? Why adding 23258 to Figure 4 and not one of the records you have added in figure 1, and vica versa?

*The core of Sarnthein et al. (2003) is broadly discussed in the manuscript. The timing of AW inflows revealed by their record is a basis of considerable part of our discussion, thus we decided to directly compare the record of Sarntheim et al. (2003) and our proxy record. Also, we have added the location of the 23258 core to Fig. 1.*

*We included GISP2 core in the manuscript to show the correlation between our results and other, geographically broader, studies. We agree that the manuscript is focused on the local environmental changes, therefore, we decided to remove the information about GISP2 core as not relevant for our study.*

Also, I think the correct label of the core presented by Sarnthein et al., 2003 is M23258… double check this.

*In the manuscript of Sarnthein et al. (2003), the label 23258-2 is used for the core, while the coring station is labeled 23258.*

P11L2: Add an enter after this paragraph

6.1 The period from 4 cal ka BP to 2.7 cal ka BP
P11L3: Rewrite subtitle: "Time interval between ~ 4 and ~2.7 cal ka BP"

General comment to this sub-chapter:
This sub-chapter is based on the data retrieved from 2 samples! I genuinely doubt how realistic it is to say this much and interpret environmental conditions for ca. 1300 years based on 2 data points. And thus, if these 2 data points do not reflect 2 cm of sediment (but 4 cm?) I strongly recommend doubling the resolution. Even tough then, it will still be a low temporal resolution, but at least, slightly improved.
*We agree that adding two data points may slightly improve resolution. However, it is not possible to obtain additional aDNA data, which is an essential part of the manuscript. In our opinion, adding two more data points of selected proxies will not allow to make any general conclusion about the period prior to ~ 2.7 cal ka BP. Instead, we decided to shorten significantly the discussion about this period and make conclusions more general.*

P11L4: Delete "During the period"
P11L5: Delete "variable" This are only 2 data points…
P11L5: What is meant with "coarsening"? more coarse compared to what? Why not just say what it is? "A relatively low mean 0-63-um fraction"
P11L7: Rewrite. "…IRD peaks were noted during the Neoglacial and attributed…"
P11L7: Be more specific on the timing of "during the Neoglacial"
P11L8: Rewrite. "… rafting resulting from glacial front fluctuations (e.g. …"
P11L8: Rewrite. "0-63-μm"
NOTE: Be consistent in the use of the hyphen between 63 and μm. It is mostly there, but not always!
P11L10: What is meant with "response"? Explain.

P11L11: …is dominated…

P11L13: …is associated with cool and salty AW…

P11L19: "." at the end of the sentence

P11L21: Explain "standing stock"

P11L24: "noted a decrease" A decrease compared to when? After/Prior to 2.7 ka? How much is meant with a decrease? Explain/describe this decrease better…

P11L28: Explain what is meant with "high-energy"

P11L29-31: For honesty´s sake, I would mention this in the beginning of the sub-chapter (see general comment to this sub-chapter).

6.2 The period after 2.7 cal ka BP….

I´m not convinced about the sub-chapters titles…. I suggest revising them. Try to make it more concise.

*The sub-chapters titles were more concise in the first version of the manuscript. However, one of the previous Referees suggested to extend them and add more information to emphasize the most important information included in the sub-chapter. Therefore, the sub-titles became more descriptive.*

P12L3-34: What is the main message of this paragraph? Make that more clear for the reader.

P12L13: Rewrite. "…IRD in ST_1.5 may…"

P12L19: Rewrite. "Both increased ice cover… delivery limit light…"

P12L20: Add a comma after therefore.

P12L22: Replace ";" by a "." Then start a new sentence "The latter may…"

P12L27-30: Rewrite sentence. Also, why referring to this study and not others from the Barents Sea.

P12L31-P13L2: Why not including sea ice reconstruction studies from the wider area (Barents Sea, east of Svalbard)? See general comments.

P13L3: "Knies et al. (2017) suggested…"

P13L6-9: Rewrite sentence. Not clear and a lot of "and´s"

P13L11: "…2.3 and 1.7 cal …melting of the sea ice cover,…"

P13L13: "…by light d18O peaks in benthic…."

P13L17: "…was dominated by both AW/…"

P13L20: "…2.3 and …"

P13L21: "…BP are M. barleeanum…"

P13L23-27: Rewrite sentence. Too many thoughts in one sentence.

P13L33: "…in the 0-63-μm…"

P14L1-2: This sentence is key for this study. So emphasize it more.

P14L2-5: Rewrite sentence.

P14L7: "…in ST_1.5 belong…"

P14L9: "Y were previously noted"

P14L12: Remove ";" and start new sentence. Place hyphen between so and called.

P14L13: "taxa are known"

P14L17: Remove ";" and start new sentence.

P14L20: Remove ";" and start new sentence.

P14L22: "…inflow at ~ 2.4 and ~ 1.7 cal ka BP. Furthermore, the…"

P14L27: "…sp. is commonly found…"

P15L2: Change "have been" by "were"

P15L13: "The decrease…" I would not call it a very pronounced decrease… So verbalize this more carefully.

P15L21-23: What do you exactly try to say here? The high abundances indicate declining sea ice cover OR the highly productive surface waters indicate declining sea ice cover? Clarify.

P15L25: Remove ";" and start new sentence.

P15L28: Add enter after this paragraph.

6.3 Paleoceanographic implications

P15L30: "…revealed a two-phase Neoglacial with…"

P15L31: "ST_1.5 proxy records,…"

P15L32: "…constant cold…

P16L1: When you refer to "evidence" … Due to the limited data points prior to 2.7 ka, I would back this up by more evidence from the literature from a wider geographical area. However still within the vicinity of your core site… And thus, as it is content-wise a bit "thin", I suggest to revise and expand this sub-chapter (see comments Figure 1). I

P16L1: "…region during the mid-Neoglacial…."

P16L2: "An alkenone…."

P16L3-6: Rewrite sentence.

7. Conclusions

P16L18: "…steered controlled…" 2 verbs? Pick one.

P16L19: "and sea-ice cover variability. …"

P16L22: "…formation of an extensive…"

Figures:

In general, figure captions are rather "thin". I suggest adding a bit more information to the figure caption in order to make it clearer what is presented.

Figure 1:

With respect to the Introduction/Study area as well as the Discussion later on, I consider this figure as too specific. And thus, I suggest to add a "geographically broader" figure into this figure. Have to panes: a) broad study area; b) a zoom of the fjord (figure as it is now). This will also allow you to add more reference cores used for the discussion within this figure. For example, I suggest to add the cores referred to in the discussion also within this figure.

Furthermore, now it is indicated that red means WSC. However, the current branched off and flowing into the Barents Sea is not any longer the WSC. So add the correct name.

Add reference of ODV in the figure caption.

Rewrite figure caption. Add more info to it as well.

Figure 2:

P27L2: Change "sampling station" by "core location"

Figure 3:

P2L8: Give the core name instead of "studied core"

Is the date at 2.5 cm used? It seems it is. More particular, 2.5 cm is given present day age and further used for the linear interpolation between present and the next dating point. Then, also indicate this in this figure but also in the text and in table 1.

Further, indicate in this figure the difference between foram versus shell dated data points

Figure 4:

Figure caption: Specify on which foraminiferal tests the isotope analysis has been executed (in the caption OR in the figure itself). Specify this study present benthic foraminifera (and not planktic).

Figure: Indicate on the X-axis the location of your 14C dates. (Also do this for Figure 4-5-6-7-8)

Be consistent in the labeling of your units (fe. Xxxx/g vs. XXX g-1). (Also for other figures)

Add hyphen for the grain size label.

Add reference of GISP 2 data.

Further I suggest to label the separate plots by a, b, c, etc. and then also refer to the figure more specifically within the manuscript. This will make it easier for the reader to follow. (Also do this for Figure 4-5-6-7-8)

Figure 5:

Figure: Atlantic Water with a capital letter as has been done within the manuscript.

Figure 6:

P30L4: "a dashed line"

Further, I´m a bit confused. It has been sad that every 5 cm of the core was sampled for DNA analyses. Which makes 12 samples for the entire core. Here, only the data till 3.3 ka is presented. So, how come you still have 12 data points?

*There is a typing error in the text, the resolution is 4 cm – it's now corrected. Moreover, sampling depths are presented in the supplementary tables.*

I also suggest to add the dotted lines indicating 2.7, 2.3 etc. similar as has been done for Figure 4&5 (This also counts for figure 7 & 8)

*We agree that the lines indicating major environmental changes may be helpful. We have added the lines to Figures 5, 6, and 7.*

Figure                                                                                                          7:
Please explain a little bit in the figure caption what these clades mean.

*What do you actually mean here? Are you asking about definition of clade or the description of environmental preferences etc. of certain clade?*

*Certain taxa are assigned to clades based on the their phylogenetic relationships. It means that sequences belonging to different clades are more "genetically distant" than sequences belonging to the same clade. Allogromiids are known to be not a coherent taxonomic groups, but are scattered between different clades (for more details, see Journal of Foraminiferal Research 32(4), 2002).*

Further, wrt presenting your data in a consistent manner… Why did you switched to present the age on the Y-axis? I suggest presenting it on the X-axis as you did for the previous figures (This also counts for figure 8).

Figure 8:
P31L6: "….taxa plotted versus age" (Be consistent with other figure captions. "… a dashed…"

Table 1:
Change "sediment depth" by "core depth"
Be consistent: Calibrated years BP vs. Cal. a BP
Add information (possibly a new column) wether the dated material is on bivalve (shell) or benthic foraminifera…

Make clear the depth at 2.5 cm is given present day age.
Why the bold dark line between the last two rows? If you want to keep it like this, then explain this line within the figure caption.

[revised manuscript text omitted]

Core sites:
1 = JM02-460GC/PC (Rasmussen and Thomsen, 2015)
2 = JM10-12GC (Rasmussen and Thomsen, 2015)
3 = NP05-86GC (Rasmussen and Thomsen, 2015)
4 = JM10-10GC (Rasmussen and Thomsen, 2015; Knies et al., 2017)
5 = JM09-020-GC (Łącka et al., 2015)
6 = 23258-2 (Sarnthein et al., 2003)
7 = JM09-KA11-GC (Berben et al., 2014)
8 = NP05-11-70-GC (Berben et al., 2017)
9 = PSh-5159N (Rasdobakken et al., 2016)

**Figure 1:** The modern oceanography of the study area (A) and the location of the studied core ST_1.5 (B) and the other cores discussed in this paper (A,B). Abbreviations of main surface currents: WSC – West Spitsbergen

[Figure]

**Figure 2:** Temperature and salinity profile from the sampling station. Temperature is marked with a dashed line, and salinity is marked with a black line. Abbreviations: AW – Atlantic Water, TAW – Transformed Atlantic Water, BSW – Brine-enriched Shelf Water.

[Figure]

**Figure 3:** Age–depth model of the ST_1.5 core. The grey silhouettes show probability distribution of calendar
dates that were obtained by calibration of individual [14]C dates used for the age model. The dotted line shows the
age-depth model derived from a linear interpolation between the dates.

[Figure]

**Figure 4:** Sedimentological and micropaleontological data plotted versus age. The sediment accumulation rate
(SAR), mean grain size of the 0-63-µm fraction, ice-rafted debris (IRD) flux and number of grains per gram of
sediment, oxygen (δ[18]O) and carbon (δ[13]C) stable isotopes in benthic foraminiferal tests, and the flux and
abundance of foraminifera are presented.

[Figure]

**Figure 5:** The abundance (expressed as the number of individuals per gram of dry sediment) and the percentage of the dominant benthic foraminifera.

[Figure]

2 **Figure 6:** The dominant components of the monothalamous assemblages. The abundance is expressed as the
3 percentage of the monothalamous sequences and the most abundantly sequenced taxa are presented. The trend is
4 indicated with a dashed line.

[Figure]

2  **Figure 7:** The percentage share of certain clades in the allogromiid sequences.

[Figure]

3  **Figure 8:** The percentage of sequences of dominant diatom taxa vs. time. The trend is indicated with a dashed

4  line.

**Table 1:** Raw and calibrated AMS$^{14}$C dates used in the age model. B stands for bivlave shells, while F stands for benthic foraminifera tests.

| Core depth [cm] | Material | Raw AMS $^{14}$C | Cal. a BP ± 2σ | Cal. a BP used in age model |
|---|---|---|---|---|
| 2.5 | *Nuculana pernula* (**B**) | 107.38 ± 0.33 pMC | - | - |
| 5.5 | *Yoldiella lenticula* (**B**) | 290 ± 30 BP | - | - |
| 14.5 | *Turitella erosa* (**B**) | 2020 ± 30 BP | 1356-1555 | 1500 |
| 43.5 | *Yoldiella solituda* (**B**) | 3010 ± 50 BP | 2484-2787 | 2700 |
| 46.5 | *Nonionellina labradorica* (**F**) | 4490± 40 BP | 4400-4701 | 4500 |
| 52.5 | *Yoldiella lenticula* (**B**) | 7545 ± 35 BP | 7803-7989 | 7890 |

---

## Author Response (AR3)

Dear Editor,

Thank you for your comments that helped to improve the manuscript. We have checked the manuscript carefully and modified it according to Editor's suggestions. Written below are Editor's comments followed by our responses.

As for the response to second Reviewer: We have followed all the suggestions of language corrections/ text modifications. We tried to avoid writing similar response that the text has been corrected after every Reviewer's comment. Therefore, we stated at the beginning of the "Specific comments" section, that in general, we have followed the Reviewer's suggestion. Our additional responses were added only when broader explanations were needed. However, some of the specific comments were not applicable anymore, because the manuscript has been significantly changed according to the general comments of both Reviewers. Therefore, I was not able to reply to some comments, because parts of the text mentioned by the Reviewer were modified or removed.

Kind regards,
Joanna Pawłowska

P1 L1: Delete the last part of the title. You use eDNA to complement the other proxies, but as you say in your response, if it can be used in this way or not have already been discussed in other papers, and is therefore not an essential part of your discussion - as it is indicated to be from the present title.
*According to the suggestion, we have deleted the second part of the title. Now the title is as follows: Multiproxy evidence of the Neoglacial expansion of Atlantic Water to eastern Svalbard*

P1 L21: General cooling: From what? Your record start at 4 ka so how can you say that a cooling took place? Cooling also indicate a trend, but you don't really have data to claim a cooling trend 4-2.7 ka either.
*We agree with this comment, thus we have used "cold conditions" instead of "general cooling".*

P1 L28: Reveal the timing: no. support the occurrence of the AW events, but cannot reveal the timing of the events.
*According to the suggestion, we have modified the sentence. Now the text is as follows: The ancient environmental DNA (aDNA) records of foraminifera and diatoms support the occurence of the major pulses of AW (~2.3 and ~1.7 cal ka BP) and the variations in sea-ice cover.*

P2 L34: sea-ice cover in West Spitsbergen: in the Fram Strait (or at the Weast Spitsbergen margin). Cannot have sea ice on land.
*Indeed. We have used „West Spitsbergen margin" instead of "West Spitsbergen".*

P3 L3: Sarnthein argues for change in ESC and/or Jan Mayen Current, so not as clear a support as indicated here. Modify/take into account.
*We have added the information about Jan Mayen Current to the text.*

P3 L9: Risebrobakken et al., 2010 fit better as a reference above.
*We agree with this comment. The reference have been moved to the previous sentence.*

P4 L25: It's a very local description of the oceanography. The paper, and your Section 5.4. would benefit from adding a few lines on the broader oceanographic setting.
P4 L31: Wrong. The North Cape Current enters the Barents Sea through the Bear Island Trough, not Storfjorden, e.g. Loeng et al., 1991, Polar Research.
Line 28-34: Add references.
*We have followed the Editor's comments and corrected the existing description of the study area and added broader information about the water masses in the Barents Sea.*

P5 L22: Reference to Fig 2?
*The reference has been added to the text.*

P5 L24: Extruded: I know you responded to the reviewer with respect to the use of this word, but did you really press the sediment out of the core and sample as it was pressed out instead of splitiing the core befoere sampling?
*Yes, it is exactly what we did. The sediment was pressed out of the core and the core was not split.*

P6 L29: AW/frontal zone indicators: Why not separate indicators of AW and front? If possible that would add very valuable information.
*In the manuscript, we have followed the classification proposed by Majewski et al. (2009). To be consistent in our classification, we have followed they suggestion and did not separate AW and frontal zones indicators.*

P8 L2: Would recommend to separate into two subsections (4.1 Chronology and 4.2 Sediment grain size).
*According to the Editor's suggestion, we divided the section 4.1 into two subsections.*

P8 L7: How did you constrain the top of the core? Please specify, e.g. if you used the year of coring (when was that?).
*Indeed, we assumed that the top of the core represents the year of coring, i.e., 2014.*

P8 L9: The sedimentation rate is equally low from 4 to 2.7 ka as it was prior to 4 ka, but you still include this interval?
*The period from ~ 7.9 cal ka BP to ~ 2.7 cal ka BP includes only 5 data points, which is clearly not enough to provide any reliable information about almost 5 ka years long period. However, based on the AMS $^{14}$C dating, we were able to define the beginning of the Neoglacial, which is known to be a period of significant oceanographic changes in the Arctic. Therefore, we decided to focus on this time period in our manuscript.*

P8 L26: relative stable - as the reviewer noted, it does not make a lot of sense to talk about stability between two data points. Would delete the sentence.
*We agree with this comment. Now the sentence is as follows: The $\delta^{18}O$ values prior to ~ 2.7 cal ka BP changed slightly between 3.55‰ and 3.69‰ vs. VPDB.*

P9 L3: number of foraminiferal individuals in a sample varied?

*Yes, the numbers refer to the number of foraminifera in a sample. The sentence has been corrected.*

P9 L5: Looks like the low was between 2.2 and 1.7 ka?
*Indeed. The sentence has been corrected.*

P9 L28: What about the overall max in glacimarine species 1.7-0.5 ka?
*The following text has been added to the manuscript: Also, the overall maximum of glaciomarine species abundance was recorded between 1.7 and 0.5 cal ka BP, ranging from 25% to 43% of foraminiferal assemblage (Fig. 5).*

P9 L10: For me being a non expert on eDNA teh explanations of allogromiids and clades are not clear. Since eDNA still is a new method for paleostudies I think it would be benefitial to add clear definitions of these associated terms.
*Following the Editor's suggestion, we have added following text to the manuscript: Herein, the term "allogromiid" refers to monothalamous foraminifera with organic or predominantly organic test walls (Gooday, 2002). Morphological and molecular evidence indicate that 'allogromiids' are not a coherent taxonomic group but are scattered between several monothalamous clades (Pawlowski et al., 2002). "Clade" refers to phylogenetic clades defined by molecular data. The clade is traditionally defined as a group of organisms that includes a common ancestor and all the descendants.*

P12 L10: Decrease in the dominant grain size within the 0-63 fraction?
*Herein, we referred to the mean grain size of the 0-63 fraction. The sentence has been corrected.*

P 13 L9: What about the timing of these events relative to yours?
*The studies cited below were did not indicate the timing of AW inflows as precisely as Sarnthein (2003). However, some of them indicated a change of environmental conditions ~ 2.7/2.5 cal ka BP, which is similar to our record. Risebrobakken et al. (2010) suggested increased influence of AW in the SW Barents Sea between 7.5 and 2.5 cal ka BP. After 2.5 cal ka BP, they postulated cooling of surface waters with a presence of AW and/or chilled AW near the bottom. Berben et al. (2014) postulated stable and strong AW inflow to the western Barents Sea during the mid-late Holocene (7.5 cal ka BP – 1.1 cal ka BP). In the period after 1.1 cal ka BP, they suggested increased seasonal inflow of AW and more extensive seasonal sea ice cover. In contrast, northern Barents Sea experienced surface water cooling and more extensive sea-ice cover prior to 2.7 cal ka BP. The increasing influence of AW was observed after 2.7 cal ka BP (Berben et al., 2017). Also Müller et al. (2012) postulated temporarily strengthening WSC in the last 3,000 years, resulting in fluctuating sea ice margin along the West Spitsbergen coast.*

P13 L11: I cannot see any clear indications of depleted d18O associated with the peak in foraminiferal flux. Why is it surprising that low d18O values are associated with AW? In the larger Nordic Seas region the AW has more depleted d18O values than in the Arctic and Polar water masses (e.g. Johannessen et al., 1994). Melt water would deplete the d18O signature of the surface layer and could be transfered to the bottom via brine formation, however, I do not really see any large responses in your d18O record associated with these two events.
*We agree that the changes in the $d^{18}O$ record are not large and the response is not really clear. The text considering the $d^{18}O$ have been removed.*

P4 L14: was found
*The sentence has been corrected*

P15 L13: isotopically heavier - heavier relative to  what?
*The expression "isotopically heavier" have been removed from the sentence, as it may be confusing.*

P16 L24: Duplessy and Lubinski - North Barents Sea; maybe northwest, but for sure north.
*Indeed, these studies were conducted in the northern Barents Sea. The text has been corrected.*

P26 L34: Mentioned before, here or in the text, make sure you don't have to be an expert in environmental DNA to understand what is ment by clades and allogromiid sequences.
*Following the Editor's suggestion, we have added the explanation of the terms "clade" and "allogromiids" to the text. We prefer not to duplicate it in the figure caption, to avoid making the caption too long and not easy to read.*

P26 L34: Check the resolution of the figures, especially fig 4 and 5 is not clear.
*Sampling points on both figures are marked by dots. Unfortunately, the quality of graphics pasted to the .doc file is rather low, so they are not clearly visible. They should be better visible on the high-quality graphics in the final version of the manuscript. Additionally, the dots have been made bigger to ensure that they will be well visible.*

P30 L1: Delete the smoothing (d13C and d18O). Invert the y-axis for the d18O records.
*The figure has been modified according to the suggestion.*

P31 L1: Absolute abundance. Make sure its clearly visible what levels you have samples from. The shading may give the impression of a better resolution than you have, especially for the early part of the record.
*As explained above, sampling points are marked in both fig. 4 and fig. 5.*

P32 L2: Provide information on how the trend is defined.
*The trend line is 2-point average. The information has been added to the figure caption.*

[revised manuscript text omitted]

---

## Author Response (AR4)

Dear Editor,

Thank you for your comments and time taken to improve our manuscript. We have modified the manuscript according to the Editor's suggestions. Written below are the Editor's comments followed by our responses (in italics).

Kind regards, Joanna Pawłowska

Page 4. line 32: None of these references focus on the oceanography upstream of the Barents Sea. Rudels do not mention the NAC, while Loeng mention the Norwegian Atlantic Current (NwAC). Most often the NAC (North Atlantic Current) is used for the North Atlantic, and then it transfers into the NwAC when entering the Nordic Seas. Please add a more relevant reference.

We agree that the name Norwegian Atlantic Current is more appropriate. We have changed the name in the text and in the figure 1. Moreover, the reference to Rudels et al. (2015) has been replaced by Blindheim and Østerhus (2005).

Page 8. Line 31: I am glad you longer say its stable, but I don't really think you can say much about changes either. Maybe the most honest thing will be to be very specific and say something like "The two d18O data points prior to 2.7 cal ka BP record values of 3.55‰ and 3.69‰ vs. VPDB".

This feeds back to comments raised by the reviewers as well, relating to how you cannot really say anything about the time interval prior to 2.7 cal ka based on your data. The change at ca. 2.7 is well known from the Barents Sea, and therefore it makes sense to show that part combined with references to how low sedimentation rates are a common feature for large parts of the western Barents Sea through the mid Holocene. But it does not make sense to make any statements on stability of variability/changes taking place based on those two data points.

Indeed, writing about trends between two data points makes no sense. We have checked the manuscript and removed/modified all the statements referring to variability/stability of proxies before 2.7 cal ka BP.

Page 13, line 14-20 and your response to past page 13, line 9: Yes, several records show a change at ca. 2.7 ka, as you do. Several of these records also show quite some variability within the last ca. 2.7 ka, even if the papers focuses on the longer term trends/conditions (e.g. Groot et al., 2014; Berben et al., 2014; 2017; Risebrobakken et al., 2010). Especially, the studies by Groot and Berben are close to your site, and shows repeated variability e.g. in benthic foraminiferal flux and sea ice through the last 2.7 ka. They may not go in detail with respect to these events, due to a different focus of their work, however, I would still expect a discussion on how your events are related to the overall variability of relevance in your study area, not just the one records that looks most similar. What happens at your site is not independent of what happens at nearby sites.

We have followed the Editor's suggestion and added broader description of environmental changes recorded in the Barents Sea in the mid-late Holocene. The added text is as follows:

The AW inflow to the Barents Sea was relatively stable during the mid-Holocene. The environmental conditions became more unstable in the late Holocene, with periodic cooling of surface waters, a presence of AW and/or chilled AW near the bottom, and more extensive seasonal sea ice cover (Risebrobakken et al., 2010; Berben et al., 2014; Groot et al. 2014). The timing of these changes differed between the study settings: in the western Barents Sea, it was ~ 1.1/1.5 cal ka BP (Berben et al. 2014; Groot et al., 2014), while in the southwestern Barents Sea, the change in environmental conditions was recorded ~ 2.5 cal ka BP (Risebrobakken et al., 2014). In contrast, the northern Barents Sea experienced surface water cooling and more extensive sea-ice cover prior to 2.7 cal ka BP. The increasing influence of AW was observed after 2.7 cal ka BP (Berben et al., 2017).

Figure 5 and 4. Please add dots where you have measurements not only in the abundance plots but also in the flux plots.

The dots have been added to the figures.